# The Semi-Random Satisfaction of Voting Axioms

**Lirong Xia**
RPI, Troy, NY 12180, USA
`xialirong@gmail.com`

## Abstract

We initiate the work towards a comprehensive picture of the worst average-case satisfaction of voting axioms in semi-random models, to provide a finer and more realistic foundation for comparing voting rules. We adopt the semi-random model and formulation in [54], where an adversary chooses arbitrarily correlated "ground truth" preferences for the agents, on top of which random noises are added. We focus on characterizing the semi-random satisfaction of two well-studied voting axioms: *Condorcet criterion* and *participation*. We prove that for any fixed number of alternatives, when the number of voters $n$ is sufficiently large, the semi-random satisfaction of the Condorcet criterion under a wide range of voting rules is $1$, $1 - \exp(-\Theta(n))$, $\Theta(n^{-0.5})$, $\exp(-\Theta(n))$, or being $\Theta(1)$ and $1 - \Theta(1)$ at the same time; and the semi-random satisfaction of participation is $1 - \Theta(n^{-0.5})$. Our results address open questions by Berg and Lepelley [3] in 1994, and also confirm the following high-level message: the Condorcet criterion is a bigger concern than participation under realistic models.

## 1   Introduction

The *"widespread presence of impossibility results"* [48] is one of the most fundamental and significant challenges in social choice theory. These impossibility results often assert that no "perfect" voting rule exists for three or more alternatives [1, 28, 45]. Nevertheless, an (imperfect) voting rule must be designed and used in practice for agents to make a collective decision. In the social choice literature, the dominant paradigm of doing so has been the *axiomatic approach*, i.e., voting rules are designed, evaluated, and compared to each other w.r.t. their satisfaction of desirable normative properties, known as *(voting) axioms*.

Most definitions of dissatisfaction of voting axioms are based on worst-case analysis. For example, a voting rule $r$ does not satisfy CONDORCET CRITERION (CC for short), if there exists a collection of votes, called a profile, where the *Condorcet winner* exists but is not chosen by $r$ as a winner. The Condorcet winner is the alternative who beats all other alternatives in their head-to-head competitions. As another example, a voting rule $r$ does not satisfy PARTICIPATION (PAR for short), if there exists a profile and a voter who has incentive to abstain from voting. An instance of dissatisfaction of PAR is also known as the *no-show paradox* [19]. Unfortunately, when the number of alternatives $m$ is at least four, no irresolute voting rule satisfies CC and PAR simultaneously [39].

While the classical worst-case analysis of (dis)satisfaction of axioms can be desirable in high-stakes applications such as political elections, it is often too coarse to serve as practical criteria for comparing different voting rules in more frequent, low-stakes applications of social choice, such as business decision-making [5], crowdsourcing [34], informational retrieval [33], meta-search engines [14], recommender systems [51], etc. A decision maker who desires both axioms would find it hard to choose between a voting rule that satisfies CC but not PAR, such as Copeland, and a voting rule that satisfies PAR but not CC, such as plurality. A finer and more quantitative measure of satisfaction of axioms is therefore called for.

35th Conference on Neural Information Processing Systems (NeurIPS 2021).

One natural and classical approach is to measure the likelihood of satisfaction of axioms under a probabilistic model of agents' preferences, in particular the independent and identically distributed (i.i.d.) uniform distribution over all rankings, known as *Impartial Culture (IC)* in social choice. This line of research was initiated and established by Gehrlein and Fishburn in a series of work in the 1970's [24, 26, 25], and has become a "*new sub-domain of the theory of social choice*" [13]. Some classical results were summarized in the 2011 book by Gehrlein and Lepelley [27], and recent progresses can be found in the 2021 book edited by Diss and Merlin [13].

While this line of work is highly significant and interesting from a theoretical point of view, its practical implications may not be as strong, because most previous work focused on a few specific distributions, especially IC, which has been widely criticized to be unrealistic (see, e.g., [42, p. 30], [23, p. 104], and [30]). Indeed, conclusions drawn under any specific distribution may not hold in practice, as "*all models are wrong*" [9]. Technically, characterizing the likelihood of satisfaction of CC and of PAR are already highly challenging w.r.t. IC, and despite that Berg and Lepelley [3] explicitly posed them as open questions in 1994, not much is known beyond a few voting rules. Therefore, the following question largely remains open.

**How likely are voting axioms satisfied under realistic models?**

The importance of successfully answering this question is two-fold. First, it tells us whether the worst-case violation of an axiom is a significant concern in practice. Second, it provides a finer and more quantitative foundation for comparing voting rules.

We believe that the *semi-random models* [15], originally investigated by Blum [7] and [8] and includes the models employed in the celebrated *smoothed analysis* proposed by Spielman and Teng [49], provides a promising framework for addressing the question. In this paper, we adopt the semi-random model by Xia [54], which models the satisfaction of a *per-profile* voting axiom $X$ by a function $X(r, P) \in \{0, 1\}$, where $r$ is a voting rule and $P$ is a profile, such that $r$ satisfies $X$ if $\min_P X(r, P) = 1$. Let $\Pi$ denote a set of distributions over all rankings over the $m$ alternatives (denoted by $\mathcal{L}(\mathcal{A})$), which represents the "ground truth" preferences for a single agent that the adversary can choose from. Let $n$ denote the number of agents. Because a higher value of $X(r, P)$ is more desirable to the decision maker, the adversary aims at minimizing expected $X(r, P)$ by choosing $\vec{\pi} \in \Pi^n$—the profile $P$ is generated from $\vec{\pi}$. The semi-random satisfaction of $X$ under $r$ with $n$ agents, denoted by $\widetilde{X}_\Pi^{\min}(r, n)$, is defined as follows [54]:

$$\widetilde{X}_\Pi^{\min}(r, n) \triangleq \inf_{\vec{\pi} \in \Pi^n} \Pr_{P \sim \vec{\pi}} X(r, P) \tag{1}$$

Notice that agents' ground truth preferences can be arbitrarily correlated, while the noises are independent, which is a standard assumption in the literature and in practice [54].

**Example 1 (Semi-Random CC under plurality).** *Let $X = $ CC and $r = \overline{Plu}$ denote the irresolute plurality rule, which chooses all alternatives that are ranked at the top most often as the (co-)winners. Suppose there are three alternatives, denoted by $\mathcal{A} = \{1, 2, 3\}$, and suppose $\Pi = \{\pi^1, \pi^2\}$, where $\pi^1$ and $\pi^2$ are distributions shown in Table 1.*

*Then, we have $\widetilde{CC}_\Pi^{\min}(\overline{Plu}, n) = \inf_{\vec{\pi} \in \{\pi^1, \pi^2\}^n} \Pr_{P \sim \vec{\pi}} CC(\overline{Plu}, P)$. When $n = 2$, the adversary has four choices of $\vec{\pi}$, i.e., $\{(\pi^1, \pi^1), (\pi^1, \pi^2), (\pi^2, \pi^1), (\pi^2, \pi^2)\}$.*

|  | 123 | 132 | 231 | 321 | 213 | 312 |
|---|---|---|---|---|---|---|
| $\pi^1$ | 1/4 | 1/4 | 1/8 | 1/8 | 1/8 | 1/8 |
| $\pi^2$ | 1/8 | 1/8 | 3/8 | 1/8 | 1/8 | 1/8 |

Table 1: $\Pi$ in Example 1.

*Each $\vec{\pi}$ leads to a distribution over the set of all profiles of two agents, i.e., $\mathcal{L}(\mathcal{A})^2$. We have $\widetilde{CC}_\Pi^{\min}(\overline{Plu}, 2) = 1$, because CC is satisfied at all profiles of two agents. As we will see later in Example 3, for all sufficiently large $n$, $\widetilde{CC}_\Pi^{\min}(\overline{Plu}, n) = \exp(-\Theta(n))$.*

## 1.1 Our Contributions

We initiate the work towards a comprehensive picture of semi-random satisfaction of voting axioms under commonly-studied voting rules, by focusing on CC and PAR in this paper due to their importance, popularity, and incompatibility [39]. Recall that $m$ is the number of alternatives and $n$ is the number of agents. Our technical contributions are two-fold.

**First, semi-random satisfaction of** CC **(Theorem 1 and 2).** We prove that, under mild assumptions, for any fixed $m \geq 3$ and any sufficiently large $n$, the semi-random satisfaction of CC under a wide range of voting rules is $1$, $1 - \exp(-\Theta(n))$, $\Theta(n^{-0.5})$, $\exp(-\Theta(n))$, or being $\Theta(1)$ and $1 - \Theta(1)$ at the same time (denoted by $\Theta(1) \wedge (1 - \Theta(1))$). The $1 - \exp(-\Theta(n))$ case is positive news, because it states that CC is satisfied almost surely when $n$ is large, regardless of the adversary's choice. The remaining three cases are negative news, because they state that the adversary can make CC to be violated with non-negligible probability, no matter how large $n$ is.

**Second, semi-random satisfaction of** PAR **(Theorems 3, 4, 5, 6).** We prove that, under mild assumptions, for any fixed $m \geq 3$ and any sufficiently large $n$, the semi-random satisfaction of PAR under a wide range of voting rules is $1 - \Theta(n^{-0.5})$. These are positive news, because they state that PAR is satisfied almost surely for large $n$, regardless of the adversary's choice. While this message may not be surprising at a high level, as the probability for a single agent to change the winner vanishes as $n \to \infty$, the theorems are useful and non-trivial, as they provide asymptotically tight rates.

In particular, straightforward corollaries of our theorems to IC address open questions posed by Berg and Lepelley [3] in 1994, and also provides a mathematical justification of two common beliefs related to PAR: first, IC exaggerates the likelihood for paradoxes to happen, and second, the dissatisfaction of PAR is not a significant concern in practice [31], especially when it is compared to our results on semi-random CC. Table 2 summarizes corollaries of our results under some commonly-studied voting rules w.r.t. IC as well as the satisfaction of CC and PAR on Preflib data [35], where $\Theta(1) \wedge (1 - \Theta(1))$ means that there exists constants $0 < \alpha < \beta \leq 1$ that do not depend on $n$, such that the likelihood is in $[\alpha, \beta]$.

Table 2: Satisfaction of CC and PAR w.r.t. IC and w.r.t. 315 Preflib profiles of linear orders under elections category. Experimental results are presented in Appendix G.

|  | Axiom | Plu. | Borda | Veto | STV | Black | MM | Sch. | RP | Copeland$_{0.5}$ |
|---|---|---|---|---|---|---|---|---|---|---|
| **Theory** | CC | $\Theta(1) \wedge (1 - \Theta(1))$ | | | | always satisfied | | | | |
| | PAR | always satisfied | | | | $1 - \Theta\left(n^{-0.5}\right)$ | | | | |
| **Preflib** | CC | 96.8% | 92.4% | 74.2% | 99.7% | 100% | 100% | 100% | 100% | 100% |
| | PAR | 100% | 100% | 100% | 99.7% | 99.4% | 100% | 100% | 100% | 99.7% |

Table 2 provides a more quantitative way of comparing voting rules. Suppose the decision maker puts 50% weight (or any fixed non-zero ratio) on both CC and PAR, and assume that the preferences are generated from IC. Then, when $n$ is sufficiently large, the last five voting rules in the table (that satisfy CC) outperform the first five voting rules in the table (the first four satisfies PAR).

**Beyond** CC **and** PAR**.** Theorems 1–6 are proved by (non-trivial) applications of a *categorization lemma* (Lemma 1), which characterizes semi-random satisfaction of a large class of axioms that can be represented by unions of finitely many polyhedra, including CC and PAR. We believe that Lemma 1 is a promising tool for analyzing other axioms in future work.

## 1.2  Related Work and Discussions

**The Condorcet criterion** (CC) was proposed by Condorcet in 1785 [11], has been one of the most classical and well-studied axioms, and has *"nearly universal acceptance"* [44, p. 46]. CC is satisfied by many commonly-studied voting rules, except positional scoring rules [18] and multi-round-score-based elimination rules, such as STV. Most previous work focused on characterizing the *Condorcet efficiency*, which is the probability for the Condorcet winner to win conditioned on its existence [17, 16, 43, 25, 40]. Beyond positional scoring rules, the study was mostly based on computer simulations, see, e.g., [20, 21, 37, 41].

**The participation axiom** (PAR) was motivated by the *no-show paradox* [19] and was proved to be incompatible with CC for every $m \geq 4$ [39]. The likelihood of PAR under commonly studied voting rules w.r.t. IC was posed as an open question by Berg and Lepelley [3] in 1994, and has been investigated in a series of works including [32, 31, 52], see [27, Chapter 4.2.2]. In particular, Lepelley and Merlin [31] analyzed the likelihood of various no-show paradoxes for three alternatives under *scoring runoff rules*, which includes STV, w.r.t. IC and other distributions, and *"strongly believe that the no-show paradox is not an important flaw of the scoring run-off voting systems"*.

**Our work vs. previous work on** CC **and** PAR**.** Our results address open questions by Berg and Lepelley [3] about the likelihood of satisfaction of CC and PAR in two dimensions: first, we conduct semi-random analysis, which extends i.i.d. models and is believed to be significantly more general and realistic. Second, our results cover a wide range of voting rules whose likelihood of satisfaction under CC or PAR even w.r.t. IC were not mathematically characterized before, including CC under STV, and PAR under maximin, Copeland, ranked pairs, Schulze, and Black's rule. While all results in this paper assume that the number of alternatives $m$ is fixed, they are already more general than many previous work that focused on $m = 3$.

**Semi-random analysis.** There is a large body of literature on the applications of semi-random analysis, especially smoothed analysis, to computational problems [50]. Its main idea, i.e., the worst average-case analysis, has been proposed and investigated in other disciplines as well. For example, it is the central idea in frequentist statistics (as in the *frequentist expected loss* and *minimax decision rules* [4]) and is also closely related to the *min of means* criteria in decision theory [29].

Recently, Baumeister et al. [2] and Xia [54] independently proposed to conduct smoothed analysis in social choice. We adopt the framework in the latter work, though our motivation and goal are quite different. We aim at providing a comprehensive picture of semi-random satisfaction of voting axioms, while [54] focused on analyzing semi-random likelihood of Condorcet's voting paradox and an impossibility on *anonymity* and *neutrality*. On the technical level, while Lemma 1 is a straightforward corollary of [55, Theorem 2], applications of results like Lemma 1 can be highly non-trivial and problem dependent as commented in [55], which is the case of this paper. We believe that Lemma 1's main merit is conceptual, as it provides a general categorization of semi-random satisfaction of a large class of per-profile axioms beyond CC and PAR for future work.

## 2 Preliminaries

For any $q \in \mathbb{N}$, we let $[q] = \{1, \ldots, q\}$. Let $\mathcal{A} = [m]$ denote the set of $m \geq 3$ *alternatives*. Let $\mathcal{L}(\mathcal{A})$ denote the set of all linear orders over $\mathcal{A}$. Let $n \in \mathbb{N}$ denote the number of agents (voters). Each agent uses a linear order $R \in \mathcal{L}(\mathcal{A})$ to represent his or her preferences, called a *vote*, where $a \succ_R b$ means that the agent prefers alternative $a$ to alternative $b$. The vector of $n$ agents' votes, denoted by $P$, is called a *(preference) profile*, sometimes called an $n$-profile. The set of $n$-profiles for all $n \in \mathbb{N}$ is denoted by $\mathcal{L}(\mathcal{A})^* = \bigcup_{n=1}^{\infty} \mathcal{L}(\mathcal{A})^n$. A *fractional* profile is a profile $P$ coupled with a possibly non-integer and/or negative weight vector $\vec{\omega}_P = (\omega_R : R \in P) \in \mathbb{R}^n$ for the votes in $P$. It follows that a non-fractional profile is a fractional profile with uniform weight, namely $\vec{\omega}_P = \vec{1}$. Sometimes the weight vector is omitted when it is clear from the context or when $\vec{\omega}_P = \vec{1}$.

For any (fractional) profile $P$, let $\mathrm{Hist}(P) \in \mathbb{R}^{m!}_{\geq 0}$ denote the anonymized profile of $P$, also called the *histogram* of $P$, which contains the total weight of every linear order in $\mathcal{L}(\mathcal{A})$ according to $P$. An *irresolute voting rule* $\bar{r} : \mathcal{L}(\mathcal{A})^* \to (2^{\mathcal{A}} \setminus \{\emptyset\})$ maps a profile to a non-empty set of winners in $\mathcal{A}$. A *resolute* voting rule $r$ is a special irresolute voting rule that always chooses a single alternative as the (unique) winner. We say that a voting rule $r$ is a *refinement* of another voting rule $\bar{r}$, if for every profile $P$, $r(P) \subseteq \bar{r}(P)$.

**(Un)weighted majority graphs and (weak) Condorcet winners.** For any (fractional) profile $P$ and any pair of alternatives $a, b$, let $P[a \succ b]$ denote the total weight of votes in $P$ where $a$ is preferred to $b$. Let $\mathrm{WMG}(P)$ denote the *weighted majority graph* of $P$, whose vertices are $\mathcal{A}$ and whose weight on edge $a \to b$ is $w_P(a, b) = P[a \succ b] - P[b \succ a]$. Let $\mathrm{UMG}(P)$ denote the *unweighted majority graph*, which is the unweighted directed graph that is obtained from $\mathrm{WMG}(P)$ by keeping the edges with strictly positive weights. Sometimes a distribution $\pi$ over $\mathcal{L}(\mathcal{A})$ is viewed as a fractional profile, where for each $R \in \mathcal{L}(\mathcal{A})$ the weight on $R$ is $\pi(R)$. In such cases, we let $\mathrm{WMG}(\pi)$ denote the weighted majority graph of the fractional profile represented by $\pi$.

The *Condorcet winner* of a profile $P$ is the alternative that only has outgoing edges in $\mathrm{UMG}(P)$. A *weak Condorcet winner* is an alternative that does not have incoming edges in $\mathrm{UMG}(P)$. Let $\mathrm{CW}(P)$ and $\mathrm{WCW}(P)$ denote the set of Condorcet winners and weak Condorcet winners in $P$, respectively. Notice that $\mathrm{CW}(P) \subseteq \mathrm{WCW}(P)$ and $|\mathrm{CW}(P)| \leq 1$. The domain of $\mathrm{CW}(\cdot)$ and $\mathrm{WCW}(\cdot)$ can be naturally extended to all weighted or unweighted directed graphs.

For example, a distribution $\hat{\pi}$, $\mathrm{WMG}(\hat{\pi})$, and $\mathrm{UMG}(\hat{\pi})$ for $m = 3$ are illustrated in Figure 1. We have $\mathrm{CW}(\hat{\pi}) = \emptyset$ and $\mathrm{WCW}(\hat{\pi}) = \{1, 2\}$. As another example, let $\pi_{\mathrm{uni}}$ denote the uniform

distribution over $\mathcal{L}(\mathcal{A})$. Then, the weight on every edge in WMG($\pi_{\text{uni}}$) is 0 and UMG($\pi_{\text{uni}}$) does not contain any edge.

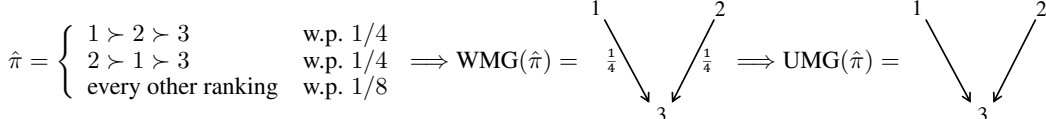

$$\hat{\pi} = \begin{cases} 1 \succ 2 \succ 3 & \text{w.p. } 1/4 \\ 2 \succ 1 \succ 3 & \text{w.p. } 1/4 \\ \text{every other ranking} & \text{w.p. } 1/8 \end{cases} \implies \text{WMG}(\hat{\pi}) = \quad \implies \text{UMG}(\hat{\pi}) = $$

Figure 1: $\hat{\pi}$, WMG($\hat{\pi}$) (only positive edges are shown), and UMG($\hat{\pi}$).

Due to the space constraint, we focus on presenting semi-random CC on positional scoring rules and multi-round score-based elimination (MRSE) rules rules in the main text, whose irresolute versions are defined below. Their resolute versions can be obtained by applying a tie-breaking mechanism on the co-winners. See Section A for definitions of other rules studied in Section 3 for PAR.

**Integer positional scoring rules.** An *(integer) positional scoring rule* $\overline{r}_{\vec{s}}$ is characterized by an integer scoring vector $\vec{s} = (s_1, \ldots, s_m) \in \mathbb{Z}^m$ with $s_1 \geq s_2 \geq \cdots \geq s_m$ and $s_1 > s_m$. For any alternative $a$ and any linear order $R \in \mathcal{L}(\mathcal{A})$, we let $\vec{s}(R, a) = s_i$, where $i$ is the rank of $a$ in $R$. Given a profile $P$ with weights $\vec{\omega}_P$, the positional scoring rule $\overline{r}_{\vec{s}}$ chooses all alternatives $a$ with maximum $\sum_{R \in P} \omega_R \cdot \vec{s}(R, a)$. For example, *plurality* uses the scoring vector $(1, 0, \ldots, 0)$, *Borda* uses the scoring vector $(m-1, m-2, \ldots, 0)$, and *veto* uses the scoring vector $(1, \ldots, 1, 0)$.

**Multi-round score-based elimination (MRSE) rules.** An irresolute MRSE rule $\overline{r}$ for $m$ alternatives is defined by a vector of $m-1$ rules $(\overline{r}_2, \ldots, \overline{r}_m)$, where for every $2 \leq i \leq m$, $\overline{r}_i$ is a positional scoring rule over $i$ alternatives that outputs a *total preorder* over them in the decreasing order of their scores. Given a profile $P$, $\overline{r}(P)$ is selected in $m-1$ rounds. For each $1 \leq i \leq m-1$, in round $i$, a loser (an alternative with the lowest score) under $\overline{r}_{m+1-i}$ is eliminated. We use the *parallel-universes tie-breaking (PUT)* [12] to select winners—an alternative $a$ is a winner if there is a way to break ties among the losers in each round, so that $a$ is the remaining alternative after $m-1$ rounds. If an MRSE rule $\overline{r}$ only uses integer position scoring rules, then it is called an *int-MRSE rule*. Commonly studied int-MRSE rules include *STV*, which uses plurality in each round, *Coombs*, which uses veto in each round, and *Baldwin's rule*, which uses Borda in each round.

**Example 2 (Irresolute STV).** *Figure 2 illustrates the execution of irresolute STV,*

*denoted by $\overline{STV}$, under $\pi_{uni}$ (the uniform distribution) and $\hat{\pi}$ (the distribution in Figure 1), where each node represents the (tied) losers of the corresponding round, and each edge represents the loser to be eliminated. We have $\overline{STV}(\pi_{uni}) = \{1, 2, 3\}$ and $\overline{STV}(\hat{\pi}) = \{1, 2\}$.*

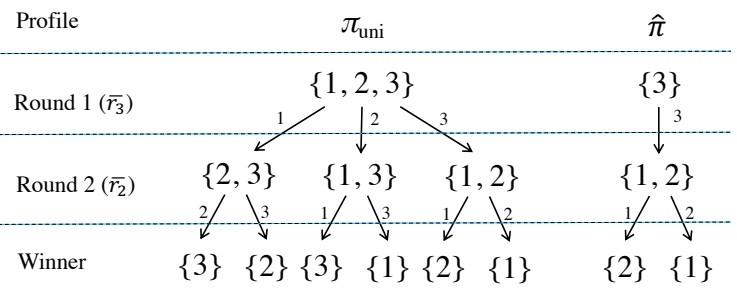

| Profile | $\pi_{\text{uni}}$ | $\hat{\pi}$ |
|---|---|---|
| Round 1 ($\overline{r}_3$) | $\{1, 2, 3\}$ | $\{3\}$ |
| Round 2 ($\overline{r}_2$) | $\{2, 3\}$ $\quad$ $\{1, 3\}$ $\quad$ $\{1, 2\}$ | $\{1, 2\}$ |
| Winner | $\{3\}$ $\{2\}$ $\{3\}$ $\{1\}$ $\{2\}$ $\{1\}$ | $\{2\}$ $\{1\}$ |

Figure 2: $\overline{STV}$ under $\pi_{\text{uni}}$ and $\hat{\pi}$ (defined in Figure 1).

**Axioms of voting.** We focus on *per-profile axioms* [54] in this paper. A per-profile axiom is defined as a function $X$ that maps a voting rule $\overline{r}$ and a profile $P$ to $\{0, 1\}$, where 0 (respectively 1) means that $\overline{r}$ dissatisfies/violates (respectively, satisfies) the axiom at $P$. Then, the classical (worst-case) satisfaction of the axiom under $\overline{r}$ is defined to be $\min_{P \in \mathcal{L}(\mathcal{A})^*} X(\overline{r}, P)$.

For example, a (resolute or irresolute) rule $\overline{r}$ satisfies CC, if $\min_{P \in \mathcal{L}(\mathcal{A})^*} \text{CC}(\overline{r}, P) = 1$, where $\text{CC}(\overline{r}, P) = 1$ if and only if either (1) there is no Condorcet winner under $P$, or (2) the Condorcet winner is a co-winner of $P$ under $\overline{r}$. A resolute rule $r$ satisfies PAR, if $\min_{P \in \mathcal{L}(\mathcal{A})^*} \text{PAR}(r, P) = 1$, where $\text{PAR}(r, P) = 1$ if and only if no voter has incentive to abstain from voting. Formally, let $P = (R_1, \ldots, R_n)$, then $[\text{PAR}(r, P) = 1] \iff [\forall j \leq n, r(P) \succeq_{R_j} r(P - R_j)]$, where $P - R_j$ is the $(n-1)$-profile that is obtained from $P$ by removing the $j$-th vote. For any pair of alternatives $a$

and $b$, we write $\{a\} \succeq_{R_j} \{b\}$ if and only if agent $j$, whose preferences are $R_j$, prefers $a$ to $b$. See Appendix B for a list of 13 well-studied per-profile axioms and one non-per-profile axiom.

**Semi-random satisfaction of axioms.** Given a per-profile axiom $X$, a set $\Pi$ of distributions over $\mathcal{L}(\mathcal{A})$, a voting rule $\overline{r}$, and $n \in \mathbb{N}$, the *semi-random satisfaction of $X$* under $\overline{r}$ with $n$ agents, denoted by $\widetilde{X}^{\min}_{\Pi}(\overline{r}, n)$, is defined in Equation (1) in the Introduction. We note that the "min" in the superscript means that the adversary aims at minimizing the satisfaction of $X$. Formally, $\Pi$ is part of the single-agent preference model defined as follows.

**Definition 1** (**Single-Agent Preference Model [54]**). *A single-agent preference model is denoted by $\mathcal{M} = (\Theta, \mathcal{L}(\mathcal{A}), \Pi)$, where $\Theta$ is the parameter space, $\mathcal{L}(\mathcal{A})$ is the sample space, and $\Pi$ consists of distributions indexed by $\Theta$. $\mathcal{M}$ is* strictly positive *if there exists $\epsilon > 0$ such that the probability of any linear order under any distribution in $\Pi$ is at least $\epsilon$. $\mathcal{M}$ is* closed *if $\Pi$ (which is a subset of the probability simplex in $\mathbb{R}^{m!}$) is a closed set in $\mathbb{R}^{m!}$.*

Example 1 illustrates a strictly positive and closed single-agent preference model for $m = 3$, where $\Pi = \{\pi^1, \pi^2\}$ and $\epsilon = 1/8$. Other examples can be found in [54, Example 2 in the appendix].

## 3 The Semi-random Satisfaction of CC and PAR

**Semi-random CC under Integer Positional Scoring Rules.** To present the results, we first define *almost Condorcet winners (ACW)* of a profile $P$, which are the two alternatives (whenever they exist) that are tied in the UMG and beat all other alternatives in head-to-head competitions.

**Definition 2** (**Almost Condorcet Winners**). *For any unweighted directed graph $G$ over $\mathcal{A}$, a pair of alternatives $a, b$ are* almost Condorcet winners (ACWs)*, denoted by $ACW(G)$, if (1) $a$ and $b$ are tied in $G$, and (2) for any other alternative $c \notin \{a, b\}$, $G$ has $a \to c$ and $b \to c$. For any profile $P$, let $ACW(P) = ACW(UMG(P))$.*

For example, 1 and 2 are ACWs of $\hat{\pi}$ (as a fractional profile) in Figure 1. By definition, for any profile $P$, $|ACW(P)|$ is either 0 or 2, and when it is 2, $WCW(P) = ACW(P)$.

We now present a full characterization of semi-random CC under integer positional scoring rules. Let $CH(\Pi)$ denote the *convex hull* of $\Pi$.

**Theorem 1** (**Semi-random CC: Integer Positional Scoring Rules**). *For any fixed $m \geq 3$, let $\mathcal{M} = (\Theta, \mathcal{L}(\mathcal{A}), \Pi)$ be a strictly positive and closed single-agent preference model, let $\overline{r}_{\vec{s}}$ be an irresolute integer positional scoring rule, and let $r_{\vec{s}}$ be a refinement of $\overline{r}_{\vec{s}}$. For any $n \geq 8m + 49$ with $2 \mid n$,*

$$\widetilde{CC}^{\min}_{\Pi}(r_{\vec{s}}, n) = \begin{cases} 1 - \exp(-\Theta(n)) & \text{if } \forall \pi \in CH(\Pi), |WCW(\pi)| \times |\overline{r}_{\vec{s}}(\pi) \cup WCW(\pi)| \leq 1 \\ \Theta(n^{-0.5}) & \text{if } \begin{cases} (1) \ \forall \pi \in CH(\Pi), CW(\pi) \cap (\mathcal{A} \setminus \overline{r}_{\vec{s}}(\pi)) = \emptyset \text{ and} \\ (2) \ \exists \pi \in CH(\Pi) \text{ s.t. } |ACW(\pi) \cap (\mathcal{A} \setminus \overline{r}_{\vec{s}}(\pi))| = 2 \end{cases} \\ \exp(-\Theta(n)) & \text{if } \exists \pi \in CH(\Pi) \text{ s.t. } CW(\pi) \cap (\mathcal{A} \setminus \overline{r}_{\vec{s}}(\pi)) \neq \emptyset \\ \Theta(1) \wedge (1 - \Theta(1)) & \text{otherwise} \end{cases}$$

*For any $n \geq 8m + 49$ with $2 \nmid n$,*

$$\widetilde{CC}^{\min}_{\Pi}(r_{\vec{s}}, n) = \begin{cases} 1 - \exp(-\Theta(n)) & \text{same as the } 2 \mid n \text{ case} \\ \exp(-\Theta(n)) & \text{if } \exists \pi \in CH(\Pi) \text{ s.t. } \begin{cases} (1) \ CW(\pi) \cap (\mathcal{A} \setminus \overline{r}_{\vec{s}}(\pi)) \neq \emptyset \text{ or} \\ (2) \ |ACW(\pi) \cap (\mathcal{A} \setminus \overline{r}_{\vec{s}}(\pi))| = 2 \end{cases} \\ \Theta(1) \wedge (1 - \Theta(1)) & \text{otherwise} \end{cases}$$

**Generality.** We believe that Theorem 1 is quite general, as it can be applied to *any* refinement of *any* irresolute integer positional scoring rule (i.e., using *any* tie-breaking mechanism) w.r.t. *any* $\Pi$ that satisfies mild conditions. The power of Theorem 1 is that it converts complicated probabilistic arguments about semi-random CC to deterministic arguments about properties of (fractional) profiles in $CH(\Pi)$, i.e., $\overline{r}_{\vec{s}}(\pi)$, $CW(\pi)$, $ACW(\pi)$, and $WCW(\pi)$, which are much easier to check. In particular, Theorem 1 can be easily applied to i.i.d. distributions (including IC) as shown in Example 3 below.

**Intuitive explanations of the conditions.** While the conditions for the cases in Theorem 1 may appear technical, they have intuitive explanations. Take the $2 \mid n$ case for example. **The 1 —**

$\exp(-\Theta(n))$ **case** happens if every $\pi \in CH(\Pi)$ is a "robust" instance of CC satisfaction, in the sense that after any infinitely small perturbation $\vec{\delta} \in \mathbb{R}^{m!}$ is introduced to $\pi$, $\vec{\pi} + \vec{\delta}$ is still an instance of CC satisfaction. For **the $\Theta(n^{-0.5})$ case,** condition (1) states that every $\pi \in CH(\Pi)$ is an instance of CC satisfaction, and condition (2) requires that some $\pi \in CH(\Pi)$ corresponds to a "non-robust" instance of CC satisfaction, in the sense that after a small perturbation $\vec{\eta}$ is added to $\pi$, CC is violated at $\pi + \vec{\eta}$. **The $\exp(-\Theta(n))$ case** happens if there exists a "robust" instance of CC dissatisfaction $\pi \in CH(\Pi)$, in the sense that after any small perturbation is introduced to $\pi$, it is still an instance of CC dissatisfaction. Otherwise, **the $\Theta(1) \wedge (1 - \Theta(1))$ case** holds.

**Odd vs. even $n$.** The $2 \nmid n$ case has similar explanations. The main difference is that when $2 \nmid n$, the UMG of any $n$-profile must be a complete graph. Therefore, when $ACW(\pi) \neq \emptyset$, with high probability an alternative in $ACW(\pi)$ is the Condorcet winner in the randomly-generated $n$-profile. Then, the $\Theta(n^{-0.5})$ case in $2 \mid n$ becomes part of the $\exp(-\Theta(n))$ case in $2 \nmid n$.

**Example 3** (**Applications of Theorem 1 to plurality**). *In the setting of Example 1, we apply Theorem 1 to any sufficiently large $n$ with $2 \mid n$ and any refinement of irresolute plurality, denoted by Plu, for the following sets of distributions.*

• $\Pi = \{\pi^1, \pi^2\}$. *We have* $\widetilde{CC}_{\Pi}^{\min}(Plu, n) = \exp(-\Theta(n))$, *because let* $\pi' = \frac{3\pi^1 + \pi^2}{4}$, *we have* $CW(\pi') = WCW(\pi') = \{2\}$, $ACW(\pi') = \emptyset$, *and* $\overline{Plu}(\pi') = \{1\}$.

• $\Pi_{IC} = \{\pi_{uni}\}$, *i.e., semi-random CC becomes likelihood of CC w.r.t. IC. We have* $\widetilde{CC}_{\Pi_{IC}}^{\min}(Plu, n) = \Theta(1) \wedge (1 - \Theta(1))$, *because* $CW(\pi_{uni}) = \emptyset$, $WCW(\pi_{uni}) = \{1, 2, 3\}$, *and* $ACW(\pi_{uni}) = \emptyset$.

**Semi-random CC under int-MRSE Rules.** Semi-random CC under an MRSE rule $\overline{r}$ depends on whether the positional scoring rules it uses satisfy the CONDORCET LOSER (CL) criterion, which requires that the Condorcet loser, whenever it exists, never wins. The Condorcet loser is the alternative that loses to all head-to-head competitions. For any voting rule $\overline{r}$, we write $CL(\overline{r}) = 1$ if and only if $\overline{r}$ satisfies CONDORCET LOSER.

To present the result, we first define *parallel universes* under an MRSE rule $\overline{r}$ at $\vec{x} \in \mathbb{R}^{m!}$, denoted by $PU_{\overline{r}}(\vec{x})$, to be the set of all elimination orders in the execution of $\overline{r}$ at $\vec{x}$. Then, for any alternative $a$, let the *possible losing rounds*, denoted by $LR_{\overline{r}}(\vec{x}, a) \subseteq [m-1]$, be the set of all rounds in the parallel universes where $a$ drops out. The formal definitions and an example can be found in Definition 26 and Example 12 in Appendix E.3, respectively.

We are now ready to present the $2 \mid n$ case of our characterization of semi-random CC under MRSE rules. The full version can be found in Appendix E.3.

**Theorem 2** (**Semi-random CC: int-MRSE rules, $2 \mid n$**). *For any fixed $m \geq 3$, let $\mathcal{M} = (\Theta, \mathcal{L}(\mathcal{A}), \Pi)$ be a strictly positive and closed single-agent preference model, let $\overline{r} = (\overline{r}_2, \ldots, \overline{r}_m)$ be an int-MRSE rule and let $r$ be a refinement of $\overline{r}$. For any $n \in \mathbb{N}$ with $2 \mid n$, we have*

$$\widetilde{CC}_{\Pi}^{\min}(r, n) = \begin{cases} 1 & \text{if } \forall 2 \leq i \leq m, CL(\overline{r}_i) = 1 \\ 1 - \exp(-\Theta(n)) & \text{if } \begin{cases} \text{(1) } \exists 2 \leq i \leq m \text{ s.t. } CL(\overline{r}_i) = 0 \text{ and} \\ \text{(2) } \forall \pi \in CH(\Pi), \forall a \in WCW(\pi), \forall i^* \in LR_{\overline{r}}(\pi, a), \\ \qquad \text{we have } CL(\overline{r}_{m+1-i^*}) = 1 \end{cases} \\ \Theta(n^{-0.5}) & \text{if } \begin{cases} \text{(1) } \forall \pi \in CH(\Pi), CW(\pi) \cap (\mathcal{A} \setminus \overline{r}(\pi)) = \emptyset \text{ and} \\ \text{(2) } \exists \pi \in CH(\Pi) \text{ s.t. } |ACW(\pi) \cap (\mathcal{A} \setminus \overline{r}(\pi))| = 2 \end{cases} \\ \exp(-\Theta(n)) & \text{if } \exists \pi \in CH(\Pi) \text{ s.t. } CW(\pi) \cap (\mathcal{A} \setminus \overline{r}(\pi)) \neq \emptyset \\ \Theta(1) \wedge (1 - \Theta(1)) & \text{otherwise} \end{cases}$$

The most interesting cases are the 1 case and the $1 - \exp(-\Theta(n))$ case. The 1 case happens when all positional scoring rules used in $\overline{r}$ satisfy CONDORCET LOSER. In this case, if the Condorcet winner exists, then it cannot be a loser in any round, which means that it is the unique winner under $\overline{r}$. The $1 - \exp(-\Theta(n))$ case happens when (1) the 1 case does not happen, and (2) for every distribution $\pi \in CH(\Pi)$, every weak Condorcet winner $a$, and every possible losing round $i^*$ for $a$, the positional scoring rule used in round $i^*$, i.e. $\overline{r}_{m+1-i^*}$, must satisfy CONDORCET LOSER. (2) guarantees that when a small permutation is added to $\pi$, if a weak Condorcet winner $a$ becomes the Condorcet winner, then it will be the unique winner under $\overline{r}$. See Example 13 in Appendix E.3 for an application of Theorem 2 to STV.

Like Theorem 1, Theorem 2 can also be easily applied to i.i.d. distributions, which corresponds to $\Pi_{\text{IC}} = \{\pi_{\text{uni}}\}$.

**Corrollary 1** (**Likelihood of** CC **under int-MRSE rules w.r.t. IC**). *For any fixed $m \geq 3$, any refinement $r$ of any int-MRSE rule $\overline{r}$, and any $n \in \mathbb{N}$,*

$$\Pr_{P \sim (\pi_{\text{uni}})^n}(\text{CC}(r, P) = 1) = \begin{cases} 1 & \text{if } \forall 2 \leq i \leq m, CL(\overline{r}_i) = 1 \\ \Theta(1) \wedge (1 - \Theta(1)) & \text{otherwise} \end{cases}$$

**Proof sketches for Theorem 1 and 2.** In light of various multivariate central limit theorems (CLTs), when $n$ is large, the profile is approximately $n \cdot \pi^*$ for $\pi^* = (\sum_{j=1}^n \pi_j)/n \in \text{CH}(\Pi)$ with high probability. Despite this high-level intuition, the conditions of the cases are quite differently from semi-random CC by definition. To see this, note that (i) the adversary may not be able to set any agent's ground truth preferences to be $\pi^* \in \text{CH}(\Pi)$, because $\pi^*$ may not be in $\Pi$ as shown in Example 3, and (ii) in the definition of semi-random CC, agent $j$'s vote is a random variable distributed as $\pi_j$, instead of the fractional vote $\pi_j$. Standard CLTs can probably be applied to prove the $1 - \exp(-\Theta(n))$ case and the $\Theta(1) \wedge (1 - \Theta(1))$ case, but they are too coarse for other cases.

To address this challenge, we model the satisfaction of CC by the union of multiple polyhedra $\mathcal{C}$ as exemplified in Section 4. This converts the semi-random CC problem to a *PMV-in-$\mathcal{C}$* problem [55] (Definition 3 below). Then, we refine [55, Theorem 2] to prove a categorization lemma (Lemma 1), and apply it to obtain Lemma 2 that characterizes semi-random CC for a large class of voting rules called *generalized irresolute scoring rules (GISRs)* [22, 53] (Definition 7 in Appendix D.1). Finally, we apply Lemma 2 to integer positional scoring rules and int-MRSE rules to obtain Theorem 1 and Theorem 2. The full proof can be found in Appendix E.2 and E.3, respectively. □

**The semi-random satisfaction of** PAR. Due to the space constraint, we briefly introduce our characterizations of semi-random PAR under commonly-studied voting rules defined in Appendix A, which belong to a large class of voting rules called *generalized scoring rules (GSRs)* [56] (Definition 7 in Appendix D.1). Formal statements and proofs of the theorems can be found in Appendix F.2–F.5.

**Theorems 3, 4, 5, 6 (Semi-random PAR under commonly-studied rules).** *For any fixed $m \geq 4$, any GSR $r$ that is a refinement of maximin, STV, Schulze, ranked pairs, Copeland, any int-MRSE, or any Condocetified positional scoring rule (such as Black's rule), and any strictly positive and closed $\Pi$ over $\mathcal{L}(\mathcal{A})$ with $\pi_{\text{uni}} \in CH(\Pi)$, there exists $N \in \mathbb{N}$ such that for every $n \geq N$, $\widetilde{\text{PAR}}_\Pi^{\min}(r, n) = 1 - \Theta(\frac{1}{\sqrt{n}})$.*

In fact, if $\pi_{\text{uni}} \notin \text{CH}(\Pi)$, then semi-random PAR converges to 1 at a faster rate, which is more positive news, as shown in Lemma 3 (Appendix F.1).

## 4  Beyond CC and PAR: The Categorization Lemma

In this section, we present a general lemma that characterizes semi-random satisfaction of per-profile axioms that can be represented by unions of polyhedra, including CC and PAR. To develop intuition, we start with an example of modeling CC under irresolute plurality as the union of the following two types of polyhedra in $\mathbb{R}^{m!}$.

• $\mathcal{C}_{\text{NCW}}$ represents that there is no Condorcet winner, which is the union of polyhedra $\mathcal{H}^G$, where $G$ is an unweighted graph over $\mathcal{A}$ that does not have a Condorcet winner, as exemplified in Example 4.

• $\mathcal{C}_{\text{CWW}}$ represents that the Condorcet winner exists and also wins the plurality election, which is the union of polyhedra $\mathcal{H}^a$ for every $a \in \mathcal{A}$, that represents $a$ being the Condorcet winner as well as a $\overline{\text{Plu}}$ co-winner, as exemplified in Example 5.

**Example 4** ($\mathcal{H}^G$). *Let $m = 3$ and let $x_{abc}$ denote the number of $[a \succ b \succ c]$ votes in a profile. The following figure shows $G$ (left) and $\mathcal{H}^G$ (right).*

$$(x_{213} + x_{231} + x_{321}) - (x_{123} + x_{132} + x_{312}) \leq -1 \quad (2)$$

$$(x_{123} + x_{132} + x_{213}) - (x_{231} + x_{321} + x_{312}) \leq -1 \quad (3)$$

$$(x_{132} + x_{312} + x_{321}) - (x_{123} + x_{213} + x_{231}) \leq 0 \quad (4)$$

$$(x_{123} + x_{213} + x_{231}) - (x_{132} + x_{312} + x_{321}) \leq 0 \quad (5)$$

*Among the four inequalities, (2) represents the $1 \to 2$ edge in $G$, (3) represents the $3 \to 1$ edge in $G$, and (4) and (5) represent the tie between $2$ and $3$ in $G$.*

**Example 5 ($\mathcal{H}^a$).** *Let $m = 3$. $\mathcal{H}^1$ is the polyhedron represented by the following four inequalities:*

$$\left.\begin{array}{c} (x_{213} + x_{231} + x_{321}) - (x_{123} + x_{132} + x_{312}) \leq -1 \\ (x_{231} + x_{321} + x_{312}) - (x_{123} + x_{132} + x_{213}) \leq -1 \end{array}\right\} \text{1 is the Condorcet winner}$$

$$\left.\begin{array}{c} (x_{213} + x_{231}) - (x_{123} + x_{132}) \leq 0 \\ (x_{321} + x_{312}) - (x_{123} + x_{132}) \leq 0 \end{array}\right\} \text{1 is a } \overline{Plu} \text{ co-winner}$$

It is not hard to see that $\overline{\text{Plu}}$ satisfies CC at a profile $P$ if and only if $\text{Hist}(P)$ is in $\mathcal{C} = \mathcal{C}_{\text{NCW}} \cup \mathcal{C}_{\text{CWW}}$, where $\mathcal{C}_{\text{NCW}} = \bigcup_{G:\text{CW}(G)=\emptyset} \mathcal{H}^G$ and $\mathcal{C}_{\text{CWW}} = \bigcup_{a \in \mathcal{A}} \mathcal{H}^a$. An example of PAR under Copeland can be found in Appendix C.1. In general, the satisfaction of a wide range of axioms can be represented by unions of finitely many polyhedra. Then, the semi-random satisfaction problem reduces to the lower bound of the following PMV-in-$\mathcal{C}$ problem.

**Definition 3 (The PMV-in-$\mathcal{C}$ problem [55]).** *Given $q, I \in \mathbb{N}$, $\mathcal{C} = \bigcup_{i \leq I} \mathcal{H}_i$, where $\forall i \leq I$, $\mathcal{H}_i \subseteq \mathbb{R}^q$ is a polyhedron, and a set $\Pi$ of distributions over $[q]$, we are interested in*

**the upper bound** $\sup_{\vec{\pi} \in \Pi^n} \Pr(\vec{X}_{\vec{\pi}} \in \mathcal{C})$, *and* **the lower bound** $\inf_{\vec{\pi} \in \Pi^n} \Pr(\vec{X}_{\vec{\pi}} \in \mathcal{C})$,

*where $\vec{X}_{\vec{\pi}}$ is the $(n, q)$-Poisson multinomial variable (PMV) that corresponds to the histogram of $n$ independent random variables distributed as $\vec{\pi}$.*

See Example 7 in Appendix C.2 for an example of PMV. The following lemma provides an asymptotic characterization on the lower bound of the PMV-in-$\mathcal{C}$ problem.

**Lemma 1 (Categorization lemma, simplified).** *For any PMV-in-$\mathcal{C}$ problem and any $n \in \mathbb{N}$, $\inf_{\vec{\pi} \in \Pi^n} \Pr(\vec{X}_{\vec{\pi}} \in \mathcal{C})$ is $0$, $\exp(-\Theta(n))$, $poly^{-1}(n)$, $\Theta(1) \wedge (1 - \Theta(1))$, $1 - poly^{-1}(n)$, $1 - \exp(-\Theta(n))$, or $1$.*

The full version of Lemma 1 (Appendix C.2) also characterizes the condition for each case, the degree of polynomial, and $\sup_{\vec{\pi} \in \Pi^n} \Pr(\vec{X}_{\vec{\pi}} \in \mathcal{C})$. Lemma 1's main merit is conceptual, as it categorizes the semi-random likelihood into seven cases for quantitative comparisons, summarized in the increasing order in the table below, which are *0*, *very unlikely (VU)*, *unlikely (U)*, *medium (M)*, *likely (L)*, *very likely (VL)*, and *1*. The first three cases (0, VU, U) are negative news, where the adversary can set the ground truth so that the axiom is almost surely violated in large elections ($n \to \infty$). The last three cases (L, VL, and 1) are positive news, because the axiom is satisfied almost surely in large elections, regardless of the adversary's choice. The M case can be interpreted positively or negatively, depending on the context.

| Name | 0 | VU | U | M | L | VL | 1 |
|---|---|---|---|---|---|---|---|
| Lem. 1 | 0 | $\exp(-\Theta(n))$ | $poly^{-1}(n)$ | $\Theta(1) \wedge (1 - \Theta(1))$ | $1 - poly^{-1}(n)$ | $1 - \exp(-\Theta(n))$ | 1 |

# 5   Future work

There are many open questions for future work. While we believe that the assumptions (of strict positiveness and closedness) on $\Pi$ are mild, it is an important and interesting open question to extend the analysis to variable $\epsilon$'s, and perhaps establish bounds that explicitly depend on $\epsilon$. What are the semi-random CC and semi-random PAR for voting rules not sdudied in this paper, such as Bucklin? What is the semi-random satisfaction of PAR when a group of agents can simultaneously abstain from voting [31]? More generally, we believe that drawing a comprehensive picture of semi-random satisfactions of other voting axioms and/or paradoxes, such as those described in Appendix B, is an important, promising, and challenging mission, and the categorization lemma (Lemma 1) can be a useful conceptual and technical tool to start with.

**Acknowledgments**

We thank attendees at COMSOC-21 and anonymous reviewers for helpful comments. This work is supported by NSF #1453542, ONR #N00014-17-1-2621, and a gift fund from Google.

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
