# Contents

# A  Definitions of More Voting Rules

**WMG-based rules.** A voting rule is said to be *weighted-majority-graph-based (WMG-based)* if its winners only depend on the WMG of the input profile. In this paper we consider the following commonly-studied WMG-based irresolute rules.

- **Copeland.** The Copeland rule is parameterized by a number $0 \leq \alpha \leq 1$, and is therefore denoted by $\overline{\mathrm{Cd}}_\alpha$. For any profile $P$, an alternative $a$ gets $1$ point for each other alternative it beats in head-to-head competitions, and gets $\alpha$ points for each tie. $\overline{\mathrm{Cd}}_\alpha$ chooses all alternatives with the highest total score as winners.

- **Maximin.** For each alternative $a$, its *min-score* is defined to be $\mathrm{MS}_P(a) = \min_{b \in \mathcal{A}} w_P(a, b)$. Maximin, denoted by $\overline{\mathrm{MM}}$, chooses all alternatives with the max min-score as winners.

- **Ranked pairs.** Given a profile $P$, an alternative $a$ is a winner under ranked pairs (denoted by $\overline{\mathrm{RP}}$) if there exists a way to fix edges in $\mathrm{WMG}(P)$ one by one in a non-increasing order w.r.t. their weights (and sometimes break ties), unless it creates a cycle with previously fixed edges, so that after all edges are considered, $a$ has no incoming edge. This is known as the *parallel-universes tie-breaking (PUT)* [12].

- **Schulze.** The *strength* of any directed path in the WMG is defined to be the minimum weight on single edges along the path. For any pair of alternatives $a, b$, let $s[a, b]$ denote the highest weight among all paths from $a$ to $b$. Then, we write $a \succeq b$ if and only if $s[a, b] \geq s[b, a]$, and Schulze [47] proved that the strict version of this binary relation, denoted by $\succ$, is transitive. The Schulze rule, denoted by $\overline{\mathrm{Sch}}$, chooses all alternatives $a$ such that for all other alternatives $b$, we have $a \succeq b$.

**Condorcetified (integer) positional scoring rules.** The rule is defined by an integer scoring vector $\vec{s} \in \mathbb{Z}^m$ and is denoted by $\overline{\mathrm{Cond}_{\vec{s}}}$, which selects the Condorcet winner when it exits, and otherwise uses $\overline{r}_{\vec{s}}$ to select the (co)-winners. For example, *Black's rule* [6] is the Condorcetified Borda rule.

# B  Per-Profile and Non-Per-Profile Axioms

In this section, we provide an (incomplete) list of $14$ commonly-studied per-profile axioms and one commonly-studied non-per-profile axiom that we do not see a clear per-profile representation.

**Per-Profile Axioms.** We present the definitions of the per-profile axioms in the alphabetical order. Their equivalent $X$ definition is often straightforward unless explicitly discussed below.

1. ANONYMITY states that the winner is insensitive to the identities of the voters. It is a per-profile axiom as shown in [54].

2. CONDORCET CRITERION is a per-profile axiom as discussed in the Introduction.

3. CONDORCET LOSER requires that a *Condorcet loser*, which is the alternative who *loses* to every head-to-head competition with other alternatives, should not be selected as the winner. It is a per-profile axiom in the same sense as CC.

4. CONSISTENCY requires that for any profile $P$ and any sub-profile $P'$ of $P$, if $r(P') = r(P \setminus P')$, then $r(P) = r(P')$. Therefore, for any profile $P$, we can define

$$[Consistency(r, P) = 1] \iff [\forall P' \subset P, [r(P') = r(P \setminus P')] \Rightarrow [r(P) = r(P')]]$$

5. GROUP-NON-MANIPULABLE is defined similarly to NON-MANIPULABLE below, except that multiple voters are allowed to simultaneously change their votes, and after doing so, at least one of them strictly prefers the old winner.

6. INDEPENDENT OF CLONES requires that the winner does not change when *clones* of an alternative is introduced. The clones and the original alternative must be ranked consecutively in each vote. Let $IoC$ denote INDEPENDENT OF CLONES. For any profile $P$, we let $IoC(r, P) = 1$ if and only if for every alternative $a$ and every profile $P'$ obtain from $P$ by introducing clones of $a$, we have $r(P) = r(P')$.

7. MAJORITY CRITERION requires that any alternative that is ranked at the top place in more than $50\%$ of the votes must be selected as the winner. *Majority criterion* is stronger than CONDORCET CRITERION.

8. MAJORITY LOSER requires that any alternative who is ranked at the bottom place in more than $50\%$ of the votes should not be selected as the winner. MAJORITY LOSER is weaker than CONDORCET LOSER.

9. MONOTONICITY requires raising up the position of the current winner in any vote will not cause it to lose. Let MONO denote MONOTONICITY. One way to define $Mono$ is the following.Let $Mono^1(r, P) = 1$ if and only if for every profile $P'$ that is obtained from $P$ by raising the position of $r(P)$ in one vote, we have $r(P') = r(P)$. Another definition is: $Mono^2(r, P) = 1$ if and only if for every profile $P'$ that is obtained from $P$ by raising the position of $r(P)$ in arbitrarily many votes, we have $r(P') = r(P)$. Notice that the classical (worst-case) MONOTONICITY is satisfied if and only if $\min_P Mono^1(r, P) = 1$ or equivalently, $\min_P Mono^2(r, P) = 1$. The semi-random satisfaction of $\min_P Mono^1$ might be different from $\min_P Mono^2$, which is beyond the scope of this paper.

10. NEUTRALITY states that the winner is insensitive to the identities of the alternatives. It is a per-profile axiom as shown in [54].

11. NON-MANIPULABLE requires that no agent has incentive to unilaterally change his/her vote to improve the winner w.r.t. his/her true preferences. More precisely, for any profile $P = (R_1, \ldots, R_n)$, we have

$$[Non - Manipulable(r, P) = 1] \Leftrightarrow \left[\forall j \le n, \forall R'_j \in \mathcal{L}(\mathcal{A}), r(P) \succeq_{R_j} r(P \cup \{R'_j\} \setminus \{R_j\})\right]$$

12. PARTICIPATION is a per-profile axiom as discussed in the Introduction.

13. REVERSAL SYMMETRY requires that the winner of any profile should not be the winner when all voters' rankings are inverted.

**Non-Per-Profile Axiom(s).** We were not able to model NON-DICTATORSHIP (ND) as a per-profile axiom studied in this paper. A voting rule is not a dictator if for each $j \le n$, there exists a profile $P$ whose winner is not ranked at the top of agent $j$'s preferences.

## C Materials for Section 4: The Categorization Lemma

While the categorization lemma (Lemma 1) was presented after Theorems 1 through 6 in the main text, the proofs of the theorems depend on the lemma. Therefore, we present materials for the categorization letter before the proofs for the theorems in the appendix.

### C.1 Modeling Satisfaction of PAR as A Union of Polyhedra

PAR **under Copeland$_\alpha$.** We now show how to approximately model the satisfaction of PAR under Copeland$_\alpha$. For every pair of unweighted directed graphs $G_1, G_2$ over $\mathcal{A}$ and every $R \in \mathcal{L}(\mathcal{A})$, we define a polyhedron $\mathcal{H}^{G_1, R, G_2}$ to represent the histograms of profile $P$ that contains an $R$-vote, $G_1 = \text{UMG}(P)$, and $G_2 = \text{UMG}(P \setminus \{R\})$. The linear inequalities used to specify the UMGs of $P$ and $(P \setminus \{R\})$ are similar to $\mathcal{H}^G$ defined above, as illustrated in the following example.

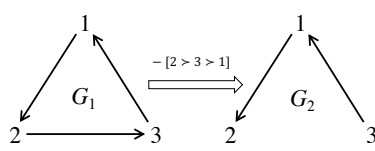

Figure 3: $G_1, G_2$, and $R$.

**Example 6.** *Let $m = 3$, $R = [2 \succ 3 \succ 1]$, and let $G_1, G_2$ denote the graphs in Figure 3. $\mathcal{H}^{G_1, R, G_2}$ is represented by the following inequalities.*

$$-x_{231} \leq -1 \tag{6}$$

$$\left.\begin{array}{l}(x_{213} + x_{231} + x_{321}) - (x_{123} + x_{132} + x_{312}) \leq -1 \\ (x_{123} + x_{132} + x_{213}) - (x_{231} + x_{321} + x_{312}) \leq -1 \\ (x_{132} + x_{312} + x_{321}) - (x_{123} + x_{213} + x_{231}) \leq -1\end{array}\right\} \tag{7}$$

$$\left.\begin{array}{l}(x_{213} + x_{231} - 1 + x_{321}) - (x_{123} + x_{132} + x_{312}) \leq -1 \\ (x_{123} + x_{132} + x_{213}) - (x_{231} - 1 + x_{321} + x_{312}) \leq -1 \\ (x_{132} + x_{312} + x_{321}) - (x_{123} + x_{213} + x_{231} - 1) \leq 0 \\ (x_{123} + x_{213} + x_{231} - 1) - (x_{132} + x_{312} + x_{321}) \leq 0\end{array}\right\} \tag{8}$$

*(6) guarantees that $P$ contains an R-vote. The three inequalities in (7) represent UMG($P$) = $G_1$, and the four inequalities in (8) represent UMG($P$) = $G_2$.*

We do not require $x_R$'s to be non-negative, which does not affect the results of the paper, because the histograms of randomly-generated profiles are always non-negative.

By enumerating $G_1$, $R$, and $G_2$ that correspond to a violation of PAR, the polyhedra that represent satisfaction of PAR under Copeland$_\alpha$ are:

$$\mathcal{C} = \bigcup\nolimits_{G_1, R, G_2 : \text{Copeland}_\alpha(G_1) \succeq_R \text{Copeland}_\alpha(G_2)} \mathcal{H}^{G_1, R, G_2}$$

### C.2 Formal Statement of the Categorization Lemma and Proof

We first introduce notation for polyhedra. Given $q \in \mathbb{N}, L \in \mathbb{N}$, an $L \times q$ integer matrix $\mathbf{A}$, a $q$-dimensional row vector $\vec{b}$, we define

$$\mathcal{H} \triangleq \left\{ \vec{x} \in \mathbb{R}^q : \mathbf{A} \cdot (\vec{x})^\top \leq \left(\vec{b}\right)^\top \right\}, \quad \mathcal{H}_{\leq 0} \triangleq \left\{ \vec{x} \in \mathbb{R}^q : \mathbf{A} \cdot (\vec{x})^\top \leq \left(\vec{0}\right)^\top \right\}$$

That is, $\mathcal{H}$ is the polyhedron represented by $\mathbf{A}$ and $\vec{b}$ and $\mathcal{H}_{\leq 0}$ is the *characteristic cone* of $\mathcal{H}$.

**Example 7** (**Poisson multinomial variable (PMV) $\vec{X}_{\vec{\pi}}$**). *In the setting of Example 1, we have $q = m! = 6$. Let $n = 2$ and $\vec{\pi} = (\pi^2, \pi^1)$. $\vec{X}_{\vec{\pi}}$ is the histogram of two random variables $Y_1, Y_2$ over $[q]$, where $Y_1$ (respectively, $Y_2$) is distributed as $\pi^2$ (respectively, $\pi^1$).*

*For example, let $\vec{x} \in \{0, 1, 2\}^{\mathcal{L}(\mathcal{A})}$ denote the vector whose $123$ and $231$ components are $1$ and all other components are $0$. We have $\Pr(\vec{X}_{\vec{\pi}} = \vec{x}) = \frac{1}{4} \times \frac{3}{8} + \frac{1}{8} \times \frac{1}{8} = \frac{7}{64}$.*

**Definition 4** (**Almost complement**). *Let $\mathcal{C}$ denote a union of finitely many polyhedra. We say that a union of finitely many polyhedra $\mathcal{C}^*$ is an almost complement of $\mathcal{C}$, if (1) $\mathcal{C} \cap \mathcal{C}^* = \emptyset$ and (2) $\mathbb{Z}^q \subseteq \mathcal{C} \cup \mathcal{C}^*$.*

$\mathcal{C}^*$ is called an "almost complement" (instead of "complement") of $\mathcal{C}$ because $\mathcal{C}^* \cup \mathcal{C} \neq \mathbb{R}^q$. Effectively, $\mathcal{C}^*_{\leq 0}$ can be viewed as the complement of $\mathcal{C}$ when only integer vectors are concerned. It it not hard to see that $\mathcal{C}$ is an almost complement of $\mathcal{C}^*$. The following result states that the characteristic cones of $\mathcal{C}$ and $\mathcal{C}^*$, which may overlap, cover $\mathbb{R}^q$.

**Proposition 1.** *For any union of finitely many polyhedra $\mathcal{C}$ and any almost complement $\mathcal{C}^*$ of $\mathcal{C}$, we have $\mathcal{C}_{\leq 0} \cup \mathcal{C}^*_{\leq 0} = \mathbb{R}^q$.*

*Proof.* Suppose for the sake of contradiction that $\mathcal{C}_{\leq 0} \cup \mathcal{C}^*_{\leq 0} \neq \mathbb{R}^q$. Let $\vec{x} \in \mathbb{R}^q \setminus (\mathcal{C}_{\leq 0} \cup \mathcal{C}^*_{\leq 0})$ with $|\vec{x}|_1 = 1$. Because $\mathcal{C}_{\leq 0}$ and $\mathcal{C}^*_{\leq 0}$ are unions of polyhedra, there exists an $\delta > 0$ neighborhood $B_\delta = \{\vec{x}' \in \mathbb{R}^q : |\vec{x}' - \vec{x}|_\infty \leq \delta\}$ of $\vec{x}$ in $\mathbb{R}^q$ that is $\eta > 0$ away from $\mathcal{C}_{\leq 0} \cup \mathcal{C}^*_{\leq 0}$. Therefore, there exists $n \in \mathbb{N}$ with $n > \frac{1}{\delta}$ such that $nB_\delta = \{n\vec{x}' : \vec{x}' \in B_\delta\}$ do not overlap $\mathcal{C} \cup \mathcal{C}^*$. Because the radius of $nB_\delta$ is larger than $1$, there exists an integer vector in $nB_\delta$, which contradicts the assumption that $\mathbb{Z}^q \subseteq \mathcal{C} \cup \mathcal{C}^*$. □

W.l.o.g., in this paper we assume that all polyhedra are represented by integer matrices $\mathbf{A}$ where the entries of each row are coprimes, which means that the greatest common divisor of all entries in the

row is 1. For any $\mathcal{C} = \bigcup_{i \leq I} \mathcal{H}_i$ where $\mathcal{H}_i$ is the polyhedron characterized by integer matrices $\mathbf{A}_i$ with coprime entries and $\vec{\mathbf{b}}_i$, its almost complement always exists and is not unique. Let us define an specific almost complement of $\mathcal{C}$ that will be commonly used in this paper.

**Definition 5** (**Standard almost complement**). *Let $\mathcal{C} = \cup_{i \leq I} \mathcal{H}_i$ denote a union of $I$ rational polyhedra characterized by $\mathbf{A}_i$ and $\vec{\mathbf{b}}_i$, we define its* standard almost complement, *denoted by $\hat{\mathcal{C}}$, as follows.*

$$\hat{\mathcal{C}} = \bigcup_{\vec{a}_i \in \mathbf{A}_i : \forall i \leq I} \bigcap_{i \leq I} \left\{ \vec{x} \in \mathbb{R}^q : -\vec{a}_i \cdot \vec{x} \leq -b_i' - 1 \right\},$$

*where $\vec{a}_i$ is a row in $\mathbf{A}_i$ and $b_i'$ is the corresponding component in $\vec{\mathbf{b}}_i$. We write $\hat{\mathcal{C}} = \bigcup_{i^* \leq \hat{I}} \hat{\mathcal{H}}_{i^*}$, where $\hat{I} \in \mathbb{N}$ and each $\hat{\mathcal{H}}_{i^*}$ is a rational polyhedron.*

It is not hard to verify that $\hat{\mathcal{C}}$ is indeed an almost complement of $\mathcal{C}$. Let us take a look at a simple example for $q = 2$.

**Example 8.** *Let $\mathcal{C} = \mathcal{H}_1 \cup \mathcal{H}_2$, where $\mathcal{H}_1 = \left\{ \vec{x} \in \mathbb{R}^2 : \begin{bmatrix} -1 & 0 \\ 2 & -1 \end{bmatrix} \cdot (\vec{x})^\top \leq \begin{bmatrix} 0 \\ -2 \end{bmatrix} \right\}$ and*

$\mathcal{H}_2 = \left\{ \vec{x} \in \mathbb{R}^2 : \begin{bmatrix} -1 & 2 \\ 1 & -2 \end{bmatrix} \cdot (\vec{x})^\top \leq \begin{bmatrix} 8 \\ 8 \end{bmatrix} \right\}$. *It follows that $\hat{\mathcal{C}} = \hat{\mathcal{H}}_1 \cup \hat{\mathcal{H}}_2 \cup \hat{\mathcal{H}}_3 \cup \hat{\mathcal{H}}_4$, where*

$$\hat{\mathcal{H}}_1 = \left\{ \vec{x} \in \mathbb{R}^2 : \begin{bmatrix} 1 & 0 \\ 1 & -2 \end{bmatrix} \cdot (\vec{x})^\top \leq \begin{bmatrix} -1 \\ -9 \end{bmatrix} \right\}, \hat{\mathcal{H}}_2 = \left\{ \vec{x} \in \mathbb{R}^2 : \begin{bmatrix} 1 & 0 \\ -1 & 2 \end{bmatrix} \cdot (\vec{x})^\top \leq \begin{bmatrix} -1 \\ -9 \end{bmatrix} \right\}$$

$$\hat{\mathcal{H}}_3 = \left\{ \vec{x} \in \mathbb{R}^2 : \begin{bmatrix} -2 & 1 \\ 1 & -2 \end{bmatrix} \cdot (\vec{x})^\top \leq \begin{bmatrix} 1 \\ -9 \end{bmatrix} \right\}, \hat{\mathcal{H}}_4 = \left\{ \vec{x} \in \mathbb{R}^2 : \begin{bmatrix} -2 & 1 \\ -1 & 2 \end{bmatrix} \cdot (\vec{x})^\top \leq \begin{bmatrix} 1 \\ -9 \end{bmatrix} \right\}$$

*Figure 4 (a) shows $\mathcal{C}$ and $\hat{\mathcal{C}}$. Figure 4 (b) shows $\mathcal{C}_{\leq 0}$ and $\hat{\mathcal{C}}_{\leq 0}$, where $\mathcal{H}_2$ is a one-dimensional polyhedron, i.e., a straight line. Note that $\mathcal{C} \cup \hat{\mathcal{C}} \neq \mathbb{R}^q$ and $\mathcal{C}_{\leq 0} \cup \hat{\mathcal{C}}_{\leq 0} = \mathbb{R}^q$.*

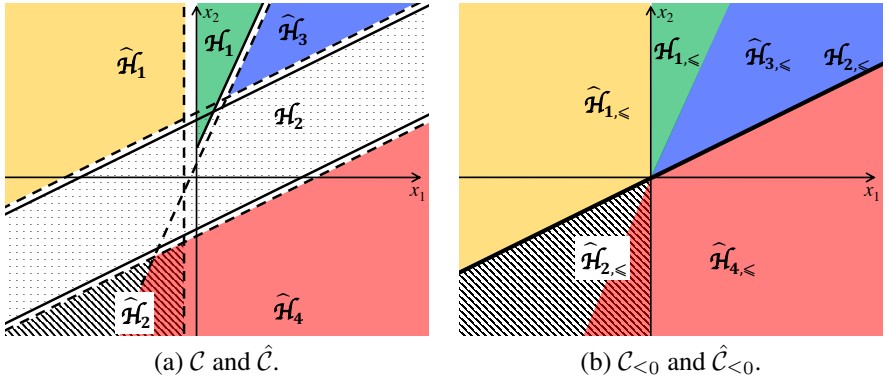

(a) $\mathcal{C}$ and $\hat{\mathcal{C}}$.  (b) $\mathcal{C}_{\leq 0}$ and $\hat{\mathcal{C}}_{\leq 0}$.

Figure 4: In (a), $\mathcal{C} = \mathcal{H}_1 \cup \mathcal{H}_2$, where $\mathcal{H}_1$ is the green area and $\mathcal{H}_2$ is a shaded area, and $\hat{\mathcal{C}} = \hat{\mathcal{H}}_1 \cup \hat{\mathcal{H}}_2 \cup \hat{\mathcal{H}}_3 \cup \hat{\mathcal{H}}_4$, where $\hat{\mathcal{H}}_2$ is a shaded area, and $\hat{\mathcal{H}}_1$, $\hat{\mathcal{H}}_3$, and $\hat{\mathcal{H}}_4$ are the yellow, red, and blue areas, respectively. In (b), $\mathcal{C}_{\leq 0} \cup \hat{\mathcal{C}}_{\leq 0} = \mathbb{R}^q$, where $\hat{\mathcal{H}}_2$ is a straight line.

To present the categorization lemma, we recall the definitions of $\alpha_n$, $\beta_n$, and Theorem 2 in [55]. We first recall the definition of the *activation graph*.

**Definition 6** (**Activation graph** [55]). *For each $\Pi$, $\mathcal{H}_i$, and $n \in \mathbb{N}$, the* activation graph, *denoted by $\mathcal{G}_{\Pi, \mathcal{C}, n}$, is defined to be the complete bipartite graph with two sets of vertices $CH(\Pi)$ and $\{\mathcal{H}_i : i \leq I\}$, and the weight on the edge $(\pi, \mathcal{H}_i)$ is defined as follows.*

$$w_n(\pi, \mathcal{H}_i) \triangleq \begin{cases} -\infty & \text{if } \mathcal{H}_{i,n}^{\mathbb{Z}} = \emptyset \\ -\frac{n}{\log n} & \text{otherwise, if } \pi \notin \mathcal{H}_{i, \leq 0} \\ \dim(\mathcal{H}_{i, \leq 0}) & \text{otherwise} \end{cases},$$

*where $\mathcal{H}_{i,n}^{\mathbb{Z}}$ is the set of non-negative integer vectors in $\mathcal{H}_i$ whose $L_1$ norm is $n$.*

Definition 6 slightly abuses notation, because its vertices $\{\mathcal{H}_i : i \leq I\}$ are not explicitly indicated in the subscript of $\mathcal{G}_{\Pi,\mathcal{C},n}$. This does not cause confusion when they are clear from the context.

When $\mathcal{H}_{i,n}^{\mathbb{Z}} = \emptyset$ we say that $\mathcal{H}_i$ is *inactive (at n)*, and when $\mathcal{H}_{i,n}^{\mathbb{Z}} \neq \emptyset$ we say that $\mathcal{H}_i$ is *active (at n)*. In addition, if the weight on any edge $(\pi, \mathcal{H}_i)$ is positive, then we say that $\pi$ is *active* and is *activated* by $\mathcal{H}_i$ (which must be active at $n$).

Roughly speaking, for any sufficiently large $n$ and $\vec{\pi} = (\pi_1, \ldots, \pi_n) \in \Pi^n$, let $\pi = \frac{1}{n} \sum_{j=1}^{n} \pi_j$, then [55, Theorem 1] implies

$$\Pr(\vec{X}_{\vec{\pi}} \in \mathcal{H}_i) \approx n^{w_n(\pi, \mathcal{H}_i) - q}$$

It follows that $\Pr(\vec{X}_{\vec{\pi}} \in \mathcal{C})$ is mostly determined by the heaviest weight on edges connected to $\pi$, denoted by $\dim_{\mathcal{C},n}^{\max}(\pi)$, which is formally defined as follows:

$$\dim_{\mathcal{C},n}^{\max}(\pi) \triangleq \max_{i \leq I} w_n(\pi, \mathcal{H}_i)$$

Then, a max-(respectively, min-) adversary aims to choose $\vec{\pi} = (\pi_1, \ldots, \pi_n) \in \Pi^n$ to maximize (respectively, minimize) $\dim_{\mathcal{C},n}^{\max}(\frac{1}{n} \sum_{j=1}^{n} \pi_j)$, which are characterized by $\alpha_n$ (respectively, $\beta_n$) defined as follows.

$$\alpha_n \triangleq \max_{\pi \in \mathrm{CH}(\Pi)} \dim_{\mathcal{C},n}^{\max}(\pi)$$
$$\beta_n \triangleq \min_{\pi \in \mathrm{CH}(\Pi)} \dim_{\mathcal{C},n}^{\max}(\pi)$$

We further define the following notation that will be frequently used in the proofs of this paper. Let $\mathcal{C}_n^{\mathbb{Z}}$ denote the set of all non-negative integer vectors in $\mathcal{C}$ whose $L_1$ norm is $n$. That is,

$$\mathcal{C}_n^{\mathbb{Z}} = \bigcup_{i \leq I} \mathcal{H}_{i,n}^{\mathbb{Z}}$$

By definition, $\mathcal{C}_n^{\mathbb{Z}} = \emptyset$ if and only if all $\mathcal{H}_i$'s are inactive at $n$. Therefore, we have

$$(\alpha_n = -\infty) \iff (\beta_n = -\infty) \iff (\mathcal{C}_n^{\mathbb{Z}} = \emptyset)$$

For completeness, we recall [55, Theorem 2] below.

**Theorem 2 in [55] (Semi-random likelihood of PMV-in-$\mathcal{C}$).** *Given any $q, I \in \mathbb{N}$, any closed and strictly positive $\Pi$ over $[q]$, and any set $\mathcal{C} = \bigcup_{i \leq I} \mathcal{H}_i$ that is the union of finitely many polyhedra with integer matrices, for any $n \in \mathbb{N}$,*

$$\sup_{\vec{\pi} \in \Pi^n} \Pr\left(\vec{X}_{\vec{\pi}} \in \mathcal{C}\right) = \begin{cases} 0 & \text{if } \alpha_n = -\infty \\ \exp(-\Theta(n)) & \text{if } -\infty < \alpha_n < 0 \\ \Theta\left(n^{\frac{\alpha_n - q}{2}}\right) & \text{otherwise (i.e. } \alpha_n \geq 0) \end{cases},$$

$$\inf_{\vec{\pi} \in \Pi^n} \Pr\left(\vec{X}_{\vec{\pi}} \in \mathcal{C}\right) = \begin{cases} 0 & \text{if } \beta_n = -\infty \\ \exp(-\Theta(n)) & \text{if } -\infty < \beta_n < 0 \\ \Theta\left(n^{\frac{\beta_n - q}{2}}\right) & \text{otherwise (i.e. } \beta_n \geq 0) \end{cases}.$$

For any almost complement $\mathcal{C}^*$ of $\mathcal{C}$, let $\alpha_n^*$ and $\beta_n^*$ denote the counterparts of $\alpha_n$ and $\beta_n$ for $\mathcal{C}^*$, respectively. We note that $\alpha_n^*$ and $\beta_n^*$ depend on the polyhedra used to representation $\mathcal{C}^*$. We are now ready to present the full version of the categorization lemma as follows.

**Lemma 1. (Categorization Lemma, Full Version).** *Given any $q, I \in \mathbb{N}$, any closed and strictly positive $\Pi$ over $[q]$, any $\mathcal{C} = \bigcup_{i \leq I} \mathcal{H}_i$ and its almost complement $\mathcal{C}^* = \bigcup_{i^* \leq I^*} \mathcal{H}_{i^*}^*$, for any $n \in \mathbb{N}$,*

$$\inf_{\vec{\pi} \in \Pi^n} \Pr\left(\vec{X}_{\vec{\pi}} \in \mathcal{C}\right) = \begin{cases} 0 & \text{if } \beta_n = -\infty \\ \exp(-\Theta(n)) & \text{if } -\infty < \beta_n < 0 \\ \Theta\left(n^{\frac{\beta_n - q}{2}}\right) & \text{if } 0 \leq \beta_n < q \\ \Theta(1) \wedge (1 - \Theta(1)) & \text{if } \alpha_n^* = \beta_n = q \\ 1 - \Theta\left(n^{\frac{\alpha_n^* - q}{2}}\right) & \text{if } 0 \leq \alpha_n^* < q \\ 1 - \exp(-\Theta(n)) & \text{if } -\infty < \alpha_n^* < 0 \\ 1 & \text{if } \alpha_n^* = \infty \end{cases}$$

$$\sup_{\vec{\pi} \in \Pi^n} \Pr\left(\vec{X}_{\vec{\pi}} \in \mathcal{C}\right) = \begin{cases} 0 & \text{if } \alpha_n = -\infty \\ \exp(-\Theta(n)) & \text{if } -\infty < \alpha_n < 0 \\ \Theta\left(n^{\frac{\alpha_n - q}{2}}\right) & \text{if } 0 \le \alpha_n < q \\ \Theta(1) \wedge (1 - \Theta(1)) & \text{if } \alpha_n = \beta_n^* = q \\ 1 - \Theta\left(n^{\frac{\beta_n^* - q}{2}}\right) & \text{if } 0 \le \beta_n^* < q \\ 1 - \exp(-\Theta(n)) & \text{if } -\infty < \beta_n^* < 0 \\ 1 & \text{if } \beta_n^* = -\infty \end{cases}$$

*Proof.* We present the proof for the $\inf$ part of Lemma 1 and the proof for the $\sup$ part is similar. Notice that $\mathbb{Z}^q \subseteq \mathcal{C} \cup \mathcal{C}^*$, we have:

$$\inf_{\vec{\pi} \in \Pi^n} \Pr\left(\vec{X}_{\vec{\pi}} \in \mathcal{C}\right) = 1 - \sup_{\vec{\pi} \in \Pi^n} \Pr\left(\vec{X}_{\vec{\pi}} \in \mathcal{C}^*\right)$$

The proof is done by combining the $\inf$ part of [55, Theorem 2] (applied to $\mathcal{C}$) and one minus the $\sup$ part of [55, Theorem 2] (applied to $\mathcal{C}^*$).

- **The $0$, $\exp(-\Theta(n))$ and $\Theta\left(n^{\frac{\beta_n - q}{2}}\right)$ cases** follow after the corresponding $\inf$ part of [55, Theorem 2] applied to $\mathcal{C}$.

- **The $\Theta(1) \wedge (1 - \Theta(1))$ case.** The condition of this case implies that the polynomial bounds in the $\inf$ part of [55, Theorem 2] (applied to $\mathcal{C}$) hold, which means that $\inf_{\vec{\pi} \in \Pi^n} \Pr\left(\vec{X}_{\vec{\pi}} \in \mathcal{C}\right) = \Theta(1)$, and the polynomial bounds in the $\sup$ part of [55, Theorem 2] (applied to $\mathcal{C}^*$) hold, which means that

$$\inf_{\vec{\pi} \in \Pi^n} \Pr\left(\vec{X}_{\vec{\pi}} \in \mathcal{C}\right) = 1 - \sup_{\vec{\pi} \in \Pi^n} \Pr\left(\vec{X}_{\vec{\pi}} \in \mathcal{C}^*\right) = 1 - \Theta(1)$$

- **The $1 - \Theta\left(n^{\frac{\alpha_n^* - q}{2}}\right)$, $1 - \exp(-\Theta(n))$, and $1$ cases** follow after one minus the $\sup$ part of [55, Theorem 2] (applied to $\mathcal{C}^*$).

$\square$

**Remarks.** The conditions for all, except $0$ and $1$, cases are different between $\sup$ and $\inf$ parts of the lemma. Moreover, the degrees of polynomial in the L and U cases may be different between $\sup$ and $\inf$ parts. Let us use the setting in Example 8 and Figure 5 to illustrate the conditions for the $\inf$ case. For the purpose of illustration, we assume that all polyhedra in $\mathcal{C}$ and $\mathcal{C}^*$ are active at $n$.

- **The $0$ (respectively, $1$) case** holds when no nonnegative integer with $L_1$ norm $n$ is in $\mathcal{C}$ (respectively, in $\mathcal{C}^*$).

- **The VU case.** Given that the $0$ and $1$ cases do not hold, the VU case holds when $\text{CH}(\Pi)$ contains a distribution $\pi_{\text{VU}}$ that is not in $\mathcal{C}_{\le 0}$. Notice that $\mathcal{C}_{\le 0}$ is a closed set and $\mathcal{C}_{\le 0} \cup \mathcal{C}_{\le 0}^* = \mathbb{R}^q$. This means that $\pi_{\text{VU}}$ is an interior point of $\mathcal{C}_{\le 0}^*$. For example, in Figure 5, $\pi_{\text{VU}}$ is not in $\mathcal{C}_{\le 0}$ and is an interior point of $\hat{\mathcal{H}}_{3,\le 0}$.

Figure 5: An Illustration of $\pi_{\text{VU}}$, $\pi_{\text{U}}$, $\pi_{\text{M}}$, and $\pi_{\text{VL}}$ for the $\inf$ part of Lemma 1.

- **The U case** holds when $\text{CH}(\Pi) \subseteq \mathcal{C}_{\le 0}$, and $\text{CH}(\Pi)$ contains a distribution $\pi_{\text{U}}$ that lies on a (low-dimensional) boundary of $\mathcal{C}_{\le 0}$. For example, in Figure 5, $\pi_{\text{U}}$ lies in a 1-dimensional polyhedron $\mathcal{H}_{2,\le 0} \subseteq \mathcal{C}_{\le 0}$, and is not in any 2-dimensional polyhedron in $\mathcal{C}_{\le 0}$.

- **The M case** holds when the U case does not hold, and CH($\Pi$) contains a distribution $\pi_M$ that lies in the intersection of a $q$-dimensional subspace of $\mathcal{C}_{\leq 0}$ and a $q$-dimensional subspace of $\mathcal{C}^*_{\leq 0}$. For example, in Figure 5, $\pi_U$ lies in $\mathcal{H}_{1,\leq 0}$ and $\hat{\mathcal{H}}_{3,\leq 0}$, both of which are 2-dimensional.

- **The L case holds** when every distribution in CH($\Pi$) is in a $q$-dimensional subspace of $\mathcal{C}_{\leq 0}$, and there exists $\pi_L \in$ CH($\Pi$) that lies in a (low-dimensional) boundary of $\mathcal{C}^*_{\leq 0}$. No such $\pi_L$ exists in Figure 5's example, but if we apply Lemma 1 to $\mathcal{C}^*$, then $\pi_U$ in Figure 5 is an example of $\pi_L$ for $\mathcal{C}^*$.

- **The VL case holds** when every distribution in CH($\Pi$) is an inner point of $\mathcal{C}_{\leq 0}$. For example, in Figure 5, $\pi_{VL}$ is an inner point of $\mathcal{H}_{1,\leq 0} \subseteq \mathcal{C}$.

## D  GISRs and Their Algebraic Properties

### D.1  Definition of GISRs

All irresolute voting rules studied in this paper are generalized irresolute scoring rules (GISR) [22, 53], whose resolute versions are known as *generalized scoring rules (GSRs)* [56]. We recall the definition of GISRs based on separating hyperplanes [57, 38].

For any real number $x$, let Sign($x$) $\in \{+, -, 0\}$ denote the sign of $x$. Given a set of $K$ hyperplanes in the $q$-dimensional Euclidean space, denoted by $\vec{H} = (\vec{h}_1, \ldots, \vec{h}_K)$, for any $\vec{x} \in \mathbb{R}^q$, we let $\text{Sign}_{\vec{H}}(\vec{x}) = (\text{Sign}(\vec{x} \cdot \vec{h}_1), \ldots, \text{Sign}(\vec{x} \cdot \vec{h}_K))$. In other words, for any $k \leq K$, the $k$-th component of $\text{Sign}_{\vec{H}}(\vec{x})$ equals to 0, if $\vec{p}$ lies in hyperplane $\vec{h}_k$; and it equals to $+$ (respectively, $-$) if $\vec{p}$ lies in the positive (respectively, negative) side of $\vec{h}_k$. Each element in $\{+, -, 0\}^K$ is called a *signature*.

**Definition 7** (**Generalized irresolute scoring rule (GISR)**). *A generalized irresolute scoring rule (GISR) $\bar{r}$ is defined by (1) a set of $K \geq 1$ hyperplanes $\vec{H} = (\vec{h}_1, \ldots, \vec{h}_K) \in (\mathbb{R}^{m!})^K$ and (2) a function $g : \{+, -, 0\}^K \rightarrow (2^{\mathcal{A}} \setminus \emptyset)$. For any profile $P$, we let $\bar{r}(P) = g(\text{Sign}_{\vec{H}}(\text{Hist}(P)))$. $\bar{r}$ is called an* integer GISR (int-GISR) *if $\vec{H} \in (\mathbb{Z}^{m!})^K$. If for all profiles $P$, we have $|\bar{r}(P)| = 1$, then $\bar{r}$ is called a generalized scoring rule (GSR). Int-GSRs are defined similarly to int-GISRs.*

**Definition 8** (**Feasible and atomic signatures**). *Given integer $\vec{H}$ with $K = |\vec{H}|$, let $\mathcal{S}_K = \{+, -, 0\}^K$. A signature $\vec{t} \in \mathcal{S}_K$ is* feasible, *if there exists $\vec{x} \in \mathbb{R}^{m!}$ such that $\text{Sign}_{\vec{H}}(\vec{x}) = \vec{t}$. Let $\mathcal{S}_{\vec{H}} \subseteq \mathcal{S}_K$ denote the set of all feasible signatures.*

*A signature $\vec{t}$ is called an* atomic signature *if and only if $\vec{t} \in \{+, -\}^K$. Let $\mathcal{S}^{\circ}_{\vec{H}}$ denote the set of all feasible atomic signatures.*

The domain of any GISR $\bar{r}$ can be naturally extended to $\mathbb{R}^{m!}$ and to $\mathcal{S}_{\vec{H}}$. Specifically, for any $\vec{t} \in \mathcal{S}_{\vec{H}}$ we let $\bar{r}(\vec{t}) = g(\vec{t})$. It suffices to define $g$ on the feasible signatures, i.e., $\mathcal{S}_{\vec{H}}$.

Notice that the same voting rule can be represented by different combinations of $(\vec{H}, g)$. In the following section we recall int-GISR representations of the voting rules studied in this paper.

### D.2  Commonly-Studied Voting Rules as GISRs

As discussed in [55], the irresolute versions of Maximin, Copeland$_\alpha$, Ranked Pairs, and Schulze belong to the class of *edge-order-based (*EO-based) rules, which are defined over the weak order on edges in WMG($P$). We recall its formal definition below.

**Definition 9** (**Edge-order-based rules**). *A (resolute or irresolute) voting rule $\bar{r}$ is* edge-order-based (EO-based)*, if for any pair of profiles $P_1$ and $P_2$ such that for every combination of four different alternatives $\{a, b, c, d\} \subset \mathcal{A}$, $[w_{P_1}(a, b) \geq w_{P_1}(c, d)] \Leftrightarrow [w_{P_2}(a, b) \geq w_{P_2}(c, d)]$, we have $\bar{r}(P_1) = \bar{r}(P_2)$.*

All EO-based rules can be represented by a GISR using a set of hyperplanes that represents the orders over WMG edges. We first recall pairwise difference vectors as follows.

**Definition 10** (**Pairwise difference vectors [54]**). *For any pair of different alternatives $a, b$, let $Pair_{a,b}$ denote the $m!$-dimensional vector indexed by rankings in $\mathcal{L}(\mathcal{A})$: for any $R \in \mathcal{L}(\mathcal{A})$, the $R$-component of $Pair_{a,b}$ is 1 if $a \succ_R b$; otherwise it is $-1$.*

We now define the hyperplanes for edge-order-based rules.

**Definition 11 ($\vec{H}_{EO}$).** $\vec{H}_{EO}$ consists of $\binom{m(m-1)}{2}$ hyperplanes indexed by $\vec{h}_{e_1,e_2}$, where $e_1 = (a_1, a_2)$ and $e_2 = (a_2, b_2)$ are two different pairs of alternatives, such that

$$\vec{h}_{e_1,e_2} = Pair_{a_1,b_1} - Pair_{a_2,b_2}$$

That is, for any (fractional) profile $P$, $\vec{h}_{e_1,e_2} \cdot \text{Hist}(P) \leq 0$ if and only if the weight on $e_1$ in WMG($P$) is no more than the weight on $e_2$ in WMG($P$). Therefore, given $\text{Sign}_{\vec{H}_{EO}}(P)$, we can compare the weights on pairs of edges, which leads to the weak order on edges in WMG($P$) w.r.t. their weights. Consequently, for any profile $P$, $\text{Sign}_{\vec{H}}(P)$ contains enough information to determine the (co-)winners under any edge-order-based rules. Formally, the GISR representations of these rules used in this paper are defined by $\vec{H}_{EO}$ and the following $g$ functions that mimic the procedures of choosing the winner(s).

**Definition 12.** *Let $\overline{MM}$, $\overline{Cd_\alpha}$, $\overline{RP}$, $\overline{Sch}$ denote the int-GISRs defined by $\vec{H}_{EO}$ and the following $g$ functions. Given a feasible signature $\vec{t} \in \mathcal{S}_{\vec{H}_{EO}}$,*

- *$g_{\text{MM}}$ first picks a representative edge $e_a$ whose weight is no more than all other outgoing edges of $a$, then compare the weights of $e_a$'s for all alternatives and choose alternatives $a$ whose $e_a$ has the highest weight as the winners.*

- *$g_{\text{Cd}_\alpha}$ compares weights on pairs of edges $a \to b$ and $b \to a$, and then calculate the Copeland$_\alpha$ scores accordingly. The winners are the alternatives with the highest Copeland$_\alpha$ score.*

- *$g_{\text{RP}}$ mimics the execution of PUT-Ranked Pairs, which only requires information about the weak order over edges w.r.t. their weights in WMG.*

- *$g_{\text{Sch}}$ first computes an edge $e_p$ with the minimum weight on any given directed path $p$, then for each pair of alternatives $a$ and $b$, computes an edge $e_{(a,b)}$ that represents the strongest edge among all paths from $a$ to $b$. $g_{\text{Sch}}$ then mimics Schulze to select the winner(s).*

While Copeland can be represented by $\vec{H}_{EO}$ and $g_{\text{Cd}_\alpha}$ as in the definition above, in this paper we use another set of hyperplanes, denoted by $\vec{H}_{\text{Cd}_\alpha}$, that represents the UMG of the profile. The reason is that in this way any refinement of $\text{Cd}_\alpha$ would break ties according to the UMG of the profile, which is needed in the proof of Theorem 4.

**Definition 13 ($\overline{\textbf{Cd}_\alpha}$ as a GISR).** *$\overline{Cd_\alpha}$ is represented by $\vec{H}_{Cd_\alpha}$ and $g_{Cd_\alpha}$ defined as follows. For every pair of different alternatives $(a, b)$, $\vec{H}_{Cd_\alpha}$ contains a hyperplane $\vec{h}_{(a,b)} = Pair_{a,b} - Pair_{b,a}$. For any profile $P$, $g_{Cd_\alpha}$ first computes the outcome of each head-to-head elections between alternatives $a$ and $b$ by checking $\vec{h}_{(a,b)} \cdot \text{Hist}(P)$, then calculate the Copeland$_\alpha$ score, and finally choose all alternatives with the maximum score as the winners.*

The GISR representation of MRSE rules is based on the fact that the winner(s) can be computed from comparing the scores between any pair of alternatives $(a, b)$ after a set of alternatives $B$ is removed. This idea is formalized in the following definition. For any $R \in \mathcal{L}(\mathcal{A})$ and any $B \subset \mathcal{A}$, let $R|_{\mathcal{A} \setminus B}$ denote the linear order over $(\mathcal{A} \setminus B)$ that is obtained from $R$ by removing alternatives in $B$.

**Definition 14 (MRSE rules as GISRs).** *Any MRSE $\overline{r} = (\overline{r}_2, \ldots, \overline{r}_m)$ is represented by $\vec{H}$ and $g_{\overline{r}}$ defined as follows. Given an int-MRSE rule $\overline{r} = (\overline{r}_2, \ldots, \overline{r}_m)$, for any pair of alternatives $a, b$ and any subset of alternatives $B \subseteq (\mathcal{A} \setminus \{a, b\})$, we let $\text{Score}_{B,a,b}^{\Delta}$ denote the vector, where for every $R \in \mathcal{L}(\mathcal{A})$, the $R$-th component of $\text{Score}_{B,a,b}^{\Delta}$ is $s_i^{m-|B|} - s_j^{m-|B|}$, where $i$ and $j$ are the ranks of $a$ and $b$ in $R|_{\mathcal{A} \setminus B}$, respectively.*

*For any pair of different alternatives $\{a, b\} \subseteq (\mathcal{A} \setminus B)$, $\vec{H}$ contains a hyperplane $\text{Score}_{B,a,b}^{\Delta}$. For any profile $P$, $g_{\overline{r}}$ mimics $\overline{r}$ to compute the PUT winners based on whether $\vec{h}_{(B,a,b)} \cdot \text{Hist}(P)$ is $< 0$, $= 0$, or $> 0$.*

In fact, the GISR representation of $\overline{r}$ in Definiton 14 corresponds to the *PUT structure* [55], which we do not discuss in this paper for simplicity of presentation. Any GSR refinement of $\overline{r}$, denoted

by $r$, uses the same $\vec{H}$ in Definiton 14 and a different $g$ function that always chooses a single loser to be eliminated in each round. The constraint is, for any profile $P$, the break-tie mechanisms used in $g$ only depends on $\text{Sign}_{\vec{H}}(P)$ (but not any other information contained in $P$). For example, lexicographic tie-breaking w.r.t. a fixed order over alternatives is allowed but using the first agent's vote to break ties is not allowed.

## D.3 Minimally Continuous GISRs

Next, we define (minimally) continuous GISR in a similar way as Freeman et al. [22], except that in this paper the domain of GISR is $\mathbb{R}^{m!}$ (in contrast to $\mathbb{R}^{m!}_{\geq 0}$ in [22]).

**Definition 15** (**(Minimally) continuous GISR**). *A GISR $\overline{r}$ is* continuous, *if for any $\vec{x} \in \mathbb{R}^{m!}$, any alternative $a$, and any sequence of vectors $(\vec{x}_1, \vec{x}_2 \ldots)$ that converges to $\vec{x}$,*

$$[\forall j \in \mathbb{N}, a \in \overline{r}(\vec{x}_j)] \Longrightarrow [a \in \overline{r}(\vec{x})]$$

*A GISR $\overline{r}$ is called* minimally continuous, *if it is continuous and there does not exist a continuous GISR $\overline{r}^*$ such that (1) for all $\vec{x} \in \mathbb{R}^{m!}$, $\overline{r}^*(\vec{x}) \subseteq \overline{r}(\vec{x})$, and (2) the inclusion is strict for some $\vec{x}$.*

Equivalently, a continuous GISR $\overline{r}$ is minimally continuous if and only if the (fractional) profiles with unique winners is a dense subset of $\mathbb{R}^{m!}$. That is, for any vector in $\mathbb{R}^{m!}$, there exists a sequence of profiles with unique winners that converge to it. As commented by Freeman et al. [22], many commonly-studied irresolute voting rules are continuous GISRs. It is not hard to verify that positional scoring rules and MRSE rules are minimally continuous GISRs, which is formally proved in the following proposition.

**Proposition 2.** *Positional scoring rules and MRSE rules are minimally continuous.*

*Proof.* Let $\vec{s} = (s_1, \ldots, s_m)$ denote the scoring vector. We first prove that $\overline{r}_{\vec{s}}$ is continuous. For any $\vec{x} \in \mathbb{R}^{m!}$, any $a \in \mathcal{A}$, and any sequence $(\vec{x}_1, \vec{x}_2, \ldots)$ that converges to $\vec{x}$ such that for all $j \geq 1$, $a \in \overline{r}(\vec{x}_j)$, we have that for every $b \in \mathcal{A}$, $\vec{s}(\vec{x}_j, a) \geq \vec{s}(\vec{x}_j, b)$. Notice that $\vec{s}(\vec{x}_j, a)$ (respectively, $\vec{s}(\vec{x}_j, b)$) converges to $\vec{s}(\vec{x}, a)$ (respectively, $\vec{s}(\vec{x}, b)$). Therefore, $\vec{s}(\vec{x}, a) \geq \vec{s}(\vec{x}, b)$, which means that $a \in \overline{r}_{\vec{s}}(\vec{x})$, i.e., $\overline{r}_{\vec{s}}$ is continuous.

To prove that $\overline{r}_{\vec{s}}$ is minimally continuous, it suffices to prove that for any $\vec{x} \in \mathbb{R}^{m!}$ and any $a \in \overline{r}_{\vec{s}}(\vec{x})$, there exists a sequence $(\vec{x}_1, \vec{x}_2, \ldots)$ that converges to $\vec{x}$ such that for all $j \geq 1$, $\overline{r}(\vec{x}_j) = \{a\}$. Let $\sigma$ denote an arbitrary cyclic permutation among $\mathcal{A} \setminus \{a\}$ and $P$ denote the following $(m-1)$-profile.

$$P = \left\{ \sigma^i(a \succ \text{others}) : 1 \leq i \leq m-1 \right\}$$

Then, for every $j \in \mathbb{N}$, we let $\vec{x}_j = \vec{x} + \frac{1}{j}\text{Hist}(P)$. It is easy to check that $\overline{r}(\vec{x}_j) = \{a\}$, which proves the minimal continuity of $\overline{r}_{\vec{s}}$.

Let $\overline{r} = (\overline{r}_2, \ldots, \overline{r}_m)$ denote the MRSE rule. We will use notation in Section E.3 to prove the proposition for $\overline{r}$. We first prove that $\overline{r}$ is continuous. Let $\vec{x} \in \mathbb{R}^{m!}$, $a \in \mathcal{A}$, and $(\vec{x}_1, \vec{x}_2, \ldots)$ be a sequence that converges to $\vec{x}$ such that for all $j \geq 1$, $a \in \overline{r}(\vec{x}_j)$. Because the number of different parallel universes is finite (more precisely, $m!$), there exists a subsequence of $(\vec{x}_1, \vec{x}_2, \ldots)$, denoted by $(\vec{x}'_1, \vec{x}'_2, \ldots)$, and a parallel universe $O \in \mathcal{L}(\mathcal{A})$ where $a$ is ranked in the last position (i.e., $a$ is the winner), such that for all $j \in \mathbb{N}$, $O$ is a parallel universe when executing $\overline{r}$ on $\vec{x}'_j$. Therefore, for all $1 \leq i \leq m-1$, in round $i$, $O[i]$ has the lowest $\overline{r}_{m+1-i}$ score in $\vec{x}'_j|_{O[i,m]}$ among alternatives in $O[i, m]$. It follows that $O[i]$ has the lowest $\overline{r}_{m+1-i}$ score in $\vec{x}|_{O[i,m]}$ among alternatives in $O[i, m]$, which means that $O$ is also a parallel universe when executing $\overline{r}$ on $\vec{x}$. This proves that $\overline{r}$ is continuous.

The proof of minimal continuity of $\overline{r}$ is similar to the proof for positional scoring rules presented above. For any $\vec{x} \in \mathbb{R}^{m!}$ and any $a \in \overline{r}_{\vec{s}}(\vec{x})$, let $O$ denote a parallel universe where $a$ is ranked in the last position. Let $P$ denote the following profile of $(m-1)! + (m-2)! + \cdots + 2!$ votes, where $O$ is the unique parallel universe.

$$P = \bigcup_{i=1}^{m-1} \left\{ O[1] \succ \cdots \succ O[i] \succ R_i : \forall R_i \in \mathcal{L}(O[i+1, m]) \right\}$$

For any $j \in \mathbb{N}$, let $\vec{x}_j = \vec{x} - \frac{1}{j}\text{Hist}(P)$. It is not hard to verify that $(\vec{x}_1, \vec{x}_2, \ldots)$ converges to $\vec{x}$, and for every $1 \leq i \leq m-1$ and every $j \in \mathbb{N}$, alternative $O[i]$ is the unique loser in round $i$, where

$-\frac{1}{j}\text{Hist}(P)$ is used as the tie-breaker. This means that for all $j \in \mathbb{N}$, $\bar{r}(\vec{x}_j) = \{a\}$, which proves the minimal continuity of $\bar{r}$. $\qquad\square$

## D.4   Algebraic Properties of GISRs

We first define the refinement relationship among (feasible or infeasible) signatures.

**Definition 16** (**Refinement relationship** $\trianglelefteq$). *For any pair of signatures $\vec{t}_1, \vec{t}_2 \in \mathcal{S}_K$, we say that $\vec{t}_1$ refines $\vec{t}_2$, denoted by $\vec{t}_1 \trianglelefteq \vec{t}_2$, if for every $k \leq K$, if $[\vec{t}_2]_k \neq 0$ then $[\vec{t}_1]_k = [\vec{t}_2]_k$. If $\vec{t}_1 \trianglelefteq \vec{t}_2$ and $\vec{t}_1 \neq \vec{t}_2$, then we say that $\vec{t}_1$ strictly refines $\vec{t}_2$, denoted by $\vec{t}_1 \triangleleft \vec{t}_2$.*

In words, $\vec{t}_1$ refines $\vec{t}_2$ if $\vec{t}_1$ differs from $\vec{t}_2$ only on the 0 components in $\vec{t}_2$. By definition, $\vec{t}_1$ refines itself. Next, given $\vec{H}$ and a feasible signature $\vec{t}$, we define a polyhedron $\mathcal{H}^{\vec{H},\vec{t}}$ to represent profiles whose signatures are $\vec{t}$.

**Definition 17** ($\mathcal{H}^{\vec{H},\vec{t}}$ ($\mathcal{H}^{\vec{t}}$ **in short**)). *For any $\vec{H} = (\vec{h}_1, \ldots, \vec{h}_K) \in (\mathbb{R}^d)^K$ and any $\vec{t} \in \mathcal{S}_{\vec{H}}$, we let*

$$\mathbf{A}^{\vec{t}} = \begin{bmatrix} \mathbf{A}^{\vec{t}}_{+} \\ \mathbf{A}^{\vec{t}}_{-} \\ \mathbf{A}^{\vec{t}}_{0} \end{bmatrix}, \text{ where}$$

- $\mathbf{A}^{\vec{t}}_{+}$ *consists of a row $-\vec{h}_i$ for each $i \leq K$ with $t_i = +$.*
- $\mathbf{A}^{\vec{t}}_{-}$ *consists of a row $\vec{h}_i$ for each $i \leq K$ with $t_i = -$.*
- $\mathbf{A}^{\vec{t}}_{0}$ *consists of two rows $-\vec{h}_i$ and $\vec{h}_i$ for each $i \leq K$ with $t_i = 0$.*

*Let $\vec{\mathbf{b}}^{\vec{t}} = [\underbrace{-\vec{1}}_{\text{for } \mathbf{A}^{\vec{t}}_{+}}, \underbrace{-\vec{1}}_{\text{for } \mathbf{A}^{\vec{t}}_{-}}, \underbrace{\vec{0}}_{\text{for } \mathbf{A}^{\vec{t}}_{0}}]$. The corresponding polyhedron is denoted by $\mathcal{H}^{\vec{H},\vec{t}}$, or $\mathcal{H}^{\vec{t}}$ in short when $\vec{H}$ is clear from the context.*

The following proposition follows immediately after the definition.

**Proposition 3.** *Given $\vec{H}$, for any pair of feasible signatures $\vec{t}_1, \vec{t}_2 \in \mathcal{S}_{\vec{H}}$, $\vec{t}_1 \trianglelefteq \vec{t}_2$ if and only if $\mathcal{H}^{\vec{t}_1}_{\leq 0} \supseteq \mathcal{H}^{\vec{t}_2}_{\leq 0}$.*

**Proposition 4** (**Algebraic characterization of (minimal) continuity**). *A GISR $\bar{r}$ is continuous, if and only if*

$$\forall \vec{t} \in \mathcal{S}_{\vec{H}}, \text{ we have } \bar{r}(\vec{t}) \supseteq \bigcup_{\vec{t}' \in \mathcal{S}_{\vec{H}} : \vec{t}' \trianglelefteq \vec{t}} \bar{r}(\vec{t}')$$

*$\bar{r}$ is minimally continuous, if and only if*

$$\forall \vec{t} \in \mathcal{S}_{\vec{H}}, \text{ we have } \bar{r}(\vec{t}) = \bigcup_{\vec{t}' \in \mathcal{S}^{\circ}_{\vec{H}} : \vec{t}' \trianglelefteq \vec{t}} \bar{r}(\vec{t}'), \text{ and } (2) \, \forall \vec{t} \in \mathcal{S}^{\circ}_{\vec{H}}, \text{ we have } |\bar{r}(\vec{t})| = 1$$

The "continuity" part of Proposition 4 states that for any feasible signature $\vec{t}$ and its refinement $\vec{t}'$, we must have $\bar{r}(\vec{t}') \subseteq \bar{r}(\vec{t})$. The "minimal continuity" part states that any minimally continuous GISR is uniquely determined by its winners under atomic signatures (where a single winner is chosen for any atomic signature).

*Proof.* **The "if" part for continuity.** Suppose for the sake of contradiction that there exists $\vec{t} \in \mathcal{S}_{\vec{H}}$ such that $\bar{r}(\vec{t}) \supseteq \bigcup_{\vec{t}' \in \mathcal{S}^{\circ}_{\vec{H}} : \vec{t}' \trianglelefteq \vec{t}} \bar{r}(\vec{t}')$ but $\bar{r}$ is not continuous. This means that there exists $\vec{x} \in \mathbb{R}^{m!}$ with $\text{Sign}_{\vec{H}}(\vec{x}) = \vec{t}$, an infinite sequence $(\vec{x}_1, \vec{x}_2, \ldots)$ that converge to $\vec{x}$, and an alternative $a \notin \bar{r}(\vec{x})$, such that for every $j \in \mathbb{N}$, $a \in \bar{r}(\vec{x}_j)$. Because the total number of (feasible) signatures is finite, there exists an infinite subsequence of $(\vec{x}_1, \vec{x}_2, \ldots)$, denoted by $(\vec{x}'_1, \vec{x}'_2, \ldots)$, and $\vec{t}' \in \mathcal{S}_{\vec{H}}$ such that for all $j \in \mathbb{N}$ we have $\text{Sign}_{\vec{H}}(\vec{x}'_j) = \vec{t}'$. Note that $(\vec{x}'_1, \vec{x}'_2, \ldots)$ also converges to $\vec{x}$. Therefore, the following holds for every $k \leq K$.

- If $t'_k = 0$, then for every $j \in \mathbb{N}$ we have $\vec{h}_k \cdot \vec{x}_j = 0$, which means that $\vec{h}_k \cdot \vec{x} = 0$, i.e. $t_k = 0$.

- If $t'_k = +$, then for every $j \in \mathbb{N}$ we have $\vec{h}_k \cdot \vec{x}_j > 0$, which means that $\vec{h}_k \cdot \vec{x} \geq 0$, i.e. $t_k \in \{0, +\}$.

- Similarly, if $t'_k = -$, then for every $j \in \mathbb{N}$ we have $\vec{h}_k \cdot \vec{x}_j < 0$, which means that $\vec{h}_k \cdot \vec{x} \leq 0$, i.e. $t_k \in \{0, -\}$.

This means that $\vec{t'} \trianglelefteq \vec{t}$. Recall that we have assumed $\overline{r}(\vec{t}) \supseteq \bigcup_{\vec{t'} \in \mathcal{S}_{\vec{H}} : \vec{t'} \trianglelefteq \vec{t}} \overline{r}(\vec{t'})$, which means that $a \in \overline{r}(\vec{t'}) \subseteq \overline{r}(\vec{t}) = \overline{r}(\vec{x})$. This contradicts the assumption that $a \notin \overline{r}(\vec{x})$.

**The "only if" part for continuity.** Suppose for the sake of contradiction that $\overline{r}$ is continuous but there exists $\vec{t} \in \mathcal{S}_{\vec{H}}$ such that $\bigcup_{\vec{t'} \in \mathcal{S}_{\vec{H}} : \vec{t'} \trianglelefteq \vec{t}} \overline{r}(\vec{t'}) \not\subseteq \overline{r}(\vec{t})$. This means that there exist $\vec{t'} \triangleleft \vec{t}$ and an alternative $a$ such that $a \in \overline{r}(\vec{t'})$ but $a \notin \overline{r}(\vec{t})$. Because both $\vec{t}$ and $\vec{t'}$ are feasible, there exists $\vec{x}, \vec{x'} \in \mathbb{R}^{m!}$ such that $\mathrm{Sign}_{\vec{H}}(\vec{x}) = \vec{t}$ and $\mathrm{Sign}_{\vec{H}}(\vec{x'}) = \vec{t'}$. It is not hard to verify that the infinite sequence $(\vec{x} + \vec{x'}, \vec{x} + \frac{1}{2}\vec{x'}, \vec{x} + \frac{1}{3}\vec{x'}, \ldots)$ converge to $\vec{x}$, and for every $j \in \mathbb{N}$, $\mathrm{Sign}_{\vec{H}}(\vec{x} + \frac{1}{j}\vec{x'}) = \vec{t'}$, which means that $a \in \overline{r}(\vec{x} + \frac{1}{j}\vec{x'})$. By continuity of $\overline{r}$ we have $a \in \overline{r}(\vec{x}) = \overline{r}(\vec{t})$, which contradicts the assumption that $a \notin \overline{r}(\vec{t})$.

**The "if" part for minimal continuity.** To simplify the presentation, we formally define refinements of GISRs as follows.

**Definition 18** (**Refinements of GISRs**). *Let $\overline{r}^*$ and $\overline{r}$ be a pair of GISR such that for every $\vec{x} \in \mathbb{R}^{m!}$, $\overline{r}^*(\vec{x}) \subseteq \overline{r}(\vec{x})$. $\overline{r}^*$ is called a* refinement *of $\overline{r}$. If additionally there exists $\vec{x} \in \mathbb{R}^{m!}$ such that $\overline{r}^*(\vec{x}) \subset \overline{r}(\vec{x})$, then $\overline{r}^*$ is called a* strict refinement *of $\overline{r}$.*

Suppose for every $\vec{t} \in \mathcal{S}_{\vec{H}}$ we have $\overline{r}(\vec{t}) = \bigcup_{\vec{t'} \in \mathcal{S}^{\circ}_{\vec{H}} : \vec{t'} \trianglelefteq \vec{t}} \overline{r}(\vec{t'})$, and for every $\vec{t} \in \mathcal{S}^{\circ}_{\vec{H}}$ we have $|\overline{r}(\vec{t})| = 1$. By the "continuity" part proved above, $\overline{r}$ is continuous. To prove that $\overline{r}$ is minimally continuous, suppose for the sake of contradiction that $\overline{r}$ has a strict refinement, denoted by $\overline{r}^*$. Clearly for every atomic feasible signature $\vec{t} \in \mathcal{S}^{\circ}_{\vec{H}}$ we have $\overline{r}^*(\vec{t}) = \overline{r}(\vec{t})$. Therefore, by the "continuity" part proved above, for every feasible signature $\vec{t} \in \mathcal{S}_{\vec{H}}$, we have

$$\overline{r}^*(\vec{t}) \supseteq \bigcup_{\vec{t'} \in \mathcal{S}_{\vec{H}} : \vec{t'} \trianglelefteq \vec{t}} \overline{r}^*(\vec{t'}) \supseteq \bigcup_{\vec{t'} \in \mathcal{S}^{\circ}_{\vec{H}} : \vec{t'} \trianglelefteq \vec{t}} \overline{r}^*(\vec{t'}) = \bigcup_{\vec{t'} \in \mathcal{S}^{\circ}_{\vec{H}} : \vec{t'} \trianglelefteq \vec{t}} \overline{r}(\vec{t'}) = \overline{r}(\vec{t}),$$

which contradicts the assumption that $\overline{r}^*$ is a strict refinement of $\overline{r}$.

**The "only if" part for minimal continuity.** Suppose $\overline{r}$ is a minimally continuous GISR. We define another GISR $\overline{r}^*$ as follows.

- For every $\vec{t} \in \mathcal{S}^{\circ}_{\vec{H}}$ we let $\overline{r}^*(\vec{t}) \subseteq \overline{r}(\vec{t})$ and $|\overline{r}^*(\vec{t})| = 1$.

- For every $\vec{t} \in \mathcal{S}_{\vec{H}}$, we let $\overline{r}^*(\vec{t}) = \bigcup_{\vec{t'} \in \mathcal{S}^{\circ}_{\vec{H}} : \vec{t'} \trianglelefteq \vec{t}} \overline{r}^*(\vec{t'})$.

By the continuity part proved above, $\overline{r}^*$ is continuous. It is not hard to verify that $\overline{r}^*$ refines $\overline{r}$. Therefore, if either condition for minimal continuity does not hold, then $\overline{r}^*$ is a strict refinement of $\overline{r}$, which contradicts the minimality of $\overline{r}$.

This proves Proposition 4. $\qquad\qquad\square$

Next, we prove some properties about $\mathcal{H}^{\vec{t}}$ that will be frequently used in the proofs of this paper. The proposition has three parts. Part (i) characterizes profiles $P$ whose histogram is in $\mathcal{H}^{\vec{t}}$; part (ii) characterizes vectors in $\mathcal{H}^{\vec{t}}_{\leq 0}$; and part (iii) states that for every atomic signature $\vec{t}$, $\mathcal{H}^{\vec{t}}_{\leq 0}$ is a full dimensional cone in $\mathbb{R}^{m!}$.

**Claim 1** (**Properties of $\mathcal{H}^{\vec{t}}$**). *Given integer $\vec{H}$, any $\vec{t} \in \mathcal{S}_{\vec{H}}$,*

*(i)* *for any integral profile $P$, $Hist(P) \in \mathcal{H}^{\vec{t}}$ if and only if $Sign_{\vec{H}}(Hist(P)) = \vec{t}$;*

*(ii)* *for any $\vec{x} \in \mathbb{R}^{m!}$, $\vec{x} \in \mathcal{H}^{\vec{t}}_{\leq 0}$ if and only if $\vec{t} \trianglelefteq Sign_{\vec{H}}(\vec{x})$;*

*(iii)* *if $\vec{t} \in \mathcal{S}^{\circ}_{\vec{H}}$ then $\dim(\mathcal{H}^{\vec{t}}_{\leq 0}) = m!$.*

*Proof.* Part (i) follows after the definition. More precisely, $\mathrm{Sign}_{\vec{H}}(\mathrm{Hist}(P)) = \vec{t}$ if and only if for every $k \leq K$, (1) $t_k = +$ if and only if $\vec{h}_k \cdot \mathrm{Hist}(P) > 0$, which is equivalent to $-\vec{h}_k \cdot \mathrm{Hist}(P) \leq -1$ because $\vec{h}_k \in \mathbb{Z}^{m!}$; (2) likewise, $t_k = -$ if and only if $\vec{h}_k \cdot \mathrm{Hist}(P) \leq -1$, and (3) if $t_k = 0$ if and only if $\vec{h}_k \cdot \mathrm{Hist}(P) \leq 0$ and $-\vec{h}_k \cdot \mathrm{Hist}(P) \leq 0$. This proves Part (i).

Part (ii) also follows after the definition. More precisely, $\vec{x} \in \mathcal{H}^{\vec{t}}_{\leq 0}$ if and only if for every $k \leq K$, (1) $t_k = +$ if and only if $-\vec{h}_k \cdot \vec{x} \leq 0$, which is equivalent to $[\mathrm{Sign}_{\vec{H}}(\vec{x})]_k \in \{0, +\}$; (2) likewise, $t_k = -$ if and only if $\vec{h}_k \cdot \vec{x} \leq 0$, which is equivalent to $[\mathrm{Sign}_{\vec{H}}(\vec{x})]_k \in \{0, -\}$, and (3) if $t_k = 0$ if and only if $\vec{h}_k \cdot \vec{x} \leq 0$ and $-\vec{h}_k \cdot \vec{x} \leq 0$, which is equivalent to $[\mathrm{Sign}_{\vec{H}}(\vec{x})]_k = 0$. This is equivalent to $\vec{t} \trianglelefteq \mathrm{Sign}_{\vec{H}}(\vec{x})$.

We now prove Part (iii). Suppose $\vec{t} \in \mathcal{S}^{\circ}_{\vec{H}}$. Let $\vec{x} \in \mathcal{H}^{\vec{t}} \cap \mathbb{R}^{m!}_{\geq 0}$ denote an arbitrary non-negative vector whose existence is guaranteed by the assumption that $\vec{t} \in \mathcal{S}^{\circ}_{\vec{H}}$. Therefore, for every $k \leq K$, either $\vec{h}_k \cdot \vec{x} \leq -1$ or $-\vec{h}_k \cdot \vec{x} \leq -1$, which means that there exists $\delta > 0$ such that any $\vec{x}'$ with $|\vec{x}' - \vec{x}|_{\infty} < \delta$, we have $\vec{h}_k \cdot \vec{x} < 0$ or $-\vec{h}_k \cdot \vec{x} < 0$. This means that $\vec{x}$ is an interior point of $\mathcal{H}^{\vec{t}}_{\leq 0}$ in $\mathbb{R}^{m!}$, which implies that $\dim(\mathcal{H}^{\vec{t}}_{\leq 0}) = m!$. $\qquad\square$

# E    Materials for Section 3: Semi-random CONDORCET CRITERION

## E.1    Lemma 2 and Its Proof

For any GISR $\bar{r}$, we first define $\mathcal{R}^{\bar{r}}_{\mathrm{CWW}}$ (respectively, $\mathcal{R}^{\bar{r}}_{\mathrm{CWL}}$) that corresponds to fractional profiles where a Condorcet winner exists and is a co-winner (respectively, not a co-winner) under $\bar{r}$. CWW (respectively, CWL) stands for "Condorcet winner wins" (respectively, "Condorcet winner loses").

$$\mathcal{R}^{\bar{r}}_{\mathrm{CWW}} = \{\vec{x} \in \mathbb{R}^{m!} : \mathrm{CW}(\vec{x}) \cap \bar{r}(\vec{x}) \neq \emptyset\}$$
$$\mathcal{R}^{\bar{r}}_{\mathrm{CWL}} = \{\vec{x} \in \mathbb{R}^{m!} : \mathrm{CW}(\vec{x}) \cap (\mathcal{A} \setminus \bar{r}(\vec{x})) \neq \emptyset\}$$

For any set $\mathcal{R} \subseteq \mathbb{R}^{m!}$, let $\mathrm{Closure}(\mathcal{R})$ denote the *closure* of $\mathcal{R}$ in $\mathbb{R}^{m!}$, that is, all points in $\mathcal{R}$ and their limiting points. Next, we introduce four conditions to present Lemma 2 below.

**Definition 19.** *Given a GISR $\bar{r}$ and $n \in \mathbb{N}$, we define the following conditions, where $\vec{x} \in \mathbb{R}^{m!}$.*

- **Always satisfaction:** $\mathbf{C_{AS}}(\bar{r}, n)$ *holds if and only if for all $P \in \mathcal{L}(\mathcal{A})^n$, $\mathrm{CC}(\bar{r}, P) = 1$.*

- **Robust satisfaction:** $\mathbf{C_{RS}}(\bar{r}, \vec{x})$ *holds if and only if $\vec{x} \notin \mathrm{Closure}(\mathcal{R}^{\bar{r}}_{CWL})$.*

- **Robust dissatisfaction:** $\mathbf{C_{RD}}(\bar{r}, \vec{x})$ *holds if and only if $CW(\vec{x}) \cap (\mathcal{A} \setminus \bar{r}(\vec{x})) \neq \emptyset$.*

- **Non-Robust satisfaction:** $\mathbf{C_{NRS}}(\bar{r}, \vec{x})$ *holds if and only if $ACW(\vec{x}) \neq \emptyset$ and $\vec{x} \notin \mathrm{Closure}(\mathcal{R}^{\bar{r}}_{CWW})$.*

In words, $\mathrm{C_{AS}}(\bar{r}, n)$ means that $\bar{r}$ always satisfies CC for $n$ agents. Robust satisfaction $\mathrm{C_{RS}}(\bar{r}, \vec{x})$ states that $\vec{x}$ is away from the dissatisfaction instances (i.e., $\mathcal{R}^{\bar{r}}_{\mathrm{CWL}}$) by a constant margin. Robust dissatisfaction $\mathrm{C_{RD}}(\bar{r}, \vec{x})$ states that the Condorcet winner exists under $\vec{x}$ and is not a co-winner under $\bar{r}$. Robust satisfaction and robust dissatisfaction are not "symmetric", because there are two sources of satisfaction: (1) no Condorcet winner exists and (2) the Condorcet winner exists and is also a winner, while there is only one source of dissatisfaction: the Condorcet winner exists but is not a winner.

The intuition behind Non-Robust satisfaction $\mathrm{C_{NRS}}(\bar{r}, \vec{x})$ may not be immediately clear by definition. It is called "satisfaction", because $\mathrm{ACW}(\vec{x}) \neq \emptyset$ implies that $\mathrm{CW}(\vec{x}) = \emptyset$, which means that $\bar{r}$

satisfies CC at $\vec{x}$. The reason behind "non-robust" is that when a small perturbation $\vec{x}'$ is introduced, $UMG(\vec{x} + \vec{x}')$ often contains a Condorcet winner that is not a co-winner under $\vec{x}$, because $\vec{x}$ is constantly far away from $\mathcal{R}^{\bar{r}}_{\text{CWW}}$.

**Example 9** (**The four conditions in Definition 19**). *Let $m = 3$ and $n = 14$. Table 3 illustrates four distributions, their UMG, the irresolute plurality winners, and their (dis)satisfaction of the four conditions introduced defined in Definition 19. $\pi^1, \pi^2$, and $\pi'$ are the same as in Example 1 and 3. Notice that $\pi'$ is a linear combination of $\pi^1$ and $\pi^2$.*

| | 123 | 132 | 231 | 321 | 213 | 312 | UMG | $\overline{Plu}$ winner(s) | $C_{AS}$ | $C_{RS}$ | $C_{RD}$ | $C_{NRS}$ |
|---|---|---|---|---|---|---|---|---|---|---|---|---|
| $\pi^1$ | $\frac{1}{4}$ | $\frac{1}{4}$ | $\frac{1}{8}$ | $\frac{1}{8}$ | $\frac{1}{8}$ | $\frac{1}{8}$ | (UMG: $2 \to 1 \leftarrow 3$) | $\{1\}$ | N | N | N | Y |
| $\pi^2$ | $\frac{1}{8}$ | $\frac{1}{8}$ | $\frac{3}{8}$ | $\frac{1}{8}$ | $\frac{1}{8}$ | $\frac{1}{8}$ | (UMG: $2 \to 1$, $2 \to 3$) | $\{2\}$ | N | Y | N | N |
| $\pi_{\text{uni}}$ | $\frac{1}{6}$ | $\frac{1}{6}$ | $\frac{1}{6}$ | $\frac{1}{6}$ | $\frac{1}{6}$ | $\frac{1}{6}$ | (UMG: no edges) | $\{1, 2, 3\}$ | N | N | N | N |
| $\frac{3\pi^1+\pi^2}{4}$ | $\frac{7}{32}$ | $\frac{7}{32}$ | $\frac{3}{16}$ | $\frac{1}{8}$ | $\frac{1}{8}$ | $\frac{1}{8}$ | (UMG: $2 \to 1$, $2 \to 3$) | $\{1\}$ | N | N | Y | N |

Table 3: Distributions and their (dis)satisfaction of conditions in Definition 19.

Let $P_{14}$ denote the $14$-profile $\{6 \times [1 \succ 2 \succ 3], 4 \times [2 \succ 3 \succ 1], 4 \times [2 \succ 1 \succ 3]\}$. It is not hard to verify that alternative $2$ is the Condorcet winner under $P_{14}$ and $\overline{Plu}(P_{14}) = \{1\}$. Therefore, $C_{AS}(\overline{Plu}, 14) = N$.

- **$\pi^1$**. $C_{RS}(\overline{Plu}, \pi^1) = N$. To see this, let $\vec{x}'$ denote the vector that corresponds to the single-vote profile $\{2 \succ 3 \succ 1\}$. For any sufficiently small $\delta > 0$, $\pi^1 + \delta\vec{x}' \in \mathcal{R}^{\overline{Plu}}_{CWL}$, because $2$ is the Condorcet winner and $1$ is the unique plurality winner. $C_{RD}(\overline{Plu}, \pi^1) = N$ because $CW(\pi^1) = \emptyset$. $C_{NRS}(\overline{Plu}, \pi^1) = Y$ because $ACW(\pi^1) = \{2, 3\}$, and for any $\vec{x}' \in \mathbb{R}^6$ and any $\delta > 0$ that is sufficiently small, in $\pi^1 + \delta\vec{x}'$ we have that $2$ or $3$ is Condorcet winner and $1$ is the unique plurality winner, which means that $\pi^1 + \delta\vec{x}' \notin \mathcal{R}^{\bar{r}}_{CWW}$.

- **$\pi^2$**. $C_{RS}(\overline{Plu}, \pi^2) = Y$ because the plurality score of $2$ is strictly higher than the plurality score of any other alternative, which means that for any $\vec{x}' \in \mathbb{R}^{m!}$, for any $\delta > 0$ that is sufficiently small, $2$ is the Condorcet winner as well as the unique plurality winner in $\pi^2 + \delta\vec{x}'$. This means that $\pi^2$ is not in the closure of vectors where CC is violated. $C_{RD}(\overline{Plu}, \pi^2) = N$ because $CW(\pi^2) \cap (\mathcal{A} \setminus \overline{Plu}(\pi^2)) = \{2\} \cap \{1, 3\} = \emptyset$. $C_{NRS}(\overline{Plu}, \pi^2) = N$ because $ACW(\pi^2) = \emptyset$.

- **$\pi_{uni}$**. $C_{RS}(\overline{Plu}, \pi_{uni}) = N$. To see this, let $\vec{x}'$ denote the vector that corresponds to the $14$-profile $P_{14}$ defined earlier in this example to prove $C_{AS}(\overline{Plu}, 14) = N$. For any $\delta > 0$ that is sufficiently small, we have $\pi_{uni} + \delta\vec{x}' \in \mathcal{R}^{\overline{Plu}}_{CWL}$, because $2$ is the Condorcet winner and $1$ is the unique plurality winner. $C_{RD}(\overline{Plu}, \pi_{uni}) = N$ because $CW(\pi_{uni}) = \emptyset$. $C_{NRS}(\overline{Plu}, \pi_{uni}) = N$ because $ACW(\pi_{uni}) = \emptyset$.

- **$\frac{3\pi^1+\pi^2}{4}$**. Let $\pi' = \frac{3\pi^1+\pi^2}{4}$. $C_{RS}(\overline{Plu}, \pi') = N$ because $\pi' \in \mathcal{R}^{\overline{Plu}}_{CWL}$. $C_{RD}(\overline{Plu}, \pi') = Y$ because $CW(\pi') \cap (\mathcal{A} \setminus \overline{Plu}(\pi')) = \{2\} \cap \{2, 3\} \neq \emptyset$. $C_{NRS}(\overline{Plu}, \pi') = N$ because $ACW(\pi') = \emptyset$.

For any condition $Y$, we use $\neg Y$ to indicate that $Y$ does not hold. For example, $\neg C_{AS}(\bar{r}, n)$ means that $C_{AS}(\bar{r}, n)$ does not hold, i.e., there exists $P \in \mathcal{L}(\mathcal{A})^n$ with $CC(\bar{r}, P) = 0$. A GISR rule $r_1$ is

a *refinement* of another voting rule $r_2$, if for all $\vec{x} \in \mathbb{R}^{m!}$, we have $r_1(\vec{x}) \subseteq r_2(\vec{x})$. We note that while the four conditions in Definition 19 are not mutually exclusive by definition, they provide a complete characterization of semi-random CC under any refinement of any minimally continuous int-GISR as shown in the lemma below.

**Lemma 2** (**Semi-random** CC: **Minimally Continuous Int-GISRs**). *For any fixed $m \geq 3$, let $\mathcal{M} = (\Theta, \mathcal{L}(\mathcal{A}), \Pi)$ be a strictly positive and closed single-agent preference model, let $\overline{r}$ be a minimally continuous int-GISR and let $r$ be a refinement of $\overline{r}$. For any $n \in \mathbb{N}$ with $2 \mid n$, we have*

$$\widetilde{\mathrm{CC}}_{\Pi}^{\min}(r, n) = \begin{cases} 1 & \textit{if } C_{AS}(\overline{r}, n) \\ 1 - \exp(-\Theta(n)) & \textit{if } \neg C_{AS}(\overline{r}, n) \textit{ and } \forall \pi \in CH(\Pi), C_{RS}(\overline{r}, \pi) \\ \Theta(n^{-0.5}) & \textit{if } \begin{cases} \textit{(1) } \forall \pi \in CH(\Pi), \neg C_{RD}(\overline{r}, \pi) \textit{ and} \\ \textit{(2) } \exists \pi \in CH(\Pi) \textit{ s.t. } C_{NRS}(\overline{r}, \pi) \end{cases} \\ \exp(-\Theta(n)) & \textit{if } \exists \pi \in CH(\Pi) \textit{ s.t. } C_{RD}(\overline{r}, \pi) \\ \Theta(1) \wedge (1 - \Theta(1)) & \textit{otherwise} \end{cases}$$

*For any $n \in \mathbb{N}$ with $2 \nmid n$, we have*

$$\widetilde{\mathrm{CC}}_{\Pi}^{\min}(r, n) = \begin{cases} 1 & \textit{same as the } 2 \mid n \textit{ case} \\ 1 - \exp(-\Theta(n)) & \textit{same as the } 2 \mid n \textit{ case} \\ \exp(-\Theta(n)) & \textit{if } \exists \pi \in CH(\Pi) \textit{ s.t. } C_{RD}(\overline{r}, \pi) \textit{ or } C_{NRS}(\overline{r}, \pi) \\ \Theta(1) \wedge (1 - \Theta(1)) & \textit{otherwise} \end{cases}$$

Lemma 2 can be applied to a wide range of resolute voting rules because it works for any refinement $r$ (i.e., using any tie-breaking mechanism) of any minimally continuous GISR (which include all voting rules discussed in this paper). Notice that $r$ is not required to be a GISR, the L case and the 0 case never happen, and the conditions of all cases depend on $\overline{r}$ but not $r$.

**Example 10** (**Applications of Lemma 2 to plurality**). *Continuing the setting of Example 9, we let Plu denote any refinement of $\overline{Plu}$. We first apply the $2 \mid n$ part of Lemma 2 to the following four cases of $\Pi$ for sufficiently large $n$ using Table 3. The first three cases correspond to i.i.d. distributions, i.e., $|\Pi| = 1$. In particular, $\Pi = \{\pi_{uni}\}$ corresponds to IC.*

- *$\Pi = \{\pi^1, \pi^2\}$. We have $\widetilde{\mathrm{CC}}_{\Pi}^{\min}(Plu, n) = \exp(-\Theta(n))$, that is, the VU case holds. This is because let $\pi' = \frac{3\pi^1 + \pi^2}{4}$, we have $\pi' \in CH(\Pi)$ and $C_{RS}(\overline{Plu}, \pi') = N$ according to Table 3.*

- *$\Pi_1 = \{\pi^1\}$. We have $\widetilde{\mathrm{CC}}_{\Pi_1}^{\min}(Plu, n) = \Theta(n^{-0.5})$, that is, the U case holds.*

- *$\Pi_2 = \{\pi^2\}$. We have $\widetilde{\mathrm{CC}}_{\Pi_2}^{\min}(Plu, n) = 1 - \exp(-\Theta(n))$, that is, the VL case holds.*

- *$\Pi_{IC} = \{\pi_{uni}\}$. We have $\widetilde{\mathrm{CC}}_{\Pi_{IC}}^{\min}(Plu, n) = \Theta(1) \wedge (1 - \Theta(1))$, that is, the M case holds.*

*When $2 \nmid n$ and $\Pi_1 = \{\pi^1\}$, we have $\widetilde{\mathrm{CC}}_{\Pi_1}^{\min}(Plu, n) = \exp(-\Theta(n))$, that is, the VU case holds.*

**Intuitive explanations.** The conditions in Lemma 2 can be explained as follows. Take the $2 \mid n$ case for example. In light of various multivariate central limit theorems, the histogram of the randomly-generated profile when the adversary chooses $\vec{\pi} = (\pi_1, \ldots, \pi_n)$ is concentrated in a $\Theta(n^{-0.5})$ neighborhood of $\sum_{j=1}^{n} \pi_j$, denoted by $B_{\vec{\pi}}$. Let $\mathrm{avg}(\vec{\pi}) = \frac{1}{n} \sum_{j=1}^{n} \pi_j$, which means that $\mathrm{avg}(\vec{\pi}) \in CH(\Pi)$. The condition for the 1 case is straightforward. Suppose the 1 case does not happen, then the VL case happens if all distributions in $CH(\Pi)$, which includes $\mathrm{avg}(\vec{\pi})$, are far from instances of dissatisfaction, so that no instance of dissatisfaction is in $B_{\vec{\pi}}$. Suppose the VL case does not happen. The U case happens if the min-adversary can find a non-robust satisfaction instance ($C_{NRS}(\overline{r}, \pi)$) but cannot find a robust dissatisfaction instance ($\neg C_{RD}(\overline{r}, \pi)$). And if the min-adversary can find a robust dissatisfaction instance ($C_{RD}(\overline{r}, \pi)$), then $B_{\vec{\pi}}$ does not contain any instance of satisfaction, which means that the VU case happens. All remaining cases are M cases.

**Odd vs. even $n$.** The $2 \nmid n$ case also admits a similar explanation. The main difference is that when $2 \nmid n$, the UMG of any $n$-profile must be a complete graph, i.e., no alternatives are tied in the UMG. Therefore, when $C_{NRS}(\overline{r}, \pi)$ is satisfied, a Condorcet winner (who is one of the two ACWs

in $\pi$) must exist and constitutes an instance of robust dissatisfaction when $2 \nmid n$. On the other hand, it is possible that the two ACWs in $\pi$ are tied in an $n$-profile when $2 \mid n$, which constitutes a case where CC is satisfied because the Condorcet winner does not exist. This happens with probability $\Theta(n^{-0.5})$. This difference leads to the $\Theta(n^{-0.5})$ case when $2 \mid n$, and it becomes part of the $\exp(-\Theta(n))$ case when $2 \nmid n$ .

**Proof sketch.** Before presenting the formal proof in the following subsection, we present a proof sketch here.

We first prove the special case $r = \overline{r}$, which is done by applying Lemma 1 in the following three steps. **Step 1.** Define $\mathcal{C}$ that characterizes the satisfaction of CC under $\overline{r}$, and an almost complement $\mathcal{C}^*$ of $\mathcal{C}$. In fact, we will let $\mathcal{C} = \mathcal{C}_{\text{NCW}} \cup \mathcal{C}_{\text{CWW}}$ as in Section 4 and Section C.1, and prove that one choice of $\mathcal{C}^*$ is the union of polyhedra that represent profiles where the Condorcet winner exists but is not an $\overline{r}$ co-winner. **Step 2.** Characterize $\alpha_n^*$ and $\beta_n$, which is technically the most involved part due to the generality of the theorem. **Step 3.** Formally apply Lemma 1.

Then, let $r$ denote an arbitrary refinement of $\overline{r}$. We define a slightly different version of CC, denoted by $\text{CC}^*$, whose satisfaction under $\overline{r}$ will be used as a lower bound on the satisfaction of CC under $r$. For any GISR $\overline{r}$ and any profile $P$, we define

$$\text{CC}^*(\overline{r}, P) = \left\{ \begin{array}{ll} 1 & \text{if } \text{CW}(P) = \emptyset \text{ or } \text{CW}(P) = \overline{r}(P) \\ 0 & \text{otherwise} \end{array} \right.$$

Compared to CC, $\text{CC}^*$ rules out profiles $P$ where a Condorcet winner exists and is not the unique winner under $\overline{r}$. Therefore, for any $\vec{\pi} \in \Pi^n$, we have

$$\Pr_{P \sim \vec{\pi}}(\text{CC}^*(\overline{r}, P) = 1) \leq \Pr_{P \sim \vec{\pi}}(\text{CC}(r, P) = 1) \leq \Pr_{P \sim \vec{\pi}}(\text{CC}(\overline{r}, P) = 1)$$

Then, we prove that semi-random $\text{CC}^*$, i.e., $\widetilde{\text{CC}^*}_\Pi^{\min}(\overline{r}, n)$, asymptotically matches $\widetilde{\text{CC}}_\Pi^{\min}(\overline{r}, n)$, which concludes the proof of Lemma 2.

### E.1.1 Proof of Lemma 2

*Proof.* The 1 cases of the theorem is trivial. **In the rest of the proof, we assume that the 1 case does not hold.** That is, there exists an $n$-profile $P$ such that $\text{CW}(P)$ exists but is not in $\overline{r}(P)$. We will prove that the theorem holds for any $n > N_{\overline{r}}$, where $N_{\overline{r}} \in \mathbb{N}$ is a constant that only depends on $\overline{r}$ that will be defined later (in Definition 24). This is without loss of generality, because when $n$ is bounded above by a constant, the 1 case belongs to the U case (i.e., $\Theta(n^{-0.5})$) and the VU case (i.e., $\exp(-\Theta(n))$).

Let $\overline{r}$ be defined by $\vec{H}$ and $g$. We first prove the theorem for the special case where $r = \overline{r}$, and then show how to modify the proof for general $r$. For any irresolute voting rule $\overline{r}$, we recall that $\text{CC}(\overline{r}, P) = 1$ if and only if either $P$ does not have a Condorcet winner, or the Condorcet winner is a co-winner under $\overline{r}$.

**Proof for the special case $r = \overline{r}$.** Recall that in this case $\overline{r}$ is a minimally continuous GISR. In light of Lemma 1, the proof proceeds in the following three steps. **Step 1.** Define $\mathcal{C}$ that characterizes the satisfaction of CONDORCET CRITERION of $\overline{r}$ and an almost complement $\mathcal{C}^*$ of $\mathcal{C}$. **Step 2.** Characterize $\Pi_{\mathcal{C},n}$, $\Pi_{\mathcal{C}^*,n}$, $\beta_n$, and $\alpha_n^*$. **Step 3.** Apply Lemma 1.

**Step 1: Define $\mathcal{C}$ and $\mathcal{C}^*$.** The definition is similar to the ones presented in Section 4 for plurality. We will define $\mathcal{C} = \mathcal{C}_{\text{NCW}} \cup \mathcal{C}_{\text{CWW}}$, where $\mathcal{C}_{\text{NCW}}$ represents the histograms of profiles that do not have a Condorcet winner, and $\mathcal{C}_{\text{CWW}}$ represents histograms of profiles where a Condorcet winner exists and is a co-winner under $\overline{r}$. $\mathcal{C}_{\text{NCW}}$ is similar to the set defined in [54, Proposition 5 in the Appendix]. For completeness we recall its definition using the notation of this paper.

Recall that $\text{Pair}_{a,b}$ is the pairwise difference vector defined in Definition 10. It follows that for any profile $P$ and any pair of alternatives $a, b$, $\text{Pair}_{a,b} \cdot \text{Hist}(P) > 0$ if and only if there is an edge $a \rightarrow b$ in $\text{UMG}(P)$; $\text{Pair}_{a,b} \cdot \text{Hist}(P) = 0$ if and only if $a$ and $b$ are tied in $\text{UMG}(P)$. Then, we use $\text{Pair}_{a,b}$'s to define polyhedra that characterize histograms of profiles whose UMGs equal to a given graph $G$.

**Definition 20** ($\mathcal{H}^G$). *Given an unweighted directed graph $G$ over $\mathcal{A}$, let $\mathbf{A}^G = \begin{bmatrix} \mathbf{A}^G_{edge} \\ \mathbf{A}^G_{tie} \end{bmatrix}$, where $\mathbf{A}^G_{edge}$ consists of rows $Pair_{b,a}$ for all edges $a \to b \in G$, and $\mathbf{A}^G_{edge}$ consists of two rows $Pair_{b,a}$ and $Pair_{a,b}$ for each tie $\{a,b\}$ in $G$. Let $\vec{\mathbf{b}}^G = [\underbrace{-\vec{1}}_{for\ \mathbf{A}^G_{edge}}, \underbrace{\vec{0}}_{for\ \mathbf{A}^G_{tie}}]$ and*

$$\mathcal{H}^G = \left\{ \vec{x} \in \mathbb{R}^{m!} : \mathbf{A}^G \cdot (\vec{x})^\top \leq \left( \vec{\mathbf{b}}^G \right)^\top \right\}$$

Next, we define polyhedra indexed by an alternative $a$ and a feasible signature $\vec{t} \in \mathcal{S}_{\vec{H}}$ that characterize the histograms of profiles $P$ where $a$ is the Condorcet winner and $\text{Sign}_{\vec{H}}(P) = \vec{t}$.

**Definition 21** ($\mathcal{H}^{a,\vec{t}}$). *Given $\vec{H} = (\vec{h}_1, \ldots, \vec{h}_K) \in (\mathbb{R}^d)^K$, $a \in \mathcal{A}$, and $\vec{t} \in \mathcal{S}_{\vec{H}}$, we let $\mathbf{A}^{a,\vec{t}} = \begin{bmatrix} \mathbf{A}^{CW=a} \\ \mathbf{A}^{\vec{t}} \end{bmatrix}$, where $\mathbf{A}^{CW=a}$ consists of pairwise difference vectors $Pair_{b,a}$ for each alternative $b \neq a$, and $\mathbf{A}^{\vec{t}}$ is the matrix used to define $\mathcal{H}^{\vec{t}}$ in Definition 17. Let $\vec{\mathbf{b}}^{a,\vec{t}} = [\underbrace{-\vec{1}}_{for\ \mathbf{A}^{CW=a}}, \underbrace{\vec{\mathbf{b}}^{\vec{t}}}_{for\ \mathbf{A}^{\vec{t}}}]$ and*

$$\mathcal{H}^{a,\vec{t}} = \{ \vec{x} \in \mathbb{R}^{m!} : \mathbf{A}^{a,\vec{t}} \cdot (\vec{x})^\top \leq \left( \vec{\mathbf{b}}^{a,\vec{t}} \right)^\top \}$$

Next, we use $\mathcal{H}^G$ and $\mathcal{H}^{a,\vec{t}}$ as building blocks to define $\mathcal{C} = \mathcal{C}_{\text{NCW}} \cup \mathcal{C}_{\text{CWW}}$ and an almost complement of $\mathcal{C}$, denoted by $\mathcal{C}_{\text{CWL}}$. At a high level, $\mathcal{C}_{\text{NCW}}$ corresponds to the profiles where no Condorcet winner exists (NCW represents "no Condorcet winner"), $\mathcal{C}_{\text{CWW}}$ corresponds to profiles where the Condorcet winner exists and is also an $\bar{r}$ co-winner (CWW represents "Condorcet winner wins"), and $\mathcal{C}_{\text{CWL}}$ corresponds to profiles where the Condorcet winner exists and is not an $\bar{r}$ co-winner (CWL represents "Condorcet winner loses").

**Definition 22** ($\mathcal{C}$ and $\mathcal{C}_{\text{CWL}}$). *Given an int-GISR characterized by $\vec{H}$ and $g$, we define*

$$\mathcal{C} = \mathcal{C}_{NCW} \cup \mathcal{C}_{CWW}, \quad \text{where } \mathcal{C}_{NCW} = \bigcup\nolimits_{G:CW(G)=\emptyset} \mathcal{H}^G \text{ and } \mathcal{C}_{CWW} = \bigcup\nolimits_{a \in \mathcal{A}, \vec{t} \in \mathcal{S}_{\vec{H}} : a \in \bar{r}(\vec{t})} \mathcal{H}^{a,\vec{t}}$$

$$\mathcal{C}_{CWL} = \bigcup\nolimits_{a \in \mathcal{A}, \vec{t} \in \mathcal{S}_{\vec{H}} : a \notin \bar{r}(\vec{t})} \mathcal{H}^{a,\vec{t}}$$

We note that some $\mathcal{H}^{a,\vec{t}}$ can be empty. To see that $\mathcal{C}_{\text{CWL}}$ is indeed an almost complement of $\mathcal{C} = \mathcal{C}_{\text{NCW}} \cup \mathcal{C}_{\text{CWW}}$, we note that $\mathcal{C} \cap \mathcal{C}_{\text{CWL}} = \emptyset$, and for any integer vector $\vec{x}$,

- if $\vec{x}$ does not have a Condorcet winner then $\vec{x} \in \mathcal{C}_{\text{NCW}} \subseteq \mathcal{C}$;

- if $\vec{x}$ has a Condorcet winner $a$, which is also an $\bar{r}$ co-winner, then $\vec{x} \in \mathcal{H}^{a,\text{Sign}_{\vec{H}}(\vec{x})} \subseteq \mathcal{C}_{\text{CWW}} \subseteq \mathcal{C}$;

- otherwise $\vec{x}$ has a Condorcet winner $a$, which is not an $\bar{r}$ co-winner. Then $\vec{x} \in \mathcal{H}^{a,\text{Sign}_{\vec{H}}(\vec{x})} \subseteq \mathcal{C}_{\text{CWL}}$.

Therefore, $\mathbb{Z}^q \subseteq \mathcal{C} \cup \mathcal{C}_{\text{CWL}}$.

**Step 2: Characterize $\Pi_{\mathcal{C},n}$, $\Pi_{\mathcal{C}_{\text{CWL}},n}$, $\beta_n$, and $\alpha_n^*$.** Recall that $\beta_n$ and $\alpha_n^*$ are defined by $\dim_{\mathcal{C},n}^{\max}(\pi)$ and $\dim_{\mathcal{C}_{\text{CWL}},n}^{\max}(\pi)$ for $\pi \in \text{CH}(\Pi)$ as follows:

$$\beta_n = \min\nolimits_{\pi \in \text{CH}(\Pi)} \dim_{\mathcal{C},n}^{\max}(\pi) = \min\nolimits_{\pi \in \text{CH}(\Pi)} \max \left( \dim_{\mathcal{C}_{\text{NCW}},n}^{\max}(\pi), \dim_{\mathcal{C}_{\text{CWW}},n}^{\max}(\pi) \right)$$

$$\alpha_n^* = \max\nolimits_{\pi \in \text{CH}(\Pi)} \dim_{\mathcal{C}_{\text{CWL}},n}^{\max}(\pi)$$

For convenience, we let $\Pi_{\mathcal{C},n}$ denote the distributions in $\text{CH}(\Pi)$, each of which is connected to an edge with positive weight in the activation graph (Definition 6). Formally, we have the following definition.

**Definition 23** ($\Pi_{\mathcal{C},n}$). *Given a set of distributions $\Pi$ over $q$, $\mathcal{C} = \bigcup_{i \leq I} \mathcal{H}_i$, and $n \in \mathbb{N}$, let*

$$\Pi_{\mathcal{C},n} = \{\pi \in CH(\Pi) : \exists i \leq I \text{ s.t. } \mathcal{H}_{i,n}^{\mathbb{Z}} \neq \emptyset \text{ and } \pi \in \mathcal{H}_{i,\leq 0}\}$$

Table 4 gives an overview of the rest of the proof in Step 2, which characterizes $\dim_{\mathcal{C},n}^{\max}(\pi)$ and $\dim_{\mathcal{C}_{\text{CWL}},n}^{\max}(\pi)$ by the membership of $\pi \in CH(\Pi)$ in $\Pi_{\mathcal{C}_{\text{NCW}},n}, \Pi_{\mathcal{C}_{\text{CWW}},n}$, and $\Pi_{\mathcal{C}_{\text{CWL}},n}$, respectively, where $n \geq N_{\overline{r}}$ for a constant $N_{\overline{r}}$ that will be defined momentarily (in Definition 24).

| | | | | | | |
|---|---|---|---|---|---|---|
| $\pi \in \Pi_{\mathcal{C}_{\text{NCW}},n}$ | $*$ | $*$ | N | Y | Y | N |
| $\pi \in \Pi_{\mathcal{C}_{\text{CWW}},n}$ | Y | Y | N | N | N | N |
| $\pi \in \Pi_{\mathcal{C}_{\text{CWL}},n}$ | Y | N | Y | Y | N | N |
| $\dim_{\mathcal{C}_{\text{NCW}},n}^{\max}(\pi)$ (Claim 3) | $*$ | $*$ | $-\frac{n}{\log n}$ | $m!$ or $m!-1$ | $m!$ | |
| $\dim_{\mathcal{C}_{\text{CWW}},n}^{\max}(\pi)$ (Claim 6) | $m!$ | $m!$ | $\leq -\frac{n}{\log n}$ | $< 0$ | $< 0$ | N/A |
| $\dim_{\mathcal{C},n}^{\max}(\pi) =$ $\max\left(\dim_{\mathcal{C}_{\text{NCW}},n}^{\max}(\pi), \dim_{\mathcal{C}_{\text{CWW}},n}^{\max}(\pi)\right)$ | $m!$ | $m!$ | $-\frac{n}{\log n}$ | $\dim_{\mathcal{C}_{\text{NCW}},n}^{\max}(\pi)$ | $m!$ | |
| $\dim_{\mathcal{C}_{\text{CWL}},n}^{\max}(\pi)$ (Claim 6) | $m!$ | $-\frac{n}{\log n}$ | $m!$ | $m!$ | $-\frac{n}{\log n}$ | |

Table 4: $\dim_{\mathcal{C},n}^{\max}(\pi)$ and $\dim_{\mathcal{C}_{\text{CWL}},n}^{\max}(\pi)$ for CC for $\pi \in CH(\Pi)$ and sufficiently large $n$.

We will first specify $N_{\overline{r}}$ in Step 2.1. Then in Step 2.2, we will characterize $\Pi_{\mathcal{C}_{\text{NCW}},n}$ and $\dim_{\mathcal{C}_{\text{NCW}},n}^{\max}(\pi)$ in Claim 3, and characterize $\Pi_{\mathcal{C}_{\text{CWW}},n}, \dim_{\mathcal{C}_{\text{CWW}},n}^{\max}(\pi), \Pi_{\mathcal{C}_{\text{CWL}},n}$, and $\dim_{\mathcal{C}_{\text{CWL}},n}^{\max}(\pi)$ in Claim 6. Finally, in Step 2.3 we will verify $\dim_{\mathcal{C},n}^{\max}(\pi)$ and $\dim_{\mathcal{C}_{\text{CWL}},n}^{\max}(\pi)$ in Table 4.

**Step 2.1. Specify $N_{\overline{r}}$.** We first prove the following claim, which provides a sufficient condition for a polyhedron to be active for sufficiently large $N$.

**Claim 2.** *For any polyhedron $\mathcal{H}$ characterized by integer matrix $\mathbf{A}$ and $\vec{b} \leq \vec{0}$, if $\dim(\mathcal{H}_{\leq 0}) = m!$ and $\mathcal{H} \cap \mathbb{R}_{>0}^{m!} \neq \emptyset$, then there exists $N \in \mathbb{N}$ such that for all $n \geq N$, $\mathcal{H}$ is active at $n$.*

*Proof.* By Minkowski-Weyl theorem (see e.g., [46, p. 100]), $\mathcal{H} = \mathcal{V} + \mathcal{H}_{\leq 0}$, where $\mathcal{V}$ is a finitely generated polyhedron. Therefore, any affine space containing $\mathcal{H}$ can be shifted to contain $\mathcal{H}_{\leq 0}$, which means that $\dim(\mathcal{H}) \geq \dim(\mathcal{H}_{\leq 0}) = m!$. Because $\mathcal{H} \cap \mathbb{R}_{>0}^{m!} \neq \emptyset$, it contains an interior point (inner point with an full dimensional neighborhood), denoted by $\vec{x}$, whose $\delta$ neighborhood (for some $0 < \delta < 1$) in $L_\infty$ is contained in $\mathcal{H} \cap \mathbb{R}_{>0}^{m!}$. Let $B$ denote the $\delta$ neighborhood of $\vec{x}$. Let $N = \frac{m!|\vec{x}|_1}{\delta}$. Then, because $\vec{b} \leq \vec{0}$ and $\frac{N}{|\vec{x}|_1} \geq 1$, for every $n > N$ and every $\vec{x}' \in B$ we have

$$\mathbf{A} \cdot \left(\frac{n}{|\vec{x}|_1}\vec{x}'\right)^\top < \frac{n}{|\vec{x}|_1}\left(\vec{b}\right)^\top \leq \left(\vec{b}\right)^\top$$

This means that $\frac{n}{|\vec{x}|_1}B \subseteq \mathcal{H} \cap \mathbb{R}_{>0}^{m!}$. Moreover, it is not hard to verify that $\frac{n}{|\vec{x}|_1}B$ contains the following non-negative integer $n$ vector

$$\left(\left\lfloor \frac{n}{|\vec{x}|_1}x_1 \right\rfloor, \ldots, \left\lfloor \frac{n}{|\vec{x}|_1}x_{m!-1} \right\rfloor, n - \sum_{i=1}^{m!-1}\left\lfloor \frac{n}{|\vec{x}|_1}x_i \right\rfloor\right)$$

This proves Claim 2. $\qquad \square$

We now define the constant $N_{\overline{r}}$ used throughout the proof.

**Definition 24** ($N_{\overline{r}}$). *Let $N_{\overline{r}}$ denote a number that is larger than $m^4$ and the maximum $N$ obtain from applying Claim 2 to all polyhedra $\mathcal{H}$ in $\mathcal{C}_{NCW}, \mathcal{C}_{CWW},$ or $\mathcal{C}_{CWL}$ where $\dim(\mathcal{H}_{\leq 0}) = m!$ and $\mathcal{H} \cap \mathbb{R}_{>0}^{m!} \neq \emptyset$.*

**Step 2.2. Characterize $\Pi_{\mathcal{C}_{\text{NCW}},n}$, $\Pi_{\mathcal{C}_{\text{CWW}},n}$, and $\Pi_{\mathcal{C}_{\text{CWL}},n}$.**

**Claim 3 (Characterizations of $\Pi_{\mathcal{C}_{\text{NCW}},n}$ and $\dim^{\max}_{\mathcal{C}_{\text{NCW}},n}(\pi)$).** *For any $n \geq m^4$ such that $\neg C_{AS}(\overline{r}, n)$ and any distribution $\pi$ over $\mathcal{A}$, we have*

- *if $2 \mid n$, then $\pi \in \Pi_{\mathcal{C}_{NCW},n}$ if and only if $CW(\pi) = \emptyset$, and*

$$\dim^{\max}_{\mathcal{C}_{NCW},n}(\pi) = \begin{cases} -\frac{n}{\log n} & \text{if } CW(\pi) \neq \emptyset \\ m! - 1 & \text{if } ACW(\pi) \neq \emptyset \\ m! & \text{otherwise (i.e. } CW(\pi) \cup ACW(\pi) = \emptyset) \end{cases}$$

- *if $2 \nmid n$, then $\pi \in \Pi_{\mathcal{C}_{NCW},n}$ if and only if $CW(\pi) \cup ACW(\pi) = \emptyset$, and*

$$\dim^{\max}_{\mathcal{C}_{NCW},n}(\pi) = \begin{cases} -\frac{n}{\log n} & \text{if } CW(\pi) \cup ACW(\pi) \neq \emptyset \\ m! & \text{otherwise (i.e. } CW(\pi) \cup ACW(\pi) = \emptyset) \end{cases}$$

*Proof.* In the proof we assume that $n \geq m^4$. We first recall the following characterization of $\mathcal{H}^G$, where part (i)-(iii) are due to [54, Claim 3 in the Appendix] and part (iv) follows after [54, Claim 6 in the Appendix].

**Claim 4 (Properties of $\mathcal{H}^G$ [54]).** *For any UMG $G$,*

(i) *for any integral profile $P$, $Hist(P) \in \mathcal{H}^G$ if and only if $G = UMG(P)$;*

(ii) *for any $\vec{x} \in \mathbb{R}^{m!}$, $\vec{x} \in \mathcal{H}^G_{\leq 0}$ if and only if $UMG(\vec{x})$ is a subgraph of $G$.*

(iii) *$\dim(\mathcal{H}^G_{\leq 0}) = m! - Ties(G)$.*

(iv) *For any $n \geq m^4$, $\mathcal{H}^G$ is active at $n$ if (1) $n$ is even, or (2) $n$ is odd and $G$ is a complete graph.*

**The $2 \mid n$ case.** By Claim 4 (iv), when $n \geq m^4$ and $2 \mid n$, every $\mathcal{H}^G$ is active. This means that $\pi \in \Pi_{\mathcal{C}_{\text{NCW}},n}$ if and only if $\pi \in \mathcal{H}^G_{\leq 0}$ for some graph $G$ that does not have a Condorcet winner. According to Claim 4 (ii), this holds if and only if there exists a supergraph of $UMG(\pi)$ (which can be $UMG(\pi)$ itself) that not have a Condorcet winner, which is equivalent to $UMG(\pi)$ does not have a Condorcet winner, i.e. $CW(\pi) = \emptyset$. It follows that $\dim^{\max}_{\mathcal{C}_{\text{NCW}},n}(\pi) = -\frac{n}{\log n}$ if and only if $CW(\pi) \neq \emptyset$.

To characterize the $m! - 1$ case and the $m!$ case for $\dim^{\max}_{\mathcal{C}_{\text{NCW}},n}(\pi)$, we first prove the following claim to characterize graphs whose complete supergraphs all have Condorcet winners.

**Claim 5.** *For any unweighted directed graph $G$ over $\mathcal{A}$, the following conditions are equivalent. (1) Every complete supergraph of $G$ has a Condorcet winner. (2) $CW(G) \cup ACW(G) \neq \emptyset$.*

*Proof.* We first prove (1)⇒(2) in the following three cases.

- **Case 1: $|WCW(G)| = 1$.** In this case we must have $CW(G) = WCW(G)$, otherwise there exists an alternative $b$ that is different from the weak Condorcet winner, denoted by $a$, such that $a$ and $b$ are tied in $G$. Notice that $b$ is not a weak Condorcet winner. Therefore, we can complete $G$ by adding $b \to a$ and breaking other ties arbitrarily, and it is not hard to see that the resulting graph does not have a Condorcet winner, which is a contradiction.

- **Case 2: $|WCW(G)| = 2$.** Let $WCW(G) = \{a, b\}$. We note that $a$ and $b$ are not tied with any other alternative. Otherwise for the sake of contradiction suppose $a$ is tied with $c \neq b$. Then, we can extend $G$ to a complete graph by assigning $c \to a$ and $a \to b$. The resulting complete graph does not have a Condorcet winner, which is a contradiction. This means that $a$ and $b$ are the almost Condorcet winners, and hence (2) holds.

- **Case 3: $|WCW(G)| \geq 3$.** In this case, we can assign directions of edges between $WCW(G)$ to form a cycle, and then assign arbitrary direction to other missing edges in $G$ to form a complete graph, which does not have a Condorcet winner and is thus a contradiction.

(2)⇒(1) is straightforward. If $\mathrm{CW}(G) \neq \emptyset$, then any supergraph of $G$ has the same Condorcet winner. If $\mathrm{ACW}(G) = \{a, b\} \neq \emptyset$, then any complete supergraph of $G$ either has $a$ as the Condorcet winner or has $b$ as the Condorcet winner. This proves Claim 5. $\qquad\square$

**The $\dim_{\mathcal{C}_{\mathrm{NCW}},n}^{\max}(\pi) = m! - 1$ case when $2 \mid n$.** Suppose $\mathrm{ACW}(\pi) = \{a, b\}$. Let $G^*$ denote a supergraph of $\mathrm{UMG}(\pi)$ where ties in $\mathrm{UMG}(\pi)$ except $\{a, b\}$ are broken arbitrarily. By Claim 4 (ii), $\pi \in \mathcal{H}_{\leq 0}^{G^*}$ and by Claim 4 (iii), $\mathcal{H}_{\leq 0}^{G^*} = m! - 1$. Recall from Claim 4 (iv) that $\mathcal{H}^{G^*}$ is active at $n$ because we assumed that $n > m^4$. Therefore, $\dim_{\mathcal{C}_{\mathrm{NCW}},n}^{\max}(\pi) \geq m! - 1$. To see that $\dim_{\mathcal{C}_{\mathrm{NCW}},n}^{\max}(\pi) \leq m! - 1$, we note that for every graph $G$ that does not have a Condorcet winner such that $\pi \in \mathcal{H}_{\leq 0}^{G}$. By Claim 4 (ii), $G$ is a supergraph of $\mathrm{UMG}(\pi)$. This means that $G$ is not a complete graph, because by Claim 5, any complete supergraph of $\mathrm{UMG}(\pi)$ must have a Condorcet winner. It follows that $\mathrm{Ties}(G) \geq 1$ and by Claim 4 (iii), $\mathcal{H}_{\leq 0}^{G} \leq m! - 1$. Therefore, $\dim_{\mathcal{C}_{\mathrm{NCW}},n}^{\max}(\pi) = m! - 1$.

**The $\dim_{\mathcal{C}_{\mathrm{NCW}},n}^{\max}(\pi) = m!$ case when $2 \mid n$.** Suppose $\mathrm{CW}(\pi) \cup \mathrm{ACW}(\pi) = \emptyset$. By Claim 5 there exists a complete supergraph $G$ of $\mathrm{UMG}(\pi)$ that does not have a Condorcet winner, which means that $\mathcal{H}^{G} \subseteq \mathcal{C}_{\mathrm{NCW}} \subseteq \mathcal{C}$. We have $\pi \in \mathcal{H}_{\leq 0}^{G}$ (Claim 4 (ii)), $\dim(\mathcal{H}_{\leq 0}^{G}) = m!$ (Claim 4 (iii)), and $\mathcal{H}^{G}$ is active at $n$ (Claim 4 (iv)). Therefore, $\dim_{\mathcal{C}_{\mathrm{NCW}},n}^{\max}(\pi) = m!$.

**The $2 \nmid n$ case.** By Claim 4 (iv), when $n \geq m^4$ and $2 \nmid n$, $\mathcal{H}^{G}$ is active if and only if $G$ is a complete graph. It follows from Claim 4 (ii) that $\pi \in \Pi_{\mathcal{C}_{\mathrm{NCW}},n}$ if and only if $\pi \in \mathcal{H}_{\leq 0}^{G}$, where $G$ is complete supergraph of $\mathrm{UMG}(\pi)$ that does not have a Condorcet winner. By Claim 4 (iii), $\dim(\mathcal{H}_{\leq 0}^{G}) = m!$. Therefore, by Claim 5, $\pi \in \Pi_{\mathcal{C}_{\mathrm{NCW}},n}$ if and only if $\mathrm{CW}(\pi) \cup \mathrm{ACW}(\pi) = \emptyset$. Moreover, whenever $\pi \in \Pi_{\mathcal{C}_{\mathrm{NCW}},n}$ we have $\dim_{\mathcal{C}_{\mathrm{NCW}},n}^{\max}(\pi) = m!$.

This proves Claim 3. $\qquad\square$

Recall that we have assumed the 1 case of the theorem does not hold, that is, $\neg \mathrm{C}_{\mathrm{AS}}(\bar{r}, n)$. The following claim characterizes $\Pi_{\mathcal{C}_{\mathrm{CWW}},n}$, $\dim_{\mathcal{C}_{\mathrm{CWW}},n}^{\max}(\pi)$, $\Pi_{\mathcal{C}_{\mathrm{CWL}},n}$, and $\dim_{\mathcal{C}_{\mathrm{CWL}},n}^{\max}(\pi)$, when $\neg \mathrm{C}_{\mathrm{AS}}(\bar{r}, n)$.

**Claim 6 (Characterizations of $\Pi_{\mathcal{C}_{\mathrm{CWW}},n}$, $\dim_{\mathcal{C}_{\mathrm{CWW}},n}^{\max}(\pi)$, $\Pi_{\mathcal{C}_{\mathrm{CWL}},n}$, and $\dim_{\mathcal{C}_{\mathrm{CWL}},n}^{\max}(\pi)$).** *Given any strictly positive $\Pi$ and any minimally continuous int-GISR $\bar{r}$, for any $n \geq N_{\bar{r}}$ (see Definition 24) such that $\neg \mathrm{C}_{\mathrm{AS}}(\bar{r}, n)$ and any $\pi \in CH(\Pi)$,*

$$\left[\pi \in \Pi_{\mathcal{C}_{\mathrm{CWW}},n}\right] \Leftrightarrow \left[\pi \in Closure(\mathcal{R}_{\mathrm{CWW}}^{\bar{r}})\right] \Leftrightarrow \left[\dim_{\mathcal{C}_{\mathrm{CWW}},n}^{\max}(\pi) = m!\right], \text{ and}$$

$$\left[\pi \in \Pi_{\mathcal{C}_{\mathrm{CWL}},n}\right] \Leftrightarrow \left[\pi \in Closure(\mathcal{R}_{\mathrm{CWL}}^{\bar{r}})\right] \Leftrightarrow \left[\dim_{\mathcal{C}_{\mathrm{CWL}},n}^{\max}(\pi) = m!\right]$$

*Proof.* We first prove properties of $\mathcal{H}^{a, \vec{t}}$ in the following claim, which has three parts. Part (i) states that $\mathcal{H}^{a, \vec{t}}$ characterizes histograms of the profiles whose signature is $\vec{t}$ and where alternative $a$ is the Condorcet winner. Part (ii) characterizes the characteristic cone of $\mathcal{H}^{a, \vec{t}}$. Part (iii) characterizes the dimension of the characteristic cone for some cases.

**Claim 7 (Properties of $\mathcal{H}^{a, \vec{t}}$).** *Given $\vec{H}$, for any $a \in \mathcal{A}$ and any $\vec{t} \in \mathcal{S}_{\vec{H}}$,*

- *(i) for any integral profile $P$, $\mathrm{Hist}(P) \in \mathcal{H}^{a, \vec{t}}$ if and only if $a$ is the Condorcet winner under $P$ and $\mathrm{Sign}_{\vec{H}}(P) = \vec{t}$;*

- *(ii) for any $\vec{x} \in \mathbb{R}^{m!}$, $\vec{x} \in \mathcal{H}_{\leq 0}^{a, \vec{t}}$ if and only if $a$ is a weak Condorcet winner under $\vec{x}$ and $\vec{t} \trianglelefteq \mathrm{Sign}_{\vec{H}}(\vec{x})$;*

- *(iii) if $\vec{t} \in \mathcal{S}_{\vec{H}}^{\circ}$ and $\mathcal{H}^{a, \vec{t}} \neq \emptyset$, then $\dim(\mathcal{H}_{\leq 0}^{a, \vec{t}}) = m!$.*

*Proof.* Part (i) follows after the definition. More precisely, $\mathbf{A}^{\mathrm{CW}=a} \cdot (\mathrm{Hist}(P))^{\top} \leq \left(-\vec{1}\right)^{\top}$ if and only if $a$ is the Condorcet winner under $P$, and by Claim 1 (i), $\mathbf{A}^{\vec{t}} \cdot (\mathrm{Hist}(P))^{\top} \leq \left(\vec{\mathbf{b}^{t}}\right)^{\top}$ if and only if $\mathrm{Sign}_{\vec{H}}(\mathrm{Hist}(P)) = \vec{t}$.

Part (ii) also follows after the definition. $\mathbf{A}^{\mathrm{CW}=a} \cdot (\vec{x})^\top \le \left(\vec{0}\right)^\top$ if and only if $a$ is a weak Condorcet winner under $P$, and by Claim 1 (ii), $\mathbf{A}^{\vec{t}} \cdot (\vec{x})^\top \le \left(\vec{0}\right)^\top$ if and only if $\vec{t} \trianglelefteq \mathrm{Sign}_{\vec{H}}(\vec{x})$.

To prove Part (iii), suppose $\vec{x} \in \mathcal{H}^{a,\vec{t}}$. Because $\vec{t} \in \mathcal{S}_{\vec{H}}^\circ$, we have $\vec{\mathbf{b}}^{a,\vec{t}} = -\vec{1}$ (Definition 21). Therefore, there exists $\delta > 0$ such that for all vector $\vec{x}'$ such that $|\vec{x}' - \vec{x}|_1 < \delta$, $\mathbf{A}^{a,\vec{t}} \cdot (\vec{x}')^\top < \left(\vec{0}\right)^\top$, which means that $\vec{x}' \in \mathcal{H}_{\le 0}^{a,\vec{t}}$. Therefore, $\mathcal{H}_{\le 0}^{a,\vec{t}}$ contains the $\delta$ neighborhood of $\vec{x}$, whose dimension is $m!$. This means that $\dim(\mathcal{H}_{\le 0}^{a,\vec{t}}) = m!$. $\qquad\square$

$\left[\pi \in \Pi_{\mathcal{C}_{\mathrm{CWW}},n}\right] \Leftarrow \left[\pi \in \mathbf{Closure}(\mathcal{R}_{\mathbf{CWW}}^{\bar{r}})\right].$ Suppose $\pi \in \mathrm{Closure}(\mathcal{R}_{\mathrm{CWW}}^{\bar{r}})$ and let $(\vec{x}_1, \vec{x}_2, \ldots)$ denote an infinite sequence in $\mathcal{R}_{\mathrm{CWW}}^{\bar{r}}$ that converges to $\pi$. Because the number of alternatives and the number of feasible signatures are finite, there exists an infinite subsequence $(\vec{x}_1', \vec{x}_2', \ldots)$ such that (1) there exists $a \in \mathcal{A}$ such that for all $j \in \mathbb{N}$, $\mathrm{CW}(\vec{x}_j') = \{a\}$, and (2) there exists $\vec{t} \in \mathcal{S}_{\vec{H}}$ such that $a \in \bar{r}(\vec{t})$ and for all $j \in \mathbb{N}$, $\mathrm{Sign}_{\vec{H}}(\vec{x}_j') = \vec{t}$. Because $\bar{r}$ is minimally continuous, by Proposition 4, there exists a feasible atomic refinement of $\vec{t}$, denoted by $\vec{t}_a \in \mathcal{S}_{\vec{H}}^\circ$, such that $\bar{r}(\vec{t}_a) = \{a\}$. Therefore, to prove that $\pi \in \Pi_{\mathcal{C}_{\mathrm{CWW}},n}$, it suffices to prove that (i) for every $n > N_{\bar{r}}$, $\mathcal{H}^{a,\vec{t}_a}$ is active, and (ii) $\pi \in \mathcal{H}_{\le 0}^{a,\vec{t}_a}$, which will be done as follows.

(i) $\mathcal{H}^{a,\vec{t}_a}$ is active. By Claim 2, it suffices to prove that $\mathcal{H}^{a,\vec{t}_a} \cap \mathbb{R}_{>0}^{m!} \ne \emptyset$. This is proved by explicitly constructing a vector in $\mathcal{H}^{a,\vec{t}_a} \cap \mathbb{R}_{\ge 0}^{m!}$ as follows. Because $\vec{t}_a$ is feasible, there exists $\vec{x}^a \in \mathbb{R}^{m!}$ such that $\mathrm{Sign}_{\vec{H}}(\vec{x}^a) = \vec{t}_a$. Recall that $\pi$ is strictly positive and $(\vec{x}_1', \vec{x}_2', \ldots)$ converges to $\pi$, there exists $j \in \mathbb{N}$ such that $\vec{x}_j' > \vec{0}$. For any $\delta > 0$, let $\vec{x}_\delta = \vec{x}_j' + \delta \vec{x}^a$. We let $\delta > 0$ denote a sufficiently small number such that the following two conditions hold.

- $\vec{x}_\delta > \vec{0}$. The existence of such $\delta$ follows after noticing that $\vec{x}_j' > \vec{0}$.

- $\mathrm{CW}(\vec{x}_\delta) = \{a\}$. The existence of such $\delta$ is due to the assumption that $\mathrm{CW}(\vec{x}_j') = \{a\}$, which means that $\mathbf{A}^{\mathrm{CW}=a} \cdot \left(\vec{x}_j'\right)^\top < \left(\vec{0}\right)^\top$, where $\mathbf{A}^{\mathrm{CW}=a}$ is defined in Definition 21. Therefore, for any sufficiently small $\delta > 0$ we have $\mathbf{A}^{\mathrm{CW}=a} \cdot (\vec{x}_\delta)^\top < \left(\vec{0}\right)^\top$, which means that $a$ is the Condorcet winner under $\vec{x}_\delta$.

Because $\vec{t}_a$ is a refinement of $\vec{t}$, we have $\mathrm{Sign}_{\vec{H}}(\vec{x}_\delta) = \vec{t}_a$. Therefore, $\vec{x}_\delta \in \mathcal{H}^{a,\vec{t}_a} \cap \mathbb{R}_{>0}^{m!}$. Following Claim 2 and the definition of $N_{\bar{r}}$ (Definition 24), we have that $\mathcal{H}^{a,\vec{t}_a}$ is active for all $n > N_{\bar{r}}$.

(ii) $\pi \in \mathcal{H}_{\le 0}^{a,\vec{t}_a}$. Because for all $j \in \mathbb{N}$, $\mathbf{A}^{\mathrm{CW}=a} \cdot \left(\vec{x}_j'\right)^\top < \left(\vec{0}\right)^\top$ and $(\vec{x}_1', \vec{x}_2', \ldots)$ converge to $\pi$, we have $\mathbf{A}^{\mathrm{CW}=a} \cdot (\pi)^\top \le \left(\vec{0}\right)^\top$, which means that $a$ is a weak Condorcet winner under $\pi$. It is not hard to verify that for every $k \le K$, if $t_k = +$ (respectively, $-$ and $0$), then we have $[\mathrm{Sign}_{\vec{H}}(\pi)]_k \in \{0, +\}$ (respectively, $\{0, -\}$ and $\{0\}$). Therefore, $\vec{t} \trianglelefteq \mathrm{Sign}_{\vec{H}}(\pi)$, which means that $\vec{t}_a \trianglelefteq \mathrm{Sign}_{\vec{H}}(\pi)$ because $\vec{t}_a \trianglelefteq \vec{t}$. By Claim 7 (ii), we have $\pi \in \mathcal{H}_{\le 0}^{a,\vec{t}_a}$.

$\left[\pi \in \Pi_{\mathcal{C}_{\mathrm{CWW}},n}\right] \Rightarrow \left[\pi \in \mathbf{Closure}(\mathcal{R}_{\mathbf{CWW}}^{\bar{r}})\right].$ Suppose $\pi \in \Pi_{\mathcal{C}_{\mathrm{CWW}},n}$, which means that there exists $a \in \mathcal{A}$ and $\vec{t} \in \mathcal{S}_{\vec{H}}$ such that $\pi \in \mathcal{H}_{\le 0}^{a,\vec{t}}$, $a \in \bar{r}(\vec{t})$, $\mathrm{CW}(\vec{t}) = \{a\}$, and $\mathcal{H}^{a,\vec{t}}$ contains a non-negative integer $n$-vector, denoted by $\vec{x}'$. By Proposition 4, because $\bar{r}$ is minimally continuous, there exists $\vec{t}_a \in \mathcal{S}_{\vec{H}}^\circ$ such that $\vec{t}_a \trianglelefteq \vec{t}$ and $\bar{r}(\vec{t}_a) = \{a\}$. Let $\vec{x}^* \in \mathcal{H}^{\vec{t}_a}$ denote an arbitrary vector, which is guaranteed to exist because $\vec{t}_a \in \mathcal{S}_{\vec{H}}^\circ$. Because $\vec{x}' \in \mathcal{H}^{a,\vec{t}}$, we have $\mathbf{A}^{\mathrm{CW}=a} \cdot (\vec{x}')^\top \le \left(-\vec{1}\right)^\top$. Therefore, there exists $\delta_a$ such that $\mathbf{A}^{\mathrm{CW}=a} \cdot (\vec{x}' + \delta_a \vec{x}^*)^\top < \left(\vec{0}\right)^\top$. Let $\vec{x} = \vec{x}' + \delta_a \vec{x}^*$. Recall that

$\pi \in \mathcal{H}_{\leq 0}^{a, \vec{t}}$, which means that $\mathbf{A}^{\mathrm{CW}=a} \cdot (\pi)^\top \leq \left(\vec{0}\right)^\top$. Therefore, for all $\delta > 0$ we have

$$\mathbf{A}^{\mathrm{CW}=a} \cdot (\pi + \delta \vec{x})^\top = \mathbf{A}^{\mathrm{CW}=a} \cdot (\pi)^\top + \delta \mathbf{A}^{\mathrm{CW}=a} \cdot (\vec{x})^\top < \left(\vec{0}\right)^\top,$$

which means that $\mathrm{CW}(\pi + \delta \vec{x}) = \{a\}$. It is not hard to verify that $\mathrm{Sign}_{\vec{H}}(\pi + \delta \vec{x}) = \vec{t}_a$, which means that $\overline{r}(\pi + \delta \vec{x}) = \{a\}$. Consequently, for every $\delta > 0$ we have $\pi + \delta \vec{x} \in \mathcal{R}_{\mathrm{CWW}}^{\overline{r}}$. Notice that the sequence $(\pi + \vec{x}, \pi + \frac{1}{2}\vec{x}, \ldots)$ converges to $\pi$. Therefore, $\pi \in \mathrm{Closure}(\mathcal{R}_{\mathrm{CWW}}^{\overline{r}})$.

$\left[\pi \in \mathbf{Closure}(\mathcal{R}_{\mathbf{CWW}}^{\overline{r}})\right] \Rightarrow \left[\mathbf{dim}_{\mathcal{C}_{\mathbf{CWW}}, n}^{\mathbf{max}}(\pi) = m!\right]$. Continuing the proof of the $[\pi \in \Pi_{\mathcal{C}_{\mathrm{CWW}}, n}] \Rightarrow \left[\pi \in \mathrm{Closure}(\mathcal{R}_{\mathrm{CWW}}^{\overline{r}})\right]$ part, because $\pi$ is strictly positive and $(\pi + \vec{x}, \pi + \frac{1}{2}\vec{x}, \ldots)$ converges to $\pi$, there exists $j \in \mathbb{N}$ such that $\pi + \frac{1}{j}\vec{x} > \vec{0}$. Recall that $\mathrm{CW}(\pi + \frac{1}{j}\vec{x}) = \{a\}$, $\mathrm{Sign}_{\vec{H}}(\pi + \frac{1}{j}\vec{x}) = \vec{t}_a$, and $\vec{t}_a$ is atomic, we have

$$\mathbf{A}^{\mathrm{CW}=a} \cdot \left(\pi + \frac{1}{j}\vec{x}\right)^\top < \left(\vec{0}\right)^\top \quad \text{and} \quad \mathbf{A}^{\vec{t}_a} \cdot \left(\pi + \frac{1}{j}\vec{x}\right)^\top < \left(\vec{0}\right)^\top$$

Therefore, there exists $\ell > 0$ such that

$$\mathbf{A}^{\mathrm{CW}=a} \cdot \left(\ell(\pi + \frac{1}{j}\vec{x})\right)^\top \leq \left(-\vec{1}\right)^\top \quad \text{and} \quad \mathbf{A}^{\vec{t}_a} \cdot \left(\ell(\pi + \frac{1}{j}\vec{x})\right)^\top \leq \left(-\vec{1}\right)^\top,$$

which means that $\ell(\pi + \frac{1}{j}\vec{x}) \in \mathcal{H}^{a, \vec{t}_a} \cap \mathbb{R}_{>0}^{m!} \neq \emptyset$. by Claim 7 (iii), we have $\dim_{\mathcal{C}_{\mathrm{CWW}}, n}^{\max}(\pi) = m!$.

$\left[\mathbf{dim}_{\mathcal{C}_{\mathbf{CWW}}, n}^{\mathbf{max}}(\pi) = m!\right] \Rightarrow [\pi \in \Pi_{\mathcal{C}_{\mathbf{CWW}}, n}]$ follows after the definition of $\Pi_{\mathcal{C}_{\mathrm{CWW}}, n}$. More concretely, $\dim_{\mathcal{C}_{\mathrm{CWW}}, n}^{\max}(\pi) = m!$ means that there exists a polyhedron $\mathcal{H} \subseteq \mathcal{C}_{\mathrm{CWW}}$ such that the weight on the edge $(\pi, \mathcal{H})$ in the activation graph is $m!$, which implies that $\pi \in \Pi_{\mathcal{C}_{\mathrm{CWW}}, n}$.

The proofs for $\Pi_{\mathcal{C}_{\mathrm{CWL}}, n}$ and $\dim_{\mathcal{C}_{\mathrm{CWL}}, n}^{\max}(\pi)$ are similar to the proofs for $\Pi_{\mathcal{C}_{\mathrm{CWW}}, n}$ and $\dim_{\mathcal{C}_{\mathrm{CWW}}, n}^{\max}(\pi)$. For completeness, we include the full proofs below.

$[\pi \in \Pi_{\mathcal{C}_{\mathbf{CWL}}, n}] \Leftarrow \left[\pi \in \mathbf{Closure}(\mathcal{R}_{\mathbf{CWL}}^{\overline{r}})\right]$. Suppose $\pi \in \mathrm{Closure}(\mathcal{R}_{\mathrm{CWL}}^{\overline{r}})$ and let $(\vec{x}_1, \vec{x}_2, \ldots)$ denote an infinite sequence in $\mathcal{R}_{\mathrm{CWL}}^{\overline{r}}$ that converges to $\pi$. Because the number of alternatives and the number of feasible signatures are finite, there exists an infinite subsequence $(\vec{x}_1', \vec{x}_2', \ldots)$ such that (1) there exists $a \in \mathcal{A}$ such that for all $j \in \mathbb{N}$, $\mathrm{CW}(\vec{x}_j') = \{a\}$, and (2) there exists $\vec{t} \in \mathcal{S}_{\vec{H}}$ such that $a \notin \overline{r}(\vec{t})$ and for all $j \in \mathbb{N}$, $\mathrm{Sign}_{\vec{H}}(\vec{x}_j') = \vec{t}$. Let $b \in \overline{r}(\vec{t})$ denote an arbitrary winner. Because $\overline{r}$ is minimally continuous, by Proposition 4, there exists a feasible atomic refinement of $\vec{t}$, denoted by $\vec{t}_b$, such that $\overline{r}(\vec{t}_b) = \{b\}$. Therefore, to prove that $\pi \in \Pi_{\mathcal{C}_{\mathrm{CWL}}, n}$, it suffices to show that (i) for every $n > N$, $\mathcal{H}^{a, \vec{t}_b}$ is active, and (ii) $\pi \in \mathcal{H}_{\leq 0}^{a, \vec{t}_b}$.

**(i) $\mathcal{H}^{a, \vec{t}_b}$ is active.** We will apply Claim 2 to prove that $\mathcal{H}^{a, \vec{t}_b}$ is active at every $n > N$. In fact, it suffices to prove that $\mathcal{H}^{a, \vec{t}_b} \cap \mathbb{R}_{>0}^{m!} \neq \emptyset$. This will be proved by explicitly constructing a vector in $\mathcal{H}^{a, \vec{t}_b} \cap \mathbb{R}_{>0}^{m!}$ as follows. Because $\vec{t}_b$ is feasible, there exists $\vec{x}^b \in \mathbb{R}^{m!}$ such that $\mathrm{Sign}_{\vec{H}}(\vec{x}^b) = \vec{t}_b$. Recall that $\pi$ is strictly positive and $(\vec{x}_1', \vec{x}_2', \ldots)$ converges to $\pi$, there exists $j \in \mathbb{N}$ such that $\vec{x}_j' > \vec{0}$. For any $\delta > 0$, let $\vec{x}_\delta = \vec{x}_j' + \delta \vec{x}^b$. We let $\delta > 0$ denote a sufficiently small number such that the following two conditions hold.

- $\vec{x}_\delta > \vec{0}$. The existence of such $\delta$ follows after noticing that $\vec{x}_j' > \vec{0}$.

- $\mathrm{CW}(\vec{x}_\delta) = \{a\}$. The existence of such $\delta$ is due to the assumption that $\mathrm{CW}(\vec{x}_j') = \{a\}$, which means that $\mathbf{A}^{\mathrm{CW}=a} \cdot \left(\vec{x}_j'\right)^\top < \left(\vec{0}\right)^\top$, where $\mathbf{A}^{\mathrm{CW}=a}$ is defined in Definition 21.

  Therefore, for any sufficiently small $\delta > 0$ we have $\mathbf{A}^{\mathrm{CW}=a} \cdot (\vec{x}_\delta)^\top < \left(\vec{0}\right)^\top$, which means that $a$ is the Condorcet winner under $\vec{x}_\delta$.

Because $\vec{t}_b$ is a refinement of $\vec{t}$, we have $\text{Sign}_{\vec{H}}(\vec{x}_\delta) = \vec{t}_b$. Therefore, $\vec{x}_\delta \in \mathcal{H}^{a,\vec{t}_b} \cap \mathbb{R}_{>0}^{m!}$. Following Claim 2 and the definition of $N_{\overline{r}}$ (Definition 24), we have that $\mathcal{H}^{a,\vec{t}_a}$ is active for all $n > N_{\overline{r}}$.

**(ii) $\pi \in \mathcal{H}_{\leq 0}^{a,\vec{t}_b}$.** Because for all $j \in \mathbb{N}$, $\mathbf{A}^{\text{CW}=a} \cdot \left(\vec{x}_j'\right)^\top < \left(\vec{0}\right)^\top$ and $(\vec{x}_1', \vec{x}_2', \ldots)$ converge to $\pi$, we have $\mathbf{A}^{\text{CW}=a} \cdot (\pi)^\top \leq \left(\vec{0}\right)^\top$, which means that $\pi$ is a weak Condorcet winner. It is not hard to verify that for every $k \leq K$, if $t_k = +$ (respectively, $-$ and $0$), then we have $[\text{Sign}_{\vec{H}}(\pi)]_k \in \{0, +\}$ (respectively, $\{0, -\}$ and $\{0\}$). Therefore, $\vec{t} \trianglelefteq \text{Sign}_{\vec{H}}(\pi)$, which means that $\vec{t}_b \trianglelefteq \text{Sign}_{\vec{H}}(\pi)$ because $\vec{t}_b \trianglelefteq \vec{t}$. It follows that $\mathbf{A}^{\vec{t}_b} \cdot (\pi)^\top \leq \left(\vec{0}\right)^\top$. This means that $\pi \in \mathcal{H}_{\leq 0}^{a,\vec{t}_b}$.

**$\left[\pi \in \Pi_{\mathcal{C}_{\text{CWL}},n}\right] \Rightarrow \left[\pi \in \text{Closure}(\mathcal{R}_{\text{CWL}}^{\overline{r}})\right]$.** Suppose $\pi \in \Pi_{\mathcal{C}_{\text{CWL}},n}$, which means that there exists $a \in \mathcal{A}$ and $\vec{t} \in \mathcal{S}_{\vec{H}}$ such that $\pi \in \mathcal{H}_{\leq 0}^{a,\vec{t}} \subseteq \mathcal{C}_{\text{CWL}}$, $a \notin \overline{r}(\vec{t})$, $\text{CW}(\pi) = \{a\}$, and $\mathcal{H}^{a,\vec{t}}$ contains a non-negative integer $n$-vector, denoted by $\vec{x}'$. Let $b \in \overline{r}(\vec{t})$ denote an arbitrary co-winner. By Proposition 4, because $\overline{r}$ is minimally continuous, there exists $\vec{t}_b \in \mathcal{S}_{\vec{H}}^\circ$ such that $\vec{t}_b \trianglelefteq \vec{t}$ and $\overline{r}(\vec{t}_b) = \{b\}$. Let $\vec{x}^* \in \mathcal{H}^{\vec{t}_b}$ denote an arbitrary vector whose existence is guaranteed by the assumption that $\vec{t}_b \in \mathcal{S}_{\vec{H}}^\circ$. Because $\vec{x}' \in \mathcal{H}^{a,\vec{t}}$, we have $\mathbf{A}^{\text{CW}=a} \cdot \left(\vec{x}'\right)^\top \leq \left(-\vec{1}\right)^\top$. Therefore, there exists $\delta_a$ such that $\mathbf{A}^{\text{CW}=a} \cdot \left(\vec{x}' + \delta_a \vec{x}^*\right)^\top < \left(\vec{0}\right)^\top$. Let $\vec{x} = \vec{x}' + \delta_a \vec{x}^*$. Recall that $\pi \in \mathcal{H}_{\leq 0}^{a,\vec{t}}$, which means that $\mathbf{A}^{\text{CW}=a} \cdot (\pi)^\top \leq \left(\vec{0}\right)^\top$. Therefore, for all $\delta > 0$ we have $\mathbf{A}^{\text{CW}=a} \cdot (\pi + \delta\vec{x})^\top < \left(\vec{0}\right)^\top$, which means that $\text{CW}(\pi + \delta\vec{x}) = \{a\}$. It is not hard to verify that $\text{Sign}_{\vec{H}}(\pi + \delta\vec{x}) = \vec{t}_b$, which means that $\overline{r}(\pi + \delta\vec{x}) = \{b\}$. This means that for every $\delta > 0$ we have $\pi + \delta\vec{x} \in \mathcal{R}_{\text{CWL}}^{\overline{r}}$. Notice that $\pi$ is the limit of the sequence $(\pi + \vec{x}, \pi + \frac{1}{2}\vec{x}, \ldots)$. Therefore, $\pi \in \text{Closure}(\mathcal{R}_{\text{CWL}}^{\overline{r}})$.

**$\left[\pi \in \text{Closure}(\mathcal{R}_{\text{CWL}}^{\overline{r}})\right] \Rightarrow \left[\dim_{\mathcal{C}_{\text{CWL}},n}^{\max}(\pi) = m!\right]$.** Continuing the proof of the $\left[\pi \in \Pi_{\mathcal{C}_{\text{CWL}},n}\right] \Rightarrow \left[\pi \in \text{Closure}(\mathcal{R}_{\text{CWL}}^{\overline{r}})\right]$ part, because $\pi$ is strictly positive and $(\pi + \vec{x}, \pi + \frac{1}{2}\vec{x}, \ldots)$ converges to $\pi$, there exists $j \in \mathbb{N}$ such that $\pi + \frac{1}{j}\vec{x} > \vec{0}$. Recall that $\text{CW}(\pi + \frac{1}{j}\vec{x}) = \{a\}$, $\text{Sign}_{\vec{H}}(\pi + \frac{1}{j}\vec{x}) = \vec{t}_b$, and $\vec{t}_b$ is atomic, which means that $\mathbf{A}^{\text{CW}=a} \cdot \left(\pi + \frac{1}{j}\vec{x}\right)^\top < \left(\vec{0}\right)^\top$ and $\mathbf{A}^{\vec{t}_b} \cdot \left(\pi + \frac{1}{j}\vec{x}\right)^\top < \left(\vec{0}\right)^\top$. Therefore, there exists $\ell > 0$ such that

$$\mathbf{A}^{\text{CW}=a} \cdot \left(\ell(\pi + \frac{1}{j}\vec{x})\right)^\top \leq \left(-\vec{1}\right)^\top \text{ and } \mathbf{A}^{\vec{t}_b} \cdot \left(\ell(\pi + \frac{1}{j}\vec{x})\right)^\top \leq \left(-\vec{1}\right)^\top,$$

which means that $\ell(\pi + \frac{1}{j}\vec{x}) \in \mathcal{H}^{a,\vec{t}_b} \cap \mathbb{R}_{>0}^{m!} \neq \emptyset$. by Claim 7 (iii), we have $\dim_{\mathcal{C}_{\text{CWL}},n}^{\max}(\pi) = m!$.

**$\left[\dim_{\mathcal{C}_{\text{CWL}},n}^{\max}(\pi) = m!\right] \Rightarrow \left[\pi \in \Pi_{\mathcal{C}_{\text{CWL}},n}\right]$** follows after the definition.

This proves Claim 6. $\qquad\qquad\square$

We are now ready to verify Table 4 column by column as follows.

- **\*YY:** $\dim_{\mathcal{C},n}^{\max}(\pi) = \max(\dim_{\mathcal{C}_{\text{NCW}},n}^{\max}(\pi), \dim_{\mathcal{C}_{\text{CWW}},n}^{\max}(\pi))$, and by Claim 6 we have $\dim_{\mathcal{C}_{\text{CWW}},n}^{\max}(\pi) = m!$. The $\dim_{\mathcal{C}_{\text{CWL}},n}^{\max}(\pi)$ part also follows after Claim 6.

- **\*YN:** The $\dim_{\mathcal{C},n}^{\max}(\pi)$ part follows after Claim 6. Recall that we have assumed $\neg\text{C}_{\text{AS}}(\overline{r}, n)$. This means that there exists an $n$-profile $P$ such that $\text{CW}(P) \neq \emptyset$ and $\text{CW}(P) \not\subseteq \overline{r}(P)$. Let $\{a\} = \text{CW}(P)$ and $\vec{t} = \text{Sign}_{\vec{H}}(P)$. It follows that $\text{Hist}(P) \in \mathcal{H}_n^{a,\vec{t},\mathbb{Z}} \neq \emptyset$ and $\mathcal{H}^{a,\vec{t}} \subseteq \mathcal{C}_{\text{CWL}}$. Because $\pi \notin \Pi_{\mathcal{C}_{\text{CWL}},n}$, according to the definition of the activation graph (Definition 6), the weight on the edge $(\pi, \mathcal{H}^{a,\vec{t}})$ is $-\frac{n}{\log n}$, and the weight on any edge connected to $\pi$ is not positive. Therefore, $\dim_{\mathcal{C}_{\text{CWL}},n}^{\max}(\pi) = -\frac{n}{\log n}$.

- **NNY:** The $\dim_{\mathcal{C},n}^{\max}(\pi)$ part follows after the definition. The $\dim_{\mathcal{C}_{\mathrm{CWL}},n}^{\max}(\pi)$ part follows after Claim 6.

- **YNY:** Recall that the "N" means that $\pi \notin \Pi_{\mathcal{C}_{\mathrm{CWW}},n}$, which implies that $\dim_{\mathcal{C}_{\mathrm{CWW}},n}^{\max}(\pi) < 0$. Therefore, $\dim_{\mathcal{C},n}^{\max}(\pi) = \max(\dim_{\mathcal{C}_{\mathrm{NCW}},n}^{\max}(\pi), \dim_{\mathcal{C}_{\mathrm{CWW}},n}^{\max}(\pi))$, which means that $\dim_{\mathcal{C},n}^{\max}(\pi) = \dim_{\mathcal{C}_{\mathrm{NCW}},n}^{\max}(\pi)$. The $\dim_{\mathcal{C}_{\mathrm{CWL}},n}^{\max}(\pi)$ part follows after Claim 6.

- **YNN:** We first prove the $\dim_{\mathcal{C},n}^{\max}(\pi)$ part. Because in this case $\pi \in \Pi_{\mathcal{C}_{\mathrm{NCW}},n}$ and $\pi \notin \Pi_{\mathcal{C}_{\mathrm{CWW}},n}$, by the definition of $\Pi_{\mathcal{C}_{\mathrm{NCW}},n}$ and $\Pi_{\mathcal{C}_{\mathrm{CWW}},n}$, we have $\dim_{\mathcal{C}_{\mathrm{NCW}},n}^{\max}(\pi) \geq 0$ and $\dim_{\mathcal{C}_{\mathrm{CWW}},n}^{\max}(\pi) \leq -\frac{n}{\log n}$. Therefore, $\dim_{\mathcal{C},n}^{\max}(\pi) = \dim_{\mathcal{C}_{\mathrm{NCW}},n}^{\max}(\pi)$. It suffices to prove that $\dim_{\mathcal{C}_{\mathrm{NCW}},n}^{\max}(\pi) = m!$. Recall from Proposition 1 that

$$\mathcal{C}_{\mathrm{NCW},\leq 0} \cup \mathcal{C}_{\mathrm{CWW},\leq 0} \cup \mathcal{C}_{\mathrm{CWL},\leq 0} = \mathbb{R}^{m!}$$

  Therefore, there exists a polyhedron $\mathcal{H}$ in $\mathcal{C}_{\mathrm{NCW}}$, $\mathcal{C}_{\mathrm{CWW}}$, or $\mathcal{C}_{\mathrm{CWL}}$ such that $\pi \in \mathcal{H}_{\leq 0}$ and $\dim(\mathcal{H}_{\leq 0}) = m!$. We now prove that $\mathcal{H}$ is indeed active. Because $\pi$ is strictly positive and $\mathcal{H}_{\leq 0}$ is convex, $\mathcal{H}_{\leq 0}$ contains an interior point in $\mathbb{R}_{>0}^{m!}$, denoted by $\vec{x}$. Formally, let $\vec{x}'$ denote an arbitrary interior point of $\mathcal{H}_{\leq 0}$. It is not hard to verify that for some sufficiently small $\delta > 0$, $\vec{x} = \dfrac{\pi + \delta \vec{x}'}{1 + \delta} \in \mathbb{R}_{>0}^{m!}$ is an interior point of $\mathcal{H}_{\leq 0}$.

  Suppose $\mathcal{H}$ is characterized by $\mathbf{A}$ and $\vec{b}$. Then, we have $\mathbf{A} \cdot (\vec{x})^\top < \left(\vec{0}\right)^\top$. Therefore, there exists $\ell > 0$ such that $\mathbf{A} \cdot (\ell \vec{x})^\top \leq \left(\vec{b}\right)^\top$, which means that $\ell \vec{x} \in \mathcal{H} \cap \mathbb{R}_{>0}^{m!} \neq \emptyset$. By Claim 2 and the definition of $N_{\overline{r}}$ (Definition 24), $\mathcal{H}$ is active at every $n > N_{\overline{r}}$.

  Recall that in the YNN case we have $\pi \notin \Pi_{\mathcal{C}_{\mathrm{CWW}},n}$ and $\pi \notin \Pi_{\mathcal{C}_{\mathrm{CWL}},n}$. Therefore, $\mathcal{H} \subseteq \mathcal{C}_{\mathrm{NCW}}$, which means that $\dim_{\mathcal{C}_{\mathrm{NCW}},n}^{\max}(\pi) = m! = \dim_{\mathcal{C},n}^{\max}(\pi)$. Following a similar reasoning as in the "*YN" case, we have $\dim_{\mathcal{C}_{\mathrm{CWL}},n}^{\max}(\pi) = -\frac{\log n}{n}$.

- **NNN:** This case is impossible because as proved in the "YNN" case, for all $n > N_{\overline{r}}$, $\pi \notin \Pi_{\mathcal{C}_{\mathrm{CWW}},n}$ and $\pi \notin \Pi_{\mathcal{C}_{\mathrm{CWL}},n}$ implies that $\pi \in \Pi_{\mathcal{C}_{\mathrm{NCW}},n}$.

**Step 3: Apply Lemma 1.** In this step, we apply the inf part of Lemma 1 by combining and simplifying conditions in Table 4.

- **The 0 case** never holds when $n \geq m^4$, because any complete graph is the UMG of some $n$-profile [54, Claim 6 in the Appendix]. In particular, any complete graph where there is no Condorcet winner is the UMG of an $n$-profile.

- **The 1 case** holds if and only if $\overline{r}$ satisfies CC for all $n$ profile $P$, i.e. $\mathrm{C_{AS}}(\overline{r}, n)$.

- **The VU case.** According to the inf part of Lemma 1, the VU case holds if and only if $\beta_n = -\frac{n}{\log n}$. Note that we do not need to assume $\mathrm{C_{AS}}(\overline{r}, n)$ in the VU case. According to Table 4, $\beta_n = -\frac{n}{\log n}$ if and only if there exists $\pi \in \mathrm{CH}(\Pi)$ such that the "NNY" column holds. Recall that the "NNN" column is impossible for any $n > N_{\overline{r}}$. Therefore, the "NNY" column holds for $\pi \in \mathrm{CH}(\Pi)$ if and only if $\pi \notin \Pi_{\mathcal{C}_{\mathrm{NCW}},n}$ and $\pi \notin \Pi_{\mathcal{C}_{\mathrm{CWW}},n}$, which is equivalent to the following condition by Claim 6

$$\pi \notin \Pi_{\mathcal{C}_{\mathrm{NCW}},n} \text{ and } \pi \notin \mathrm{Closure}(\mathcal{R}_{\mathrm{CWW}}^{\overline{r}}) \tag{9}$$

  Next, we simplify (9) for $2 \mid n$ and $2 \nmid n$, respectively.

  - **$2 \mid n$.** By the $2 \mid n$ part of Claim 3, $\pi \notin \Pi_{\mathcal{C}_{\mathrm{NCW}},n}$ if and only if $\pi$ has a Condorcet winner. We prove that in this case (9) is equivalent to:

$$\mathrm{CW}(\pi) \cap (\mathcal{A} \setminus \overline{r}(\pi)) \neq \emptyset \tag{10}$$

    **(9)$\Rightarrow$(10).** Suppose $\pi$ has a Condorcet winner, denoted by $a$, and (9) holds. For the sake of contradiction suppose that (10) does not hold, which means that $a \in \overline{r}(\pi)$. Then, following a similar construction as in the proof of Claim 6, the minimal continuity of $\overline{r}$ implies that there exist $\vec{t}_a \in \mathcal{S}_{\vec{H}}^{\circ}$ with $\vec{t}_a \trianglelefteq \mathrm{Sign}_{\vec{H}}(\pi)$ and $\overline{r}(\vec{t}_a) = \{a\}$, and

$\vec{x} \in \mathcal{H}^{\vec{t}_a}$ such that for every $\delta > 0$ we have $\pi + \delta \vec{x} \in \mathcal{R}^{\overline{r}}_{\text{CWW}}$. Then $(\pi + \vec{x}, \pi + \frac{1}{2}\vec{x}, \ldots)$ converges to $\pi$, which contradicts the assumption that $\pi \notin \text{Closure}(\mathcal{R}^{\overline{r}}_{\text{CWW}})$.

**(10)$\Rightarrow$(9).** Let $a \in \text{CW}(\pi) \cap (\mathcal{A} \setminus \overline{r}(\pi))$, which means that $\{a\} = \text{CW}(\pi)$ and $a \notin \overline{r}(\pi)$. Suppose for the sake of contradiction that (9) does not hold. Due to Claim 3, we have $\pi \notin \Pi_{\mathcal{C}_{\text{NCW}},n}$. Therefore, $\pi \in \text{Closure}(\mathcal{R}^{\overline{r}}_{\text{CWW}})$. This means that there exists a sequence $(\vec{x}_1, \vec{x}_2, \ldots)$ in $\mathcal{R}^{\overline{r}}_{\text{CWW}}$ that converge to $\pi$. It follows that there exists $j^* \in \mathbb{N}$ such that for all $j > j^*$, $a$ is the Condorcet winner under $\vec{x}_j$, which means that $a \in \overline{r}(\vec{x}_j)$ because $\vec{x}_j \in \mathcal{R}^{\overline{r}}_{\text{CWW}}$. Therefore, by the continuity of $\overline{r}$, we have $a \in \overline{r}(\pi)$, which means that $\text{CW}(\pi) \cap (\mathcal{A} \setminus \overline{r}(\pi)) = \emptyset$. This is a contradiction to (10).

Therefore, when $2 \mid n$, the VU case holds if and only if there exists $\pi \in \text{CH}(\Pi)$ such that (10) holds, which is as described in the statement of the theorem, i.e.

$$\exists \pi \in \text{CH}(\Pi) \text{ s.t. } \text{C}_{\text{RD}}(\overline{r}, \pi)$$

– **$2 \nmid n$.** By the $2 \nmid n$ part of Claim 3, $\pi \notin \Pi_{\mathcal{C}_{\text{NCW}},n}$ is equivalent to $\text{CW}(\pi) \cup \text{ACW}(\pi) \neq \emptyset$, i.e. either $\text{CW}(\pi) \neq \emptyset$ or $\text{ACW}(\pi) \neq \emptyset$. When $\text{CW}(\pi) \neq \emptyset$, as in the $2 \mid n$ case, (9) becomes (10). When $\text{ACW}(\pi) \neq \emptyset$, (9) becomes $\text{C}_{\text{NRS}}(\overline{r}, \pi) = 1$. Therefore, when $2 \nmid n$ the VU case holds if and only if the condition in the statement of the theorem holds, i.e.

$$\exists \pi \in \text{CH}(\Pi) \text{ s.t. } \text{C}_{\text{RD}}(\overline{r}, \pi) \text{ or } \text{C}_{\text{NRS}}(\overline{r}, \pi)$$

- **The U case.** According to the inf part of Lemma 1, the U case holds if and only if $0 \leq \beta_n < m!$. According to Table 4, $0 \leq \beta_n < m!$ if and only if

  **(i)** for every $\pi \in \text{CH}(\Pi)$ the NNY column of Table 4 does not hold, and

  **(ii)** there exists $\pi \in \text{CH}(\Pi)$ such that the YNY column of Table 4 holds and $\dim^{\max}_{\mathcal{C}_{\text{NCW}},n}(\pi) < m!$.

Part (ii) can be simplified as follows. By Claim 3, $\dim^{\max}_{\mathcal{C}_{\text{NCW}},n}(\pi) < m!$ if and only if $2 \mid n$ and $\text{ACW}(\pi) \neq \emptyset$, and in this case $\dim^{\max}_{\mathcal{C}_{\text{NCW}},n}(\pi) = m! - 1$. We show that it suffices to additionally require that $\pi \notin \Pi_{\mathcal{C}_{\text{CWW}},n}$ (i.e. the "N"), or in other words, given $\dim^{\max}_{\mathcal{C}_{\text{NCW}},n}(\pi) = m! - 1$, $\pi \notin \Pi_{\mathcal{C}_{\text{CWW}},n}$ implies $\pi \in \Pi_{\mathcal{C}_{\text{CWL}},n}$ (i.e. the second "Y"). Suppose for the sake of contradiction that $\dim^{\max}_{\mathcal{C}_{\text{NCW}},n}(\pi) = m! - 1$, $\pi \notin \Pi_{\mathcal{C}_{\text{CWW}},n}$, and $\pi \notin \Pi_{\mathcal{C}_{\text{CWL}},n}$. Notice that this corresponds to the "YNN" column in Table 4, which means that $\dim^{\max}_{\mathcal{C}_{\text{NCW}},n}(\pi) = m!$, which is a contradiction. By Claim 6, $\pi \notin \Pi_{\mathcal{C}_{\text{CWW}},n}$ if and only if $\pi \notin \text{Closure}(\mathcal{R}^{\overline{r}}_{\text{CWW}})$. Therefore, part (ii) is equivalent to

$$\exists \pi \in \text{CH}(\Pi) \text{ s.t. } \text{C}_{\text{NRS}}(\overline{r}, \pi)$$

Summing up, the U case holds if and only if the condition in the statement of the theorem holds, i.e.

$$2 \mid n, \text{ and (1) } \forall \pi \in \text{CH}(\Pi), \neg\text{C}_{\text{RD}}(\overline{r}, \pi), \text{ and (2) } \exists \pi \in \text{CH}(\Pi) \text{ s.t. } \text{C}_{\text{NRS}}(\overline{r}, \pi)$$

- **The L case** never holds when $n \geq m^4$, because according to Table 4, $\alpha^*_n = \max_{\pi \in \text{CH}(\Pi)} \dim^{\max}_{\mathcal{C}_{\text{CWL}},n}(\pi)$ is either $-\frac{n}{\log n}$ or $m!$, which means that it is never in $[0, m!)$.

- **The VL case.** According to the inf part of Lemma 1, the VL case holds if and only if the 1 case does not hold and $\alpha^*_n = -\frac{n}{\log n}$. According to Table 4, this happens in the "$*$YN" column or the "YNN" column, which is equivalent to only requiring that the last "N" holds (because "NNN" is impossible), i.e. for all $\pi \in \text{CH}(\Pi)$, $\pi \notin \Pi_{\mathcal{C}_{\text{CWL}},n}$. By Claim 6, the VL case holds if and only if if and only if the condition in the statement of the theorem holds, i.e.

$$\neg\text{C}_{\text{AS}}(\overline{r}, n) \text{ and } \forall \pi \in \text{CH}(\Pi), \text{C}_{\text{RS}}(\overline{r}, \pi)$$

- **The M case** corresponds to the remaining cases.

**Proof for general refinement $r$ of $\bar{r}$.** We now turn to the proof of the theorem for an arbitrary refinement of $\bar{r}$, denoted by $r$. We first define a slightly different version of CC, denoted by $\mathrm{CC}^*$, which will be used as the lower bound on the (semi-random) satisfaction of the regular CC. For any GISR $\bar{r}$ and any profile $P$, we define

$$\mathrm{CC}^*(\bar{r}, P) = \begin{cases} 1 & \text{if } \mathrm{CW}(P) = \emptyset \text{ or } \mathrm{CW}(P) = \bar{r}(P) \\ 0 & \text{otherwise} \end{cases}$$

In words, $\mathrm{CC}^*(\bar{r}, P) =$ if and only if (1) the Condorcet winner does not exist, or (2) the Condorcet winner exists and is the *unique* winner under $P$ according to $\bar{r}$. Compared to the standard Condorcet criterion CC, $\mathrm{CC}^*$ rules out profiles $P$ where a Condorcet winner exists and is not the unique winner. $\mathrm{CC}^*$ and CC coincide with each other when $\bar{r}$ is a resolute rule. Because for any profile $P$ we have $r(P) \subseteq \bar{r}(P)$, for any $\vec{\pi} \in \Pi^n$ we have

$$\Pr_{P \sim \vec{\pi}}(\mathrm{CC}^*(\bar{r}, P) = 1) \leq \Pr_{P \sim \vec{\pi}}(\mathrm{CC}(r, P) = 1) \leq \Pr_{P \sim \vec{\pi}}(\mathrm{CC}(\bar{r}, P) = 1)$$

Therefore,

$$\widetilde{\mathrm{CC}^*}_{\Pi}^{\min}(\bar{r}, n) \leq \widetilde{\mathrm{CC}}_{\Pi}^{\min}(r, n) \leq \widetilde{\mathrm{CC}}_{\Pi}^{\min}(\bar{r}, n) \tag{11}$$

n order to prove the theorem, it suffices to prove that the lower bound in (11), i.e., $\widetilde{\mathrm{CC}^*}_{\Pi}^{\min}(\bar{r}, n)$, has the same dichotomous characterization as $\widetilde{\mathrm{CC}}_{\Pi}^{\min}(\bar{r}, n)$. To this end, we first define a union of polyhedra, denoted by $\mathcal{C}'$, and its almost complement $\mathcal{C}'_{\mathrm{CWL}}$ that are similar to Definition 22 as follows.

**Definition 25** ($\mathcal{C}'$ and $\mathcal{C}'_{\mathrm{CWL}}$). *Given an int-GISR characterized by $\vec{H}$ and $g$, we define*

$$\mathcal{C}' = \mathcal{C}_{\mathrm{NCW}} \cup \mathcal{C}'_{\mathrm{CWW}}, \quad \text{where } \mathcal{C}'_{\mathrm{CWW}} = \bigcup_{a \in \mathcal{A}, \vec{t} \in \mathcal{S}_{\vec{H}}: \bar{r}(\vec{t}) = \{a\}} \mathcal{H}^{a, \vec{t}}$$

$$\mathcal{C}'_{\mathrm{CWL}} = \bigcup_{a \in \mathcal{A}, \vec{t} \in \mathcal{S}_{\vec{H}}: \bar{r}(\vec{t}) \neq \{a\}} \mathcal{H}^{a, \vec{t}}$$

Notice that $\mathcal{C}_{\mathrm{NCW}}$ used in Definition 25 was define in Definition 22. Just like $\mathcal{C}_{\mathrm{CWL}}$ is an almost complement of $\mathcal{C}$, $\mathcal{C}'_{\mathrm{CWL}}$ is an almost complement of $\mathcal{C}'$. Formally, we first note that $\mathcal{C}' \cap \mathcal{C}'_{\mathrm{CWL}} = \emptyset$, and for any integer vector $\vec{x}$,

- if $\vec{x}$ does not have a Condorcet winner then $\vec{x} \in \mathcal{C}_{\mathrm{NCW}} \subseteq \mathcal{C}'$;

- if $\vec{x}$ has a Condorcet winner $a$, which is the unique $\bar{r}$ winner, then $\vec{x} \in \mathcal{H}^{a, \mathrm{Sign}_{\vec{H}}(\vec{x})} \subseteq \mathcal{C}'_{\mathrm{CWW}} \subseteq \mathcal{C}$;

- otherwise $\vec{x}$ has a Condorcet winner $a$, which is either not a $\bar{r}$ co-winner or $|\bar{r}(\vec{x})| \geq 2$. In both cases $\vec{x} \in \mathcal{H}^{a, \mathrm{Sign}_{\vec{H}}(\vec{x})} \subseteq \mathcal{C}'_{\mathrm{CWL}}$.

Therefore, $\mathbb{Z}^q \subseteq \mathcal{C}' \cup \mathcal{C}'_{\mathrm{CWL}}$. The proof for $\widetilde{\mathrm{CC}^*}_{\Pi}^{\min}(\bar{r}, n)$ is similar to the proof for $\widetilde{\mathrm{CC}}_{\Pi}^{\min}(\bar{r}, n)$ presented earlier. The main difference is that $\mathcal{C}$, $\mathcal{C}_{\mathrm{CWW}}$, and $\mathcal{C}_{\mathrm{CWL}}$ are replaced by $\mathcal{C}'$, $\mathcal{C}'_{\mathrm{CWW}}$, and $\mathcal{C}'_{\mathrm{CWL}}$, respectively. The key part is to prove a counterpart to Table 4, which follows after proving $\Pi_{\mathcal{C}'_{\mathrm{CWW}}, n} = \Pi_{\mathcal{C}_{\mathrm{CWW}}, n}$ and $\Pi_{\mathcal{C}'_{\mathrm{CWL}}, n} = \Pi_{\mathcal{C}_{\mathrm{CWL}}, n}$ for every $n > N_{\bar{r}}$, as formally shown in the following claim.

**Claim 8.** *For any $n > N_{\bar{r}}$, we have $\Pi_{\mathcal{C}'_{CWW}, n} = \Pi_{\mathcal{C}_{CWW}, n}$ and $\Pi_{\mathcal{C}'_{CWL}, n} = \Pi_{\mathcal{C}_{CWL}, n}$.*

*Proof.* The main difference between $\mathcal{C}'_{\mathrm{CWW}}$ (respectively, $\mathcal{C}'_{\mathrm{CWL}}$) and $\mathcal{C}_{\mathrm{CWW}}$ (respectively, $\mathcal{C}_{\mathrm{CWL}}$) is the memberships of polyhedra $\mathcal{H}^{a, \vec{t}}$, where $a \in \bar{r}(\vec{t})$ and $\bar{r}(\vec{t}) \geq 2$. Therefore, to prove the claim, it suffices to show that the membership of $\mathcal{H}^{a, \vec{t}}$ does not affect $\Pi_{\mathcal{C}'_{\mathrm{CWW}}, n}$ (respectively, $\Pi_{\mathcal{C}'_{\mathrm{CWL}}, n}$) compared to $\Pi_{\mathcal{C}_{\mathrm{CWW}}, n}$ (respectively, $\Pi_{\mathcal{C}_{\mathrm{CWL}}, n}$).

It suffices to show that for any polyhedron $\mathcal{H}^{a, \vec{t}}$, where $a \in \bar{r}(\vec{t})$ and $\bar{r}(\vec{t}) \geq 2$, for any $\pi \in \mathrm{CH}(\Pi)$ and any $n > N_{\bar{r}}$, if $\mathcal{H}^{a, \vec{t}}$ is active and $\pi \in \mathcal{H}^{a, \vec{t}}_{\leq 0}$, then there exist $\mathcal{H}^{a, \vec{t}_a}_{\leq 0} \subseteq \mathcal{C}_{\mathrm{CWW}} \cap \mathcal{C}'_{\mathrm{CWW}}$ and $\mathcal{H}^{a, \vec{t}_b}_{\leq 0} \subseteq \mathcal{C}_{\mathrm{CWL}} \cap \mathcal{C}'_{\mathrm{CWL}}$ such that (1) $\mathcal{H}^{a, \vec{t}_a}_{\leq 0}$ and $\mathcal{H}^{a, \vec{t}_b}_{\leq 0}$ are active at $n$, and (2) $\pi \in \mathcal{H}^{a, \vec{t}_a}_{\leq 0} \cap \mathcal{H}^{a, \vec{t}_b}_{\leq 0}$.

In other words, if a distribution $\pi \in \mathrm{CH}(\Pi)$ is in $\mathcal{C}'_{\mathrm{CWW}}$, $\mathcal{C}'_{\mathrm{CWL}}$, $\mathcal{C}_{\mathrm{CWW}}$, or $\mathcal{C}_{\mathrm{CWL}}$ due to $\mathcal{H}^{a,\vec{t}}$, then it is also in the same set without considering its edge to $\mathcal{H}^{a,\vec{t}}$ in the activation graph. As we will see soon, (1) follows after the assumption that $n > N_{\overline{r}}$ and (2) follows after the minimal continuity of $\overline{r}$. Formally, the proof proceeds in the following three steps.

(i) **Define $\vec{t}_a$ and $\vec{t}_b$.** Let $b \neq a$ denote a co-winner under $\pi$, i.e., $\{a, b\} \subseteq \overline{r}(\pi)$. Because $\overline{r}$ is minimally continuous, by Proposition 4, there exists a feasible atomic signature $\vec{t}_a \in \mathcal{S}^{\circ}_{\vec{H}}$ (respectively, $\vec{t}_b \in \mathcal{S}^{\circ}_{\vec{H}}$) such that $\vec{t}_a \trianglelefteq \vec{t}$ (respectively, $\vec{t}_b \trianglelefteq \vec{t}$) and $\overline{r}(\vec{t}_a) = \{a\}$ (respectively, $\overline{r}(\vec{t}_b) = \{b\}$).

(ii) **Prove that $\mathcal{H}^{a,\vec{t}_a}_{\leq 0}$ and $\mathcal{H}^{a,\vec{t}_b}_{\leq 0}$ are active at any $n > N_{\overline{r}}$.** Because $\vec{t}_a$ is feasible, there exists $\vec{x} \in \mathbb{R}^{m!}$ such that $\mathrm{Sign}_{\vec{H}}(\vec{x}) = \vec{t}_a$. Therefore, recall that $\pi$ is strictly positive (by $\epsilon$), for some sufficiently small $\delta > 0$, we have $\pi + \delta\vec{x} \in \mathbb{R}^{m!}_{>0}$, $\mathrm{CW}(\pi + \delta\vec{x}) = \{a\}$, and $\mathrm{Sign}_{\vec{H}}(\pi + \delta\vec{x}) = \vec{t}_a$. This means that $\pi + \delta\vec{x}$ is an interior point of $\mathcal{H}^{a,\vec{t}_a}$ (which also means that $\dim(\mathcal{H}^{a,\vec{t}_a}) = m!$). Recall that the $\vec{b}$ part of $\mathcal{H}^{a,\vec{t}_a}$ (Definition 17 and 21) is non-positive, we have $\mathcal{H}^{a,\vec{t}_a} \subseteq \mathcal{H}^{a,\vec{t}_a}_{\leq 0}$, which means that $\dim(\mathcal{H}^{a,\vec{t}_a}_{\leq 0}) = m!$ as well. Therefore, according to Claim 2 and the definition of $N_{\overline{r}}$ (Definition 24), $\mathcal{H}^{a,\vec{t}_a}$ is active at any $n > N_{\overline{r}}$. Similarly, we have that $\mathcal{H}^{a,\vec{t}_b}$ is active at any $n > N_{\overline{r}}$.

(iii) **Prove that $\pi \in \mathcal{H}^{a,\vec{t}_a}_{\leq 0} \cap \mathcal{H}^{a,\vec{t}_b}_{\leq 0}$.** Recall that $\pi \in \mathcal{H}^{a,\vec{t}}_{\leq 0}$. Therefore, according to Claim 7 (ii), we have $\vec{t} \trianglelefteq \mathrm{Sign}_{\vec{H}}(\pi)$, which means that $\vec{t}_a \trianglelefteq \mathrm{Sign}_{\vec{H}}(\pi)$, because $\vec{t}_a \trianglelefteq \vec{t}$. By Claim 7 (ii) again, we have $\pi \in \mathcal{H}^{a,\vec{t}_a}_{\leq 0}$. Similarly, we can prove that $\pi \in \mathcal{H}^{a,\vec{t}_b}_{\leq 0}$.

This completes the proof of Claim 8. $\qquad\qquad\square$

Therefore, $\widetilde{\mathrm{CC}^*}^{\mathrm{min}}_{\Pi}(\overline{r}, n)$ has the same characterization as $\widetilde{\mathrm{CC}}^{\mathrm{min}}_{\Pi}(\overline{r}, n)$, which concludes the proof of Lemma 2 due to (11). $\qquad\qquad\square$

## E.2 Proof of Theorem 1

**Theorem 1. (Semi-random CC: Integer Positional Scoring Rules).** *Let $\mathcal{M} = (\Theta, \mathcal{L}(\mathcal{A}), \Pi)$ be a strictly positive and closed single-agent preference model, let $\overline{r}_{\vec{s}}$ be a minimally continuous int-GISR and let $r_{\vec{s}}$ be a refinement of $\overline{r}_{\vec{s}}$. For any $n \geq 8m + 49$ with $2 \mid n$, we have*

$$\widetilde{\mathrm{CC}}^{\mathrm{min}}_{\Pi}(r_{\vec{s}}, n) = \begin{cases} 1 - \exp(-\Theta(n)) & \textit{if } \forall \pi \in CH(\Pi), |WCW(\pi)| \times |\overline{r}(\pi) \cup WCW(\pi)| \leq 1 \\ \Theta(n^{-0.5}) & \textit{if } \begin{cases} \textit{(1) } \forall \pi \in CH(\Pi), CW(\pi) \cap (\mathcal{A} \setminus \overline{r}_{\vec{s}}(\pi)) = \emptyset \textit{ and} \\ \textit{(2) } \exists \pi \in CH(\Pi) \textit{ s.t. } |ACW(\pi) \cap (\mathcal{A} \setminus \overline{r}_{\vec{s}}(\pi))| = 2 \end{cases} \\ \exp(-\Theta(n)) & \textit{if } \exists \pi \in CH(\Pi) \textit{ s.t. } CW(\pi) \cap (\mathcal{A} \setminus \overline{r}_{\vec{s}}(\pi)) \neq \emptyset \\ \Theta(1) \textit{ and } 1 - \Theta(1) & \textit{otherwise} \end{cases}$$

*For any $n \geq 8m + 49$ with $2 \nmid n$, we have*

$$\widetilde{\mathrm{CC}}^{\mathrm{min}}_{\Pi}(r_{\vec{s}}, n) = \begin{cases} 1 - \exp(-\Theta(n)) & \textit{same as the } 2 \mid n \textit{ case} \\ \exp(-\Theta(n)) & \textit{if } \exists \pi \in CH(\Pi) \textit{ s.t. } \begin{cases} \textit{(1) } CW(\pi) \cap (\mathcal{A} \setminus \overline{r}_{\vec{s}}(\pi)) \neq \emptyset \textit{ or} \\ \textit{(2) } |ACW(\pi) \cap (\mathcal{A} \setminus \overline{r}_{\vec{s}}(\pi))| = 2 \end{cases} \\ \Theta(1) \textit{ and } 1 - \Theta(1) & \textit{otherwise} \end{cases}$$

*Proof.* We apply Lemma 2 to prove the theorem. For any integer irresolute positional scoring rule $\overline{r}_{\vec{s}}$, we prove the following claim to simplify $\mathrm{Closure}(\mathcal{R}^{\overline{r}_{\vec{s}}}_{\mathrm{CWW}})$ and $\mathrm{Closure}(\mathcal{R}^{\overline{r}_{\vec{s}}}_{\mathrm{CWL}})$.

**Claim 9.** *For any $\pi \in CH(\Pi)$,*

$$\left[\pi \in \mathit{Closure}(\mathcal{R}^{\overline{r}_{\vec{s}}}_{CWW})\right] \Leftrightarrow \left[WCW(\pi) \cap \overline{r}_{\vec{s}}(\pi) \neq \emptyset\right]$$

$$\left[\pi \in \mathit{Closure}(\mathcal{R}^{\overline{r}_{\vec{s}}}_{CWL})\right] \Leftrightarrow \left[\exists a \neq b \textit{ s.t. } a \in WCW(\pi) \textit{ and } b \in \overline{r}_{\vec{s}}(\pi)\right]$$

*Proof.* The proof is done in the following steps.

$\left[\pi \in \textbf{Closure}(\mathcal{R}_{\text{CWW}}^{\bar{r}_{\vec{s}}})\right] \Rightarrow [\textbf{WCW}(\pi) \cap \bar{r}_{\vec{s}}(\pi) \neq \emptyset].$ Suppose $\pi \in \text{Closure}(\mathcal{R}_{\text{CWW}}^{\bar{r}_{\vec{s}}})$, which means that there exists a sequence $(\vec{x}_1, \vec{x}_2, \ldots)$ in $\mathcal{R}_{\text{CWW}}^{\bar{r}_{\vec{s}}}$ that converges to $\pi$. It follows that there exists an alternative $a \in \mathcal{A}$ and a subsequence of $(\vec{x}_1, \vec{x}_2, \ldots)$, denoted by $(\vec{x}_1', \vec{x}_2', \ldots)$ such that for every $j \in \mathbb{N}$, $\text{CW}(\vec{x}_j') = \{a\}$ and $a \in \bar{r}_{\vec{s}}(\vec{x}_j')$. This means that the following holds.

- $a$ is a weak Condorcet winner under $\pi$. Notice that for any $b \neq a$ and any $j \in \mathbb{N}$, we have $\text{Pair}_{b,a} \cdot \vec{x}_j' < 0$, which means that $\text{Pair}_{b,a} \cdot \pi \leq 0$.

- $a \in \bar{r}_{\vec{s}}(\pi)$. Notice that for any $b \neq a$ and any $j \in \mathbb{N}$, the total score of $a$ is higher than or equal to the total score of $b$ in $\vec{x}_j'$. Therefore, the same holds for $\pi$, which means that $a \in \bar{r}_{\vec{s}}(\pi)$.

Therefore, $a$ is a weak Condorcet winner as well as a $\bar{r}_{\vec{s}}$ co-winner, which implies $\text{WCW}(\pi) \cap \bar{r}_{\vec{s}}(\pi) \neq \emptyset$.

$\left[\pi \in \textbf{Closure}(\mathcal{R}_{\text{CWW}}^{\bar{r}_{\vec{s}}})\right] \Leftarrow [\textbf{WCW}(\pi) \cap \bar{r}_{\vec{s}}(\pi) \neq \emptyset].$ Suppose $\text{WCW}(\pi) \cap \bar{r}_{\vec{s}}(\pi) \neq \emptyset$ and let $a \in \text{WCW}(\pi) \cap \bar{r}_{\vec{s}}(\pi)$. We will explicitly construct a sequence of vectors in $\mathcal{R}_{\text{CWW}}^{\bar{r}_{\vec{s}}}$ that converges to $\pi$. Let $\sigma$ denote a cyclic permutation among $\mathcal{A} \setminus \{a\}$ and let $P$ denote the following $(m-1)$-profile

$$P = \{\sigma^i(a \succ \text{others}) : 1 \leq i \leq m-1\} \tag{12}$$

It is not hard to verify that $\text{CW}(P) = \bar{r}_{\vec{s}}(P) = \{a\}$. Therefore, for any $\delta > 0$ we have

$$\text{CW}(\pi + \delta \cdot \text{Hist}(P)) = \bar{r}_{\vec{s}}(\pi + \delta \cdot \text{Hist}(P)) = \{a\},$$

which means that $\pi + \delta \cdot \text{Hist}(P) \in \text{Closure}(\mathcal{R}_{\text{CWW}}^{\bar{r}_{\vec{s}}})$. It follows that $(\pi + \frac{1}{j}\text{Hist}(P) : j \in \mathbb{N})$ is a sequence in $\text{Closure}(\mathcal{R}_{\text{CWW}}^{\bar{r}_{\vec{s}}})$ that converges to $\pi$, which means that $\pi \in \text{Closure}(\mathcal{R}_{\text{CWW}}^{\bar{r}_{\vec{s}}})$.

$\left[\pi \in \textbf{Closure}(\mathcal{R}_{\text{CWL}}^{\bar{r}_{\vec{s}}})\right] \Rightarrow [\exists a \neq b \textbf{ s.t. } a \in \textbf{WCW}(\pi) \textbf{ and } b \in \bar{r}_{\vec{s}}(\pi)].$ Suppose $\pi \in \text{Closure}(\mathcal{R}_{\text{CWL}}^{\bar{r}_{\vec{s}}})$, which means that there exists a sequence $(\vec{x}_1, \vec{x}_2, \ldots)$ in $\mathcal{R}_{\text{CWL}}^{\bar{r}_{\vec{s}}}$ that converges to $\pi$. It follows that there exists a pair of different alternatives $a, b \in \mathcal{A}$ and a subsequence of $(\vec{x}_1, \vec{x}_2, \ldots)$, denoted by $(\vec{x}_1', \vec{x}_2', \ldots)$ such that for every $j \in \mathbb{N}$, $\text{CW}(\vec{x}_j') = \{a\}$ and $b \in \bar{r}_{\vec{s}}(\vec{x}_j')$. Following a similar proof as in the $\mathcal{R}_{\text{CWL}}^{\bar{r}_{\vec{s}}}$ part, we have that $a$ is a weak Condorcet winner under $\pi$ and $b \in \bar{r}_{\vec{s}}(\pi)$.

$\left[\pi \in \textbf{Closure}(\mathcal{R}_{\text{CWL}}^{\bar{r}_{\vec{s}}})\right] \Leftarrow [\exists a \neq b \textbf{ s.t. } a \in \textbf{WCW}(\pi) \textbf{ and } b \in \bar{r}_{\vec{s}}(\pi)].$ Let $a \neq b$ be two alternatives such that $a \in \text{WCW}(\pi)$ and $b \in \bar{r}_{\vec{s}}(\pi)$. We define a profile $P$ where $\text{CW}(P) = \{a\}$ and $\bar{r}_{\vec{s}}(P) = \{b\}$, whose existence is guaranteed by the following claim, which is slightly different from [18, Theorem 6] for scoring vectors $\vec{s} = (s_1, \ldots, s_m)$ with $s_1 > s_2 > \cdots > s_m$.

**Claim 10.** *For any $m \geq 3$, any positional scoring rule with scoring vector $\vec{s} = (s_1, \ldots, s_m)$ where $s_1 > s_m$, any $n \geq 8m + 49$, and any pair of different alternatives $a \neq b$, there exists an $n$-profile $P$ such that $\text{CW}(P) = \{a\}$ and $\bar{r}_{\vec{s}}(P) = \{b\}$.*

*Proof.* We explicitly construct an $n$-profile $P$ where the Condorcet winner exists and is different from the unique $\bar{r}_{\vec{s}}$ winner. Then, we apply a permutation over $\mathcal{A}$ to $P$ to make $a$ the Condorcet and $b$ the unique $\bar{r}_{\vec{s}}$ winner. The construction is done in two cases: $s_2 = s_m$ and $s_2 > s_m$.

- **Case 1: $s_2 = s_m$.** In this case $\bar{r}_{\vec{s}}$ corresponds to the plurality rule. We let

$$P = \left\lfloor \frac{n-1}{2} \right\rfloor \times [2 \succ 1 \succ 3 \succ \text{others}] + \left\lfloor \frac{n-3}{2} \right\rfloor \times [3 \succ 1 \succ 2 \succ \text{others}]$$

$$+ \left(n + 1 - 2\left\lfloor \frac{n-1}{2} \right\rfloor\right) \times [1 \succ 2 \succ 3 \succ \text{others}]$$

It is not hard to verify that the alternative 1 is the Condorcet winner and 2 is the unique plurality winner.

- **Case 2: $s_2 > s_m$.** Let $2 \le k \le m - 1$ denote the smallest number such that $s_k > s_{k+1}$. Let $A_1 = [4 \succ \cdots \succ k+1]$ and $A_2 = [k+2 \succ \cdots \succ m]$, and let $P^*$ denote the following 7-profile.

$$P^* = \{3 \times [1 \succ 2 \succ A_1 \succ 3 \succ A_2] + 2 \times [2 \succ 3 \succ A_1 \succ 1 \succ A_2]$$
$$+ [3 \succ 1 \succ A_1 \succ 2 \succ A_2] + [2 \succ 1 \succ A_1 \succ 3 \succ A_2]\}$$

It is not hard to verify that 1 is the Condorcet winner under $P^*$, and the total score of 1 is $3s_1 + 2s_2 + 2s_{k+1} < 3s_1 + 3s_2 + s_{k+1}$, which is the total score of 2. Note that the total score of any alternative in $A_1$ is $7s_k$, which might be larger than the score of 2. If $3s_1 + 3s_2 + s_{k+1} \ge 7s_k$, then we let $b = 2$; otherwise we let $b = 4$. Let $P_b$ denote the following $(m-1)$-profile that will be used as a tie-breaker. Let $\sigma$ denote an arbitrary cyclic permutation among $\mathcal{A} \setminus \{b\}$.

$$P_b = \{\sigma^i([b \succ \text{others}]) : 1 \le i \le m - 1\}$$

Let

$$P = \left\lfloor \frac{n - m + 1}{7} \right\rfloor \times P^* + P_b + \left(n - m + 1 - 7 \left\lfloor \frac{n - m + 1}{7} \right\rfloor\right) \times [b \succ \text{others}]$$

It is not hard to verify that when $n \ge 8m + 49$, $\mathrm{CW}(P) = \{1\}$, $\overline{r}_{\vec{s}}(P) = \{b\}$, and $b \ne 1$.

This proves Claim 10. $\qquad\square$

Let $P$ denote the profile guaranteed by Claim 10. For any $\delta > 0$ we have

$$\mathrm{CW}(\pi + \delta \cdot \mathrm{Hist}(P)) = \{a\} \text{ and } \overline{r}(\pi + \delta \cdot \mathrm{Hist}(P)) = \{b\},$$

which means that $\pi + \delta \cdot \mathrm{Hist}(P) \in \mathrm{Closure}(\mathcal{R}_{\mathrm{CWL}}^{\overline{r}_{\vec{s}}})$. It follows that $(\pi + \frac{1}{j}\mathrm{Hist}(P) : j \in \mathbb{N})$ is a sequence in $\mathcal{R}_{\mathrm{CWL}}^{\overline{r}_{\vec{s}}}$ that converges to $\pi$, which means that $\pi \in \mathrm{Closure}(\mathcal{R}_{\mathrm{CWL}}^{\overline{r}_{\vec{s}}})$. This proves Claim 9. $\qquad\square$

Claim 9 implies that for all $n \ge 8m + 49$, the 1 case doe not hold, i.e., $\mathrm{C_{AS}}(\overline{r}_{\vec{s}}, n) = 0$. We now apply Claim 9 to simplify the conditions in Lemma 2.

- $\mathrm{C_{RS}}(\overline{r}_{\vec{s}}, \pi)$. By definition, this holds if and only if $\pi \notin \mathrm{Closure}(\mathcal{R}_{\mathrm{CWL}}^{\overline{r}_{\vec{s}}})$, which is equivalent to $\nexists a \ne b$ s.t. $a \in \mathrm{WCW}(\pi)$ and $b \in \overline{r}_{\vec{s}}(\pi)$. In other words, either $\mathrm{WCW}(\pi) = \emptyset$ or $(\mathrm{WCW}(\pi) = \overline{r}_{\vec{s}}(\pi)$ and $|\mathrm{WCW}(\pi)| = 1)$. Notice that $\overline{r}_{\vec{s}}(\pi) \ne \emptyset$. Therefore, $\mathrm{C_{RS}}(\overline{r}_{\vec{s}}, \pi)$ is equivalent to $|\mathrm{WCW}(\pi)| \times |\overline{r}_{\vec{s}}(\pi) \cup \mathrm{WCW}(\pi)| \le 1$.

- $\mathrm{C_{NRS}}(\overline{r}_{\vec{s}}, \pi)$. By definition, this holds if and only if $\mathrm{ACW}(\pi) \ne \emptyset$ and $\pi \notin \mathrm{Closure}(\mathcal{R}_{\mathrm{CWW}}^{\overline{r}_{\vec{s}}})$, which is equivalent to $\mathrm{ACW}(\pi) \ne \emptyset$ and $\mathrm{WCW}(\pi) \cap \overline{r}_{\vec{s}}(\pi) = \emptyset$. The latter is equivalent to $\mathrm{WCW}(\pi) \cap (\mathcal{A} \setminus \overline{r}_{\vec{s}}(\pi)) = \mathrm{WCW}(\pi)$. We note that when $\mathrm{ACW}(\pi) \ne \emptyset$, we have $\mathrm{WCW}(\pi) = \mathrm{ACW}(\pi)$. Therefore, $\mathrm{C_{NRS}}(\overline{r}_{\vec{s}}, \pi)$ is equivalent to $|\mathrm{ACW}(\pi) \cap (\mathcal{A} \setminus \overline{r}_{\vec{s}}(\pi))| = 2$.

Theorem 1 follows after Lemma 2 with the simplified conditions discussed above. $\qquad\square$

### E.3 Definitions, Full Statement, and Proof for Theorem 2

For any $O \in \mathcal{L}(\mathcal{A})$, any $1 \le i < i' \le m$, and any $a \in \mathcal{A}$, let $O[i]$ denote the alternative ranked at the $i$-th place in $O$, let $O[i, i']$ denote the set of alternatives ranked from the $i$-th place to the $i'$-th place in $O$, and let $O^{-1}[a]$ denote the rank of $a$ in $O$. For any $A \subseteq \mathcal{A}$ and any $\vec{x} \in \mathbb{R}^{m!}$ that represents the histogram of a profile, let $\vec{x}|_A \in \mathbb{R}^{|A|!}$ denote the histogram of the profile restricted to alternatives in $A$.

**Example 11.** *Let $O = [3 \rhd 1 \rhd 2].$*[1] *We have $O[2] = 1$, $O^{-1}(2) = 3$, and $O[2,3] = \{1,2\}$. Let $\hat{\pi}$ denote the (fractional) profile in Figure 1. We have $\hat{\pi}|_{O[2,3]} = (\underbrace{0.5}_{1 \succ 2}, \underbrace{0.5}_{2 \succ 1}).$*

**Definition 26 (Parallel universes and possible losing rounds under MRSE rules).** *For any MRSE rule $\overline{r} = (\overline{r}_2, \ldots, \overline{r}_m)$ and any $\vec{x} \in \mathbb{R}^{m!}$, the set of parallel universes under $\overline{r}$ at $\vec{x}$, denoted by $PU_{\overline{r}}(\vec{x}) \subseteq \mathcal{L}(\mathcal{A})$, is the set of all elimination orders under PUT. Formally,*

$$PU_{\overline{r}}(\vec{x}) = \{O \in \mathcal{L}(\mathcal{A}) : \forall 1 \le i \le m-1, O[i] \in \arg\min_a Score_{\overline{r}_{m+1-i}}(\vec{x}|_{O[i,m]}, a)\},$$

*where $Score_{\overline{r}_{m+1-i}}(\vec{x}|_{O[i,m]}, a)$ is the total score of $a$ under the positional scoring rule $\overline{r}_{m+1-i}$, where the profile is $\vec{x}|_{O[i,m]}$.*

*For any alternative $a$, let the possible losing rounds, denoted by $LR_{\overline{r}}(\vec{x}, a) \subseteq [m-1]$, be the set of all rounds in the parallel universes where $a$ drops out. Formally,*

$$LR_{\overline{r}}(\vec{x}, a) = \{O^{-1}[a] : O \in PU_{\overline{r}}(\vec{x})\}$$

**Example 12.** *In the setting of Example 2, we let $\overline{r} = \overline{STV}$. $PU_{\overline{STV}}(\pi_{uni})$ consists of linear orders that correspond to all paths from the root to leaves in Figure 2. Therefore, $PU_{\overline{STV}}(\pi_{uni}) = \mathcal{L}(\mathcal{A})$. For every $a \in \mathcal{A}$, $LR_{\overline{STV}}(\pi_{uni}, a)$ corresponds to the rounds where $a$ is in a node of that round in Figure 2. Therefore, for every $a \in \mathcal{A}$, we have $LR_{\overline{STV}}(\pi_{uni}, a) = \{1, 2\}$.*

*For $\hat{\pi}$ in Figure 1, we have: $PU_{\overline{STV}}(\hat{\pi}) = \{[3 \rhd 1 \rhd 2], [3 \rhd 2 \rhd 1]\}$,[2] $LR_{\overline{STV}}(\hat{\pi}, 1) = LR_{\overline{STV}}(\hat{\pi}, 2) = \{2\}$, and $LR_{\overline{STV}}(\hat{\pi}, 3) = \{1\}$.*

**Theorem 2. (Semi-random CC: int-MRSE rules).** *Let $\mathcal{M} = (\Theta, \mathcal{L}(\mathcal{A}), \Pi)$ be a strictly positive and closed single-agent preference model, let $\overline{r} = (\overline{r}_2, \ldots, \overline{r}_m)$ be an int-MRSE and let $r$ be a refinement of $\overline{r}$. For any $n \in \mathbb{N}$ with $2 \mid n$, we have*

$$\widetilde{CC}^{\min}_{\Pi}(r, n) = \begin{cases} 1 & \text{if } \forall 2 \le i \le m, CL(\overline{r}_i) = 1 \\ 1 - \exp(-\Theta(n)) & \text{if } \begin{cases} (1) \exists 2 \le i \le m \text{ s.t. } CL(\overline{r}_i) = 0 \text{ and} \\ (2) \forall \pi \in CH(\Pi), \forall a \in WCW(\pi) \text{ and } \forall i^* \in LR_{\overline{r}}(\pi, a), \\ \qquad \text{we have } CL(\overline{r}_{m+1-i^*}) = 1 \end{cases} \\ \Theta(n^{-0.5}) & \text{if } \begin{cases} (1) \forall \pi \in CH(\Pi), CW(\pi) \cap (\mathcal{A} \setminus \overline{r}(\pi)) = \emptyset \text{ and} \\ (2) \exists \pi \in CH(\Pi) \text{ s.t. } |ACW(\pi) \cap (\mathcal{A} \setminus \overline{r}(\pi))| = 2 \end{cases} \\ \exp(-\Theta(n)) & \text{if } \exists \pi \in CH(\Pi) \text{ s.t. } CW(\pi) \cap (\mathcal{A} \setminus \overline{r}(\pi)) \ne \emptyset \\ \Theta(1) \text{ and } 1 - \Theta(1) & \text{otherwise} \end{cases}$$

*For any $n \in \mathbb{N}$ with $2 \nmid n$, we have*

$$\widetilde{CC}^{\min}_{\Pi}(r, n) = \begin{cases} 1 & \text{same as the } 2 \mid n \text{ case} \\ 1 - \exp(-\Theta(n)) & \text{same as the } 2 \mid n \text{ case} \\ \exp(-\Theta(n)) & \text{if } \exists \pi \in CH(\Pi) \text{ s.t. } \begin{cases} (1) CW(\pi) \cap (\mathcal{A} \setminus \overline{r}(\pi)) \ne \emptyset \text{ or} \\ (2) |ACW(\pi) \cap (\mathcal{A} \setminus \overline{r}(\pi))| = 2 \end{cases} \\ \Theta(1) \text{ and } 1 - \Theta(1) & \text{otherwise} \end{cases}$$

**Intuitive explanations.** The conditions for U, VU, and M cases are the same as their counterparts in Theorem 1. The most interesting cases are the 1 case and the VL case. The 1 case happens when all positional scoring rule used in $\overline{r}$ satisfy CONDORCET LOSER. This is true because for any positional scoring rule that satisfies CONDORCET LOSER, the Condorcet winner, when it exists, cannot have the lowest score among all alternatives. Therefore, like in Baldwin's rule, the Condorcet winner never loses in any round, which means that it must be the unique winner under $\overline{r}$.

The VL case happens when (1) the 1 case does not happen, and (2) for every distribution $\pi \in CH(\Pi)$, every weak Condorcet winner $a$, and every round $i^*$ where $a$ is eliminated in a parallel universe, the positional scoring rule used in round $i^*$, i.e. $\overline{r}_{m+1-i^*}$ for $m + 1 - i^*$ alternatives, must satisfy CONDORCET LOSER. (2) makes sense because it guarantees that when a small permutation is added to $\pi$, if a weak Condorcet winner $a$ becomes the Condorcet winner, then it will be the unique winner under $\overline{r}$, because in every round $i^*$ where $a$ can possibly be eliminated before the perturbation (i.e. $i^*$ is a possible losing round), the voting rule used in that round, i.e. $\overline{r}_{m+1-i^*}$, will not eliminate $a$ after $a$ has become a Condorcet winner. The following example shows the VL case under $\overline{STV}$.

---

[1] Again, we use $\rhd$ in contrast to $\succ$ to indicate that $O$ is a parallel universe instead of an agent's preferences.

[2] We use $\rhd$ to indicate the elimination order to avoid confusion with $\succ$.

**Example 13** (**Applications of Theorem 2 to STV**). *In the setting of Example 12, let STV denote an arbitrary refinement of $\overline{STV} = (\bar{r}_2, \bar{r}_3)$. The 1 case does not hold for sufficiently large $n$, because $\bar{r}_3$ (plurality) does not satisfy* CONDORCET LOSER.

*When $\Pi_{IC} = \{\pi_{uni}\}$, Theorem 2 implies that for any sufficiently large $n$ with $2 \mid n$, the $\Theta(1) \wedge (1 - \Theta(1))$ case holds. The $1 - \exp(-\Theta(n))$ case does not hold, because its condition (2) fails: $1 \in WCW(\pi_{uni})$ and round 1 is a possible losing round for alternative 1 (i.e., $1 \in LR_{\overline{STV}}(\pi_{uni}, 1)$), yet $\bar{r}_3$ does not satisfy* CONDORCET LOSER. *The $\Theta(n^{-0.5})$ case does not hold, because its condition (2) fails: $ACW(\pi_{uni}) = \emptyset$. The $\exp(-\Theta(n))$ case does not hold because $CW(\pi_{uni}) = \emptyset$.*

*Proof.* We apply Lemma 2 to prove the theorem. We first prove the following claim, which states that when $n$ is sufficiently large, $C_{AS}(\bar{r}, n) = 1$ if and only if all scoring rules used in $\bar{r}$ satisfy the Condorcet loser criterion.

**Claim 11.** *For int-MRSE $\bar{r}$, there exists $N \in n$ such that for every $n > N$, $C_{AS}(\bar{r}, n)$ holds if and only if for all $2 \leq i \leq m$, $CL(\bar{r}_i) = 1$.*

*Proof.* **The $\Leftarrow$ direction.** Suppose for all $2 \leq i \leq m$, $CL(\bar{r}_i) = 1$ and for the sake of contradiction, suppose $C_{AS}(\bar{r}, n) = 0$, which means that there exists an $n$-profile $P$ such that $CW(P) = \{a\}$ and $a \notin \bar{r}(P)$. This means that $LR_{\bar{r}}(\pi, a) \neq \emptyset$. Let $O \in LR_{\bar{r}}(\pi, a)$ denote an arbitrary possible losing round of $a$ and let $i^* = O^{-1}[a]$, which means that $a$ has the lowest total score in the restriction of $P$ on the remaining alternatives (i.e. $O[i^*, m]$), when $\bar{r}_{m+1-i^*}$ is used. In other words,

$$a \in \arg\min_b \text{Score}_{\bar{r}_{m+1-i^*}}(P|_{O[i^*, m]}, b)$$

Notice that $a$ is the Condorcet winner under $P$, which means that $a$ is also the Condorcet winner under $P|_{O[i^*, m]}$. We now obtain a profile $P_{i^*}$ over $O[i^*, m]$ from $P|_{O[i^*, m]}$, which constitutes a violation of CONDORCET LOSER for $\bar{r}_{m+1-i^*}$. Let $n' = |P|$.

$$P_{i^*} = (n' + 1) \times \mathcal{L}(O[i^*, m]) - P$$

That is, $P_{i^*}$ is obtained from $(n' + 1)$ copies of all linear orders over $O[i^*, m]$ by subtracting linear orders in $P$. It is not hard to verify that $a$ is the Condorcet loser as well as an $\bar{r}_{m+1-i^*}$ co-winner in $P_{i^*}$, because all alternatives are tied in the WMG of $(n' + 1) \times \mathcal{L}(O[i^*, m])$ and are tied w.r.t. their total $\bar{r}_{m+1-i^*}$ scores under $(n' + 1)\mathcal{L}(O[i^*, m])$. This is a contradiction to the assumption that all $\bar{r}_i$'s satisfies the Condorcet loser criterion.

**The $\Rightarrow$ direction.** For the sake of contradiction, suppose $CL(\bar{r}_{i^*}) = 1$ for some $2 \leq i^* \leq m$, which means that there exist a profile $P_1$ over $m+1-i^*$ alternatives $\{i^*, \ldots, m\}$, such that alternative $i^*$ is the Condorcet loser and a co-winner of $\bar{r}_{m+1-i^*}$ under $P_1$. We will construct a profile $P$ over $\mathcal{A}$ to show that $C_{AS}(\bar{r}, n) = 0$ for every sufficiently large $n$. We will show that alternatives in $O[1, i^* - 1]$ are eliminated in the first $i^* - 1$ round of executing $\bar{r}$ on $P$. Then $i^*$ will be eliminated in the next round.

First, we define a profile $P'$ over $O[i^*, m]$ where $i^*$ is the Condorcet winner as well as the unique $\bar{r}_{m+1-i^*}$ loser. Let $\sigma$ denote an arbitrary cyclic permutation among $O[i^* + 1, m]$, and let

$$P_2 = \{\sigma^i(a \succ O[i^* + 1, m]) : 1 \leq i \leq m - i^*\},$$

where alternatives in $O[i^* + 1, m]$ are ranked alphabetically. Let $n_1 = |P_1|$ and

$$P' = m(n_1 + 1) \times \mathcal{L}(O[i^*, m]) - m \times P_1 - P_2$$

It is not hard to verify that $P'$ is indeed a profile, i.e., the weight on each ranking is a non-negative integer. $i^*$ is the Condorcet winner under $P'$ because $i^*$ is the Condorcet loser in $P_1$, and $|P_2| < m$. $i^*$ is the unique loser under $P'$ because for any other alternative $a \in O[i^*, m]$, we have

$\text{Score}_{\bar{r}_{m+1-i^*}}(m(n' + 1) \times \mathcal{L}(O[i^*, m]), i^*) = \text{Score}_{\bar{r}_{m+1-i^*}}(m(n' + 1) \times \mathcal{L}(O[i^*, m]), a),$

$\text{Score}_{\bar{r}_{m+1-i^*}}(P_1, i^*) \geq \text{Score}_{\bar{r}_{m+1-i^*}}(P_1, a),$ and

$\text{Score}_{\bar{r}_{m+1-i^*}}(P_2, i^*) > \text{Score}_{\bar{r}_{m+1-i^*}}(P_2, a).$

Next, we let $P^*$ denote the profile obtained from $P'$ by appending $O[1] \succ O[2] \succ \cdots \succ O[i^* - 1]$ in the bottom. More precisely, we let

$$P^* = \{R \succ O[1] \succ O[2] \succ \cdots \succ O[i^* - 1] : R \in P'\}$$

Finally, we are ready to define $P$. Let $\sigma_1$ denote an arbitrary cyclic permutation among alternatives in $O[1, i^* - 1]$. Let $n' = |P'|$ and $P = P^1 \cup P^2 \cup P^3$, defined as follows.

- $P^1$ consists of $n'$ copies of $\{\sigma_1^i(P^*) : 1 \le i \le i^* - 1\}$. This part has $(n')^2(i^* - 1)$ rankings and is mainly used to guarantee that $O[1, i^* - 1]$ are removed in the first $i^* - 1$ rounds.

- $P^2$ consists of $\left\lfloor \frac{n - (n')^2(i^* - 1)}{n'} \right\rfloor$ copies of $P^*$. This part guarantees that $i^*$ is the Condorcet winner. We require $n$ to be sufficiently large so that $\lfloor \frac{n - (n')^2(i^* - 1)}{n'} \rfloor > n'$.

- $P^3$ consists of $n - |P_1| - |P_2|$ copies of $[O[m] \succ O[m-1] \succ \cdots \succ O[1]]$, which guarantees that $|P| = n$. Note that the number of rankings in this part is no more than $n'$.

Let $N = (n')^2$. For any $n > N$, notice that the second part has at least $n'$ copies of $P^*$, where $i^*$ is the Condorcet winner. Therefore, $i^*$ is the Condorcet winner under $P$. It is not hard to verify that $O[1, i^* - 1]$ are removed in the first $i^* - 1$ rounds under $\overline{r}$, and in the $i^*$-th round alternative $i^*$ is unique $\overline{r}_{m+1-i^*}$ loser, which means that $i^* \notin \overline{r}(P)$. This concludes the proof of Claim 11. $\qquad\square$

We prove the following claim to simplify $\text{Closure}(\mathcal{R}^{\overline{r}}_{\text{CWW}})$ and $\text{Closure}(\mathcal{R}^{\overline{r}}_{\text{CWL}})$.

**Claim 12.** *For any int-MRSE $\overline{r}$ and any $\pi \in CH(\Pi)$,*

$$\left[ \pi \in Closure(\mathcal{R}^{\overline{r}}_{CWW}) \right] \Leftrightarrow \left[ WCW(\pi) \cap \overline{r}(\pi) \ne \emptyset \right]$$
$$\left[ \pi \in Closure(\mathcal{R}^{\overline{r}}_{CWL}) \right] \Leftrightarrow \left[ \exists a \in WCW(\pi) \text{ and } i^* \in LR_{\overline{r}}(\pi, a) \text{ s.t. } CL(\overline{r}_{m+1-i^*}) = 0 \right]$$

*Proof.* The proof for the $\mathcal{R}^{\overline{r}}_{\text{CWW}}$ part is similar to the proof of Claim 9. We present the formal proof below for completeness.

$\left[ \boldsymbol{\pi \in \textbf{Closure}(\mathcal{R}^{\overline{r}}_{\textbf{CWW}})} \right] \Rightarrow \left[ \textbf{WCW}(\boldsymbol{\pi}) \cap \overline{\boldsymbol{r}}(\boldsymbol{\pi}) \ne \boldsymbol{\emptyset} \right]$. Suppose $\pi \in \text{Closure}(\mathcal{R}^{\overline{r}}_{\text{CWW}})$, which means that exists a sequence $(\vec{x}_1, x_2, \ldots)$ in $\mathcal{R}^{\overline{r}}_{\text{CWW}}$ that converges to $\pi$. It follows that there exists an alternative $a \in \mathcal{A}$ and a subsequence of $(\vec{x}_1, \vec{x}_2, \ldots)$, denoted by $(\vec{x}'_1, x'_2, \ldots)$, and $O \in \mathcal{L}(\mathcal{A})$ where $O[m] = a$, such that for every $j \in \mathbb{N}$, $\text{CW}(\vec{x}'_j) = \{a\}$ and $O \in \text{PU}_{\overline{r}}(\vec{x}'_j)$. This means that the following holds.

- $a$ is a weak Condorcet winner under $\pi$.
- $a \in \overline{r}(\pi)$. More precisely, $O \in \text{PU}_{\overline{r}}(\pi)$. To see this, recall that $O \in \text{PU}_{\overline{r}}(\vec{x}'_j)$ is equivalent to
$$\forall 2 \le i \le m, O[i] \in \arg\min_b \text{Score}_{\overline{r}_i}(\vec{x}'_j|_{O[i,m]}, b)$$
Therefore, the same relationship holds for $\pi$, namely
$$\forall 2 \le i \le m, O[i] \in \arg\min_b \text{Score}_{\overline{r}_i}(\pi|_{O[i,m]}, b),$$
which means that $O \in \text{PU}_{\overline{r}}(\pi)$.

Therefore, $a$ is a weak Condorcet winner as well as a $\overline{r}$ co-winner, which implies that $\text{WCW}(\pi) \cap \overline{r}(\pi) \ne \emptyset$.

$\left[ \boldsymbol{\pi \in \textbf{Closure}(\mathcal{R}^{\overline{r}}_{\textbf{CWW}})} \right] \Leftarrow \left[ \textbf{WCW}(\boldsymbol{\pi}) \cap \overline{\boldsymbol{r}}(\boldsymbol{\pi}) \ne \boldsymbol{\emptyset} \right]$. Suppose $\text{WCW}(\pi) \cap \overline{r}(\pi) \ne \emptyset$ and let $a \in \text{WCW}(\pi) \cap \overline{r}(\pi)$. We will explicitly construct a sequence of vectors in $\mathcal{R}^{\overline{r}}_{\text{CWW}}$ that converges to $\pi$. Because $a \in \overline{r}(\pi)$, there exists a parallel universe $O \in \text{PU}_{\overline{r}}(\pi)$ such that $O[m] = a$. Let $\vec{x} = -\text{Hist}(\{O\})$, i.e. we will use "negative" $O$ to break ties, so that for every $1 \le i \le m - 1$, $O[i]$ is eliminated in round $i$. For any $\delta > 0$, it is not hard to verify that $O \in \text{PU}_{\overline{r}}(\pi + \delta\vec{x})$. In fact, $\text{PU}_{\overline{r}}(\pi + \delta\vec{x}) = \{O\}$, i.e.

$$\forall 2 \le i \le m, \{O[i]\} = \arg\min_b \text{Score}_{\overline{r}_i}((\pi + \delta\vec{x})|_{O[i,m]}, b),$$

which means that $\{a\} = \overline{r}(\pi + \delta\vec{x})$. Notice that $a$ is the Condorcet winner under $\pi + \delta\vec{x}$ for any sufficiently small $\delta > 0$. Therefore, for any sufficiently small $\delta > 0$ we have $\pi + \delta\vec{x} \in \mathcal{R}^{\overline{r}}_{\text{CWW}}$. Because the sequence $(\pi + \vec{x}, \pi + \frac{1}{2}\vec{x}, \ldots)$ in $\mathcal{R}^{\overline{r}}_{\text{CWW}}$ converges to $\pi$, we have $\pi \in \text{Closure}(\mathcal{R}^{\overline{r}}_{\text{CWW}})$.

$\left[\pi \in \mathbf{Closure}(\mathcal{R}^{\overline{r}}_{\mathbf{CWL}})\right] \Rightarrow \left[\exists a \in \mathbf{WCW}(\pi) \text{ and } i^* \in \mathbf{LR}_{\overline{r}}(\pi, a) \text{ s.t. } \mathbf{CL}(\overline{r}_{m+1-i^*}) = 0\right].$
Suppose $\pi \in \text{Closure}(\mathcal{R}^{\overline{r}}_{\text{CWL}})$, which means that there exists a sequence $(\vec{x}_1, \vec{x}_2, \dots)$ in $\mathcal{R}^{\overline{r}}_{\text{CWL}}$ that converges to $\pi$. It follows that there exists $a \in \mathcal{A}, O \in \mathcal{L}(\mathcal{A})$ with $O[m] \neq a$, and a subsequence of $(\vec{x}_1, \vec{x}_2, \dots)$, denoted by $(\vec{x}'_1, \vec{x}'_2, \dots)$ such that for every $j \in \mathbb{N}$, $\text{CW}(\vec{x}'_j) = \{a\}$ and $O \in \text{PU}_{\overline{r}}(\vec{x}'_j)$. Let $i^* = O^{-1}[a]$, i.e. $i^*$ is the round where $a$ loses in the parallel universe $O$, which means that for every $j \in \mathbb{N}$,

$$a \in \arg\min_b \text{Score}_{\overline{r}_{m+1-i^*}}(\vec{x}'_j|_{O[i^*, m]}, b).$$

Notice that $a$ is the Condorcet winner among $O[i^*, m]$. This means that $\overline{r}_{m+1-i^*}$ does not satisfy the Condorcet loser criterion, because for any sufficiently large $\psi > 0$, $a$ is the Condorcet loser as well as a co-winner in $\psi \cdot \text{Hist}(O[i^*, m]) - \vec{x}'_j|_{O[i^*, m]}$. Because $(\vec{x}'_1, \vec{x}'_2, \dots)$ converges to $\pi$, it is not hard to verify that $a \in \text{WCW}(\pi)$ and $O \in \text{PU}_{\overline{r}}(\pi)$. Therefore, we have $a \in \text{WCW}(\pi)$, $i^* \in \text{LR}_{\overline{r}}(\pi, a)$, and $\text{CL}(\overline{r}_{m+1-i^*}) = 0$.

$\left[\pi \in \mathbf{Closure}(\mathcal{R}^{\overline{r}}_{\mathbf{CWL}})\right] \Leftarrow \left[\exists a \in \mathbf{WCW}(\pi) \text{ and } i^* \in \mathbf{LR}_{\overline{r}}(\pi, a) \text{ s.t. } \mathbf{CL}(\overline{r}_{m+1-i^*}) = 0\right].$
Let $a \in \text{WCW}(\pi)$ and $i^* \in \text{LR}_{\overline{r}}(\pi, a)$ such that $\text{CL}(\overline{r}) = 0$. Furthermore, we let $O^* \in \text{PU}_{\overline{r}}(\pi)$ denote the parallel universe such that $O[i^*] = a$. Because $\overline{r}_{m+1-i^*}$ does not satisfy the Condorcet loser criterion, there exists profile $P_a$ over $O[i^*, m]$ where $a$ is the Condorcet loser but $a \in \overline{r}_{m+1-i^*}(P_a)$. In fact, there exists a profile $P_a^*$ where $a$ is the Condorcet loser but $\{a\} = \overline{r}_{m+1-i^*}(P^*)$, i.e. $a$ is the unique winner under $P_a^*$. To see this, let $\sigma$ denote an arbitrary cyclic permutation among $O[i^* + 1, m]$, and let

$$P = \{\sigma^i(a \succ O[i^* + 1, m]) : 1 \leq i \leq m - i^*\}$$

It is not hard to verify that the score of $a$ is strictly larger than the score of any other alternative under $P$. Therefore, when $\delta > 0$ is sufficiently small, $a$ is the Condorcet loser as well as the unique winner under $P_a^* = P_a + \delta \cdot P$. Now, we define a profile $P'$ over $\mathcal{A}$ by stacking $O[1, i^* - 1]$ on top of each (fractional) ranking in $P_a^*$. In other words, a ranking $[O[1] \succ \cdots \succ O[i^* - 1] \succ R^*]$ is in $P'$ if and only if $R^* \in P_a^*$, and the two rankings have the same weights (in $P'$ and $P_a^*$, respectively).

Let $\vec{x} = -\text{Hist}(P')$. It is not hard to verify that for any $\delta > 0$, $a$ is the Condorcet winner under $\pi + \delta \vec{x}$ and in the first $i^*$ rounds of the execution of $\overline{r}$, $O[1], O[2], \dots, O[i^*]$ are eliminated in order. In particular, $O[i^*] = a$ is eliminated in the $i^*$-th round, which means that $a \notin \overline{r}(\pi + \delta \vec{x})$. Consequently, $\pi + \delta \vec{x} \in \mathcal{R}^{\overline{r}}_{\text{CWL}}$. Notice that $(\pi + \frac{1}{j} \vec{x} : j \in \mathbb{N})$ is a sequence in $\mathcal{R}^{\overline{r}}_{\text{CWL}}$ that converges to $\pi$, which means that $\pi \in \text{Closure}(\mathcal{R}^{\overline{r}}_{\text{CWL}})$. This proves Claim 12. $\qquad\square$

We now apply Claim 12 to simplify the conditions in Lemma 2.

- $\text{C}_{\text{RS}}(\overline{r}, \pi)$. By definition, this holds if and only if $\pi \notin \text{Closure}(\mathcal{R}^{\overline{r}}_{\text{CWL}})$, which is equivalent to $\nexists a \in \text{WCW}(\pi)$ and $i^* \in \text{LR}_{\overline{r}}(\pi, a)$ s.t. $\text{CL}(\overline{r}_{m+1-i^*}) = 0$. In other words, for all $a \in \text{WCW}(\pi)$ and all $i^* \in \text{LR}_{\overline{r}}(\pi, a)$, $\overline{r}_{m+1-i^*}$ satisfies CONDORCET LOSER, or equivalently, $\forall a \in \text{WCW}(\pi)$ and $\forall i^* \in \text{LR}_{\overline{r}}(\pi, a), \text{CL}(\overline{r}_{m+1-i^*}) = 1$.

- $\text{C}_{\text{NRS}}(\overline{r}, \pi)$. By definition, this holds if and only if $\text{ACW}(\pi) \neq \emptyset$ and $\pi \notin \text{Closure}(\mathcal{R}^{\overline{r}}_{\text{CWW}})$, which is equivalent to $\text{ACW}(\pi) \neq \emptyset$ and $\text{WCW}(\pi) \cap \overline{r}(\pi) = \emptyset$. The latter is equivalent to $\text{WCW}(\pi) \cap (\mathcal{A} \setminus \overline{r}(\pi)) = \text{WCW}(\pi)$. We note that when $\text{ACW}(\pi) \neq \emptyset$, we have $\text{WCW}(\pi) = \text{ACW}(\pi)$. Therefore, $\text{C}_{\text{NRS}}(\overline{r}, \pi)$ is equivalent to $|\text{ACW}(\pi) \cap (\mathcal{A} \setminus \overline{r}(\pi))| = 2$.

Theorem 2 follows after Lemma 2 with the simplified conditions discussed above. $\qquad\square$

# F  Materials for Section 3: Semi-random PARTICIPATION

## F.1  Lemma 3 and Its Proof

We first introduce some notation to present the theorem.

**Definition 27** ($\oplus$ **operator**). *For any pair of signatures $\vec{t}_1, \vec{t}_2 \in \mathcal{S}_K$, we define $\vec{t}_1 \oplus \vec{t}_2$ to be the following signature:*

$$\forall k \leq K, [\vec{t}_1 \oplus \vec{t}_2]_k = \begin{cases} [\vec{t}_1]_k & \text{if } [\vec{t}_1]_k = [\vec{t}_2]_k \\ 0 & \text{otherwise} \end{cases}$$

For example, when $K = 3$, $\vec{t}_1 = (+, -, 0)$, and $\vec{t}_2 = (+, 0, 0)$, we have $\vec{t}_1 \oplus \vec{t}_2 = (+, 0, 0)$. By definition, we have $\vec{t}_1 \trianglelefteq \vec{t}_1 \oplus \vec{t}_2$ and $\vec{t}_2 \trianglelefteq \vec{t}_1 \oplus \vec{t}_2$.

**Definition 28 ($\text{Vio}^r_{\text{PAR}}(n)$ and $\ell_n$).** *For any GSR $r$ and any $n \in \mathbb{N}$, we define*

$$\text{Vio}^r_{\text{PAR}}(n) = \left\{ Sign_{\vec{H}}(P) \oplus Sign_{\vec{H}}(P \setminus \{R\}) : P \in \mathcal{L}(\mathcal{A})^n, R \in \mathcal{L}(\mathcal{A}), r(P \setminus \{R\}) \succ_R r(P) \right\}$$

$$\ell_n = m! - \max_{\vec{t} \in \text{Vio}^r_{\text{PAR}}(n): \exists \pi \in CH(\Pi),\ s.t.\ \vec{t} \trianglelefteq Sign_{\vec{H}}(\pi)} \dim(\mathcal{H}^{\vec{t}}_{\leq 0})$$

In words, $\text{Vio}^r_{\text{PAR}}(n)$ consists of all signatures $\vec{t}$ that is obtained by combining two feasible signatures, i.e., $Sign_{\vec{H}}(P)$ and $Sign_{\vec{H}}(P \setminus \{R\})$, by the $\oplus$ operator, where $P$ and $R$ constitutes an violation of PAR. Notice that $r(P \setminus \{R\}) \succ_R r(P)$ implicitly assumes that $P$ contains an $R$ vote. Then, $\ell_n$ is defined to be $m!$ minus the maximum dimension of polyhedron $\mathcal{H}^{\vec{t}}$, among all $\vec{t}$ in $\text{Vio}^r_{\text{PAR}}(n)$ that refines $Sign_{\vec{H}}(\pi)$ for some $\pi \in CH(\Pi)$.

**Lemma 3 (Semi-random PAR: Int-GSR).** *Let $\mathcal{M} = (\Theta, \mathcal{L}(\mathcal{A}), \Pi)$ be a strictly positive and closed single-agent preference model, let $r$ be an int-GSR. For any $n \in \mathbb{N}$,*

$$\widetilde{\text{PAR}}^{\min}_{\Pi}(r, n) = \begin{cases} 1 & \text{if } \widetilde{\text{Vio}}^r_{\text{PAR}}(n) = \emptyset \\ 1 - \exp(-\Theta(n)) & \text{otherwise if } \forall \pi \in CH(\Pi) \text{ and } \vec{t} \in \text{Vio}^r_{\text{PAR}}(n), \vec{t} \ntrianglelefteq Sign_{\vec{H}}(\pi) \\ 1 - \Theta(n^{-\ell_n/2}) & \text{otherwise, i.e. } \exists \pi \in CH(\Pi) \text{ and } \vec{t} \in \text{Vio}^r_{\text{PAR}}(n) \text{ s.t. } \vec{t} \trianglelefteq Sign_{\vec{H}}(\pi) \end{cases}$$

Applying Lemma 3 to a voting rule $r$ often involves the following steps. First, we choose an GSR representation of $r$ by specifying the $\vec{H}$ and $g$, though according to Lemma 3 the asymptotic bound does not depend on such choice. Second, we characterize $\text{Vio}^r_{\text{PAR}}(n)$ and verify whether it is empty. If $\text{Vio}^r_{\text{PAR}}(n)$ is empty then the 1 case holds. Third, if $\text{Vio}^r_{\text{PAR}}(n)$ is non-empty but none of $\vec{t} \in \text{Vio}^r_{\text{PAR}}(n)$ refines $Sign_{\vec{H}}(\pi)$ for any $\pi \in CH(\Pi)$, then the VL case holds. Finally, if neither 1 nor VL case holds, then the L case holds, where the degree of polynomial depends on $\ell_n$. Characterizing $\text{Vio}^r_{\text{PAR}}(n)$ and $\ell_n$ can be highly challenging, as it aims at summarizing all violations of PAR for $n$-profiles (using signatures under $\vec{H}$).

*Proof.* The high-level idea of the proof is similar to the proof of Lemma 2. In light of Lemma 1, the proof proceeds in the following three steps. **Step 1.** Define $\mathcal{C}$ that characterizes the satisfaction of PARTICIPATION of $r$, and an almost complement $\mathcal{C}^*$ of $\mathcal{C}$. **Step 2.** Characterize possible values of $\alpha_n^*$ and their conditions, and then notice that $\alpha_n^*$ is at most $m! - 1$, which means that only the 1, VL, or L case in Lemma 1 hold. This means that the value of $\beta_n$ does not matter. **Step 3.** Apply Lemma 1.

**Step 1.** Given two feasible signatures $\vec{t}_1, \vec{t}_2 \in \mathcal{S}_{\vec{H}}$ and a ranking $R \in \mathcal{L}(\mathcal{A})$, we first formally define a polyhedron $\mathcal{H}^{\vec{t}_1, R, \vec{t}_2}$ to characterize the profiles whose signature is $\vec{t}_1$ and after removing a voter whose preferences are $R$, the signature of the new profile becomes $\vec{t}_2$.

**Definition 29 ($\mathcal{H}^{\vec{t}_1, R, \vec{t}_2}$).** *Given $\vec{H} = (\vec{h}_1, \ldots, \vec{h}_K) \in (\mathbb{Z}^{m!})^K$, $\vec{t}_1, \vec{t}_2 \in \mathcal{S}_{\vec{H}}$, and $R \in \mathcal{L}(\mathcal{A})$, we*

$$\text{let } \mathbf{A}^{\vec{t}_1, R, \vec{t}_2} = \begin{bmatrix} -Hist(R) \\ \mathbf{A}^{\vec{t}_1} \\ \mathbf{A}^{\vec{t}_2} \end{bmatrix}, \vec{\mathbf{b}}^{\vec{t}_1, R, \vec{t}_2} = [-1, \underbrace{\vec{\mathbf{b}}^{\vec{t}_1}}_{\text{for } \mathbf{A}^{\vec{t}_1}}, \underbrace{\vec{\mathbf{b}}^{\vec{t}_2} + Hist(R) \times \left( \mathbf{A}^{\vec{t}_2} \right)^{\top}}_{\text{for } \mathbf{A}^{\vec{t}_2}}] \text{ and}$$

$$\mathcal{H}^{\vec{t}_1, R, \vec{t}_2} = \left\{ \vec{x} \in \mathbb{R}^{m!} : \mathbf{A}^{\vec{t}_1, R, \vec{t}_2} \cdot (\vec{x})^{\top} \leq \left( \vec{\mathbf{b}}^{\vec{t}_1, R, \vec{t}_2} \right)^{\top} \right\}$$

Notice that $Hist(R) \in \{0, 1\}^{m!}$ is the vector whose $R$-component is 1 and all other components are 0's. The $\mathbf{A}^{\vec{t}_2}$ part in Definition 29 is equivalent to $\mathbf{A}^{\vec{t}_2} \cdot (\vec{x} - Hist(R))^{\top} \leq \left( \vec{\mathbf{b}}^{\vec{t}_2} \right)^{\top}$. We prove properties of $\mathcal{H}^{\vec{t}_1, R, \vec{t}_2}$ in the following claim.

**Claim 13 (Properties of $\mathcal{H}^{\vec{t}_1, R, \vec{t}_2}$).** *Given integer $\vec{H}$. For any $\vec{t}_1, \vec{t}_2 \in \mathcal{S}_{\vec{H}}$, any $R \in \mathcal{L}(\mathcal{A})$,*

*(i) for any integral profile $P$, $Hist(P) \in \mathcal{H}^{\vec{t}_1, R, \vec{t}_2}$ if and only if $Sign_{\vec{H}}(P) = \vec{t}_1$ and $Sign_{\vec{H}}(P \setminus \{R\}) = \vec{t}_2$;*

*(ii) for any $\vec{x} \in \mathbb{R}_{\geq 0}^{m!}$, $\vec{x} \in \mathcal{H}_{\leq 0}^{\vec{t}_1, R, \vec{t}_2}$ if and only if $\vec{t}_1 \oplus \vec{t}_2 \trianglelefteq \text{Sign}_{\vec{H}}(\vec{x})$;*

*(iii) If there exists $\vec{x} \in \mathcal{H}_{\leq 0}^{\vec{t}_1, R, \vec{t}_2}$ such that $[\vec{x}]_R > 0$, then $\dim(\mathcal{H}_{\leq 0}^{\vec{t}_1, R, \vec{t}_2}) = \dim(\mathcal{H}_{\leq 0}^{\vec{t}_1 \oplus \vec{t}_2})$.*
*Moreover, if $\vec{t}_1 \neq \vec{t}_2$ and $\mathcal{H}_{\leq 0}^{\vec{t}_1, R, \vec{t}_2} \neq \emptyset$, then $\dim(\mathcal{H}_{\leq 0}^{\vec{t}_1, R, \vec{t}_2}) \leq m! - 1$.*

*Proof.* Part (i) follows after the definition. Part (ii) also follows after the definition. Recall that $\vec{x} \in \mathcal{H}_{\leq 0}^{\vec{t}_1, R, \vec{t}_2}$ if and only if $\mathbf{A}^{\vec{t}_1} \cdot (\vec{x})^\top \leq \left(\vec{0}\right)^\top$, $\mathbf{A}^{\vec{t}_2} \cdot (\vec{x})^\top \leq \left(\vec{0}\right)^\top$, and the $R$ component of $\vec{x}$ is non-negative, which is automatically satisfied for every $\vec{x} \in \mathbb{R}_{\geq 0}^{m!}$. The first sets of inequalities holds if and only if $\mathbf{A}^{\vec{t}_1 \oplus \vec{t}_2} \cdot (\vec{x})^\top \leq \left(\vec{0}\right)^\top$.

To prove the first part of Part (iii), let $\mathbf{A}_1^+$ and $\mathbf{A}_2^+$ denote the essential equalities of $\mathbf{A}^{\vec{t}_1, R, \vec{t}_2}$ and $\mathbf{A}^{\vec{t}_1 \oplus \vec{t}_2}$, respectively. We show that $\mathbf{A}_1^+$ and $\mathbf{A}_2^+$ contains the same set of row vectors (while some rows may appear different number of times in $\mathbf{A}_1^+$ and $\mathbf{A}_2^+$). As noted in the proof of Part (ii), the set of row vectors in $\mathbf{A}^{\vec{t}_1, R, \vec{t}_2}$ is the same as the set of row vectors in $\mathbf{A}^{\vec{t}_1 \oplus \vec{t}_2}$, except that the former contains $-\text{Hist}(R)$. Recall that we have assumed that there exists $\vec{x} \in \mathcal{H}_{\leq 0}^{\vec{t}_1, R, \vec{t}_2}$ such that $[\vec{x}]_R > 0$, which means that $-\text{Hist}(R) \cdot (\vec{x})^\top$ does not hold for every vector in $\mathcal{H}_{\leq 0}^{\vec{t}_1, R, \vec{t}_2}$. Therefore, $-\text{Hist}(R)$ is not a row in $\mathbf{A}_1^+$, which means that $\mathbf{A}_1^+$ and $\mathbf{A}_2^+$ contains the same set of row vectors. Then, we have

$$\dim(\mathcal{H}_{\leq 0}^{\vec{t}_1, R, \vec{t}_2}) = m! - \text{Rank}(\mathbf{A}_1^+) = m! - \text{Rank}(\mathbf{A}_2^+) = \dim(\mathcal{H}_{\leq 0}^{\vec{t}_1 \oplus \vec{t}_2})$$

The second part of Part (iii) is proved by noticing that when $\vec{t}_1 \neq \vec{t}_2$, $\vec{t}_1 \oplus \vec{t}_2$ contains at least one 0. Suppose $[\vec{t}_1 \oplus \vec{t}_2]_k = 0$. This means that for all $\vec{x} \in \mathcal{H}_{\leq 0}^{\vec{t}_1, R, \vec{t}_2}$, we have $\vec{h}_k \cdot \vec{x} = 0$, which means that $\dim(\mathcal{H}_{\leq 0}^{\vec{t}_1, R, \vec{t}_2}) \leq m! - 1$. $\qquad\qquad\square$

We now use $\mathcal{H}^{\vec{t}_1, R, \vec{t}_2}$ to define $\mathcal{C}$ and $\mathcal{C}^*$.

**Definition 30** ($\mathcal{C}$ and $\mathcal{C}^*$ for PARTICIPATION). *Given an int-GSR $r$ characterized by $\vec{H}$ and $g$, we define*

$$\mathcal{C} = \bigcup_{\vec{t}_1, \vec{t}_2 \in \mathcal{S}_{\vec{H}}, R \in \mathcal{L}(\mathcal{A}): r(\vec{t}_1) \succeq_R r(\vec{t}_2)} \mathcal{H}^{\vec{t}_1, R, \vec{t}_2}$$

$$\mathcal{C}^* = \bigcup_{\vec{t}_1, \vec{t}_2 \in \mathcal{S}_{\vec{H}}, R \in \mathcal{L}(\mathcal{A}): r(\vec{t}_1) \prec_R r(\vec{t}_2)} \mathcal{H}^{\vec{t}_1, R, \vec{t}_2}$$

In words, $\mathcal{C}$ consists of polyhedra $\mathcal{H}^{\vec{t}_1, R, \vec{t}_2}$ that characterize the histograms of profiles such that after any $R$-vote is removed, the winner under $r$ is not improved w.r.t. $R$. $\mathcal{C}^*$ consists of polyhedra $\mathcal{H}^{\vec{t}_1, R, \vec{t}_2}$ that characterize the histograms of profiles such that after removing an $R$-vote, the winner under $r$ is strictly improved w.r.t. $R$. It is not hard to see that $\mathcal{C}^*$ is an almost complement of $\mathcal{C}$.

It follows from Claim 13 (i) that for any $n$-profile $P$, PAR is satisfied (respectively, dissatisfied) at $P$ if and only if $\text{Hist}(P) \in \mathcal{C}$ (respectively, $\text{Hist}(P) \in \mathcal{C}^*$).

**Step 2: Characterize $\alpha_n^*$.** In this step we discuss the values and conditions for $\alpha_n^*$ (for $\mathcal{C}^*$) in the following three cases.

$\alpha_n^* = -\infty$. This case holds if and only if PAR holds for all $n$-profiles, which is equivalent to $\text{Vio}_{\text{PAR}}^r(n) = \emptyset$.

$\alpha_n^* = -\frac{n}{\log n}$. This case holds if and only if (1) PAR is not satisfied at all $n$-profiles, which is equivalent to $\text{Vio}_{\text{PAR}}^r(n) \neq \emptyset$, and (2) the activation graph $\mathcal{G}_{\Pi, \mathcal{C}^*, n}$ does not contain any non-negative edges, which is equivalent to $\forall \pi \in \text{CH}(\Pi)$ and $\forall \mathcal{H}^{\vec{t}_1, R, \vec{t}_2} \subseteq \mathcal{C}^*$ that is active at $n$, we have $\pi \notin \mathcal{H}_{\leq 0}^{\vec{t}_1, R, \vec{t}_2}$. We will prove that part (2) is equivalent to the following:

$$(2) \iff \left[\forall \pi \in \text{CH}(\Pi) \text{ and } \vec{t} \in \text{Vio}_{\text{PAR}}^r(n), \vec{t} \not\trianglelefteq \text{Sign}_{\vec{H}}(\pi)\right] \tag{13}$$

We first prove the "⇒" direction of (13). Suppose for the sake of contradiction that this is not true. That is, $\mathcal{G}_{\Pi,\mathcal{C}^*,n}$ does not contain any non-negative edges and there exist $\pi \in \mathrm{CH}(\Pi)$ and $\vec{t} \in \mathrm{Vio}_{\mathrm{PAR}}^r(n)$ such that $\vec{t} \not\trianglelefteq \mathrm{Sign}_{\vec{H}}(\pi)$. Let $P$ denote the $n$-profile such that $\mathrm{Sign}_{\vec{H}}(P) = \vec{t}_1$, $\mathrm{Sign}_{\vec{H}}(P \setminus \{R\}) = \vec{t}_2$, $r(P \setminus \{R\}) \succ_R r(P)$, and $\vec{t} = \vec{t}_1 \oplus \vec{t}_2$. By Claim 13 (i), $\mathrm{Hist}(P) \in \mathcal{H}^{\vec{t}_1,R,\vec{t}_2}$, which means that $\mathcal{H}^{\vec{t}_1,R,\vec{t}_2}$ is active at $n$. By Claim 13 (ii), $\mathrm{Hist}(P) \in \mathcal{H}_{\leq 0}^{\vec{t}_1,R,\vec{t}_2}$. These imply that the weight on the edge $(\pi, \mathcal{H}^{\vec{t}_1,R,\vec{t}_2})$ in $\mathcal{G}_{\Pi,\mathcal{C}^*,n}$ is non-negative (whose weight is $\dim(\mathcal{H}_{\leq 0}^{\vec{t}_1,R,\vec{t}_2})$), which contradicts the assumption that (2) holds.

Next, we prove the "⇐" direction of (13). Suppose for the sake of contradiction that (2) does not hold, which means that there exists an edge $(\pi, \mathcal{H}^{\vec{t}_1,R,\vec{t}_2})$ in $\mathcal{G}_{\Pi,\mathcal{C}^*,n}$ whose weight is non-negative. Equivalently, $\mathcal{H}^{\vec{t}_1,R,\vec{t}_2}$ is active at $n$ and $\pi \in \mathcal{H}_{\leq 0}^{\vec{t}_1,R,\vec{t}_2}$. By Claim 13 (ii), $\vec{t}_1 \oplus \vec{t}_2 \in \mathrm{Vio}_{\mathrm{PAR}}^r(n)$. Recall that $\pi$ is strictly positive, and then by Claim 13 (ii), we have $t_1 \oplus \vec{t}_2 \trianglelefteq \mathrm{Sign}_{\vec{H}}(\pi)$. However, this contradict the assumption.

These prove (13).

$\boldsymbol{\alpha_n^* > 0.}$   For this case, we prove

$$\alpha_n^* = \max_{\vec{t} \in \mathrm{Vio}_{\mathrm{PAR}}^r(n): \exists \pi \in \mathrm{CH}(\Pi),\text{ s.t. } \vec{t} \trianglelefteq \mathrm{Sign}_{\vec{H}}(\pi)} \dim(\mathcal{H}_{\leq 0}^{\vec{t}}), \tag{14}$$

We first prove the "≤" direction in (14). For any edge $(\pi, \mathcal{H}^{\vec{t}_1,R,\vec{t}_2})$ in $\mathcal{G}_{\Pi,\mathcal{C}^*,n}$ whose weight is non-negative, $\mathcal{H}^{\vec{t}_1,R,\vec{t}_2}$ is active at $n$. Therefore, there exists an $n$-profile $P$ such that $\mathrm{Hist}(P) \in \mathcal{H}^{\vec{t}_1,R,\vec{t}_2}$. Let $\vec{t} = \vec{t}_1 \oplus \vec{t}_2$. We have $\vec{t} \in \mathrm{Vio}_{\mathrm{PAR}}^r(n)$. By Claim 13 (ii), we have $\vec{t} \trianglelefteq \mathrm{Sign}_{\vec{H}}(\pi)$. By Claim 13 (iii), we have $\dim(\mathcal{H}_{\leq 0}^{\vec{t}_1,R,\vec{t}_2}) = \dim(\mathcal{H}_{\leq 0}^{\vec{t}})$. Therefore, the "≤" direction in (14) holds.

Next, we prove the $\geq$ direction of (14). For any $\vec{t} \in \mathrm{Vio}_{\mathrm{PAR}}^r(n)$ and $\pi \in \mathrm{CH}(\Pi)$ such that $\vec{t} \trianglelefteq \mathrm{Sign}_{\vec{H}}(\pi)$, let $P$ denote an $n$-profile and let $R$ denote a ranking that justify $\vec{t}$'s membership in $\mathrm{Vio}_{\mathrm{PAR}}^r(n)$, and let $\vec{t}_1 = \mathrm{Sign}_{\vec{H}}(P)$ and $\vec{t}_2 = \mathrm{Sign}_{\vec{H}}(P \setminus \{R\})$, which means that $\vec{t} = \vec{t}_1 \oplus \vec{t}_2$. By Claim 13 (i), $\mathrm{Hist}(P) \in \mathcal{H}^{\vec{t}_1,R,\vec{t}_2} \subseteq \mathcal{C}^*$, which means that $\mathcal{H}^{\vec{t}_1,R,\vec{t}_2}$ is active at $n$. By Claim 13 (ii), $\pi \in \mathcal{H}_{\leq 0}^{\vec{t}_1,R,\vec{t}_2}$. By Claim 13 (iii), $\dim(\mathcal{H}_{\leq 0}^{\vec{t}_1,R,\vec{t}_2}) = \dim(\mathcal{H}_{\leq 0}^{\vec{t}})$. This means that the weight on the edge $(\pi, \mathcal{H}^{\vec{t}_1,R,\vec{t}_2})$ in $\mathcal{G}_{\Pi,\mathcal{C}^*,n}$ is $\dim(\mathcal{H}_{\leq 0}^{\vec{t}})$, which implies the "≥" direction in (14) holds.

Therefore, (14) holds. Notice that by Claim 13 (iii), $\alpha_n^* \leq m! - 1$.

**Step 3: Applying Lemma 1.**  Lemma 3 follows after a straightforward application of Lemma 1 and Step 2. Notice that $\Pi_{\mathcal{C},n}$ and $\beta_n$ are irrelevant in this proof because only the $1, 1 - \exp(n)$, and $1 - \mathcal{H}(n)$ cases will happen. This completes the proof of Lemma 3.   □

### F.2   Proof of Theorem 3

Recall from Definition 9 that an EO-based rule is determined by the total preorder over edges in WMG w.r.t. their weights. Theorem 3 characterizes semi-random PAR for any EO-based int-GSR refinements of maximin, Ranked Pairs, and Schulze.

**Theorem 3** (Semi-random PAR: maximin, Ranked Pairs, Schulze).  *For any $m \geq 4$, any EO-based int-GSR $r$ that is a refinement of maximin, STV, Schulze, or ranked Pairs, and any strictly positive and closed $\Pi$ over $\mathcal{L}(\mathcal{A})$ with $\pi_{uni} \in \mathrm{CH}(\Pi)$, there exists $N \in \mathbb{N}$ such that for every $n \geq N$,*

$$\widetilde{\mathrm{PAR}}_{\Pi}^{\min}(r, n) = 1 - \Theta\left(\frac{1}{\sqrt{n}}\right)$$

*Proof.* Because $r$ is EO-based, w.l.o.g., we assume that its int-GSR representation uses $\vec{H}_{\mathrm{EO}}$ (Definition 11).

**Overview.**   The proof is done by applying Lemma 3 to show that for any sufficiently large $n$, the 1 case and the VL case do not happen, and $\ell_n = 1$ in the L case. This is done by explicitly constructing an $n$-profile $P$, under which PAR is violated when an $R$-vote is removed (which means

that $\vec{t} = \mathrm{Sign}_{\vec{H}_{\mathrm{EO}}}(P) \oplus \mathrm{Sign}_{\vec{H}_{\mathrm{EO}}}(P \setminus \{R\}) \in \mathrm{Vio}^r_{\mathrm{PAR}}(n)$ and therefore the 1 case does not hold), then show that $\vec{t} \trianglelefteq \pi_{\mathrm{uni}}$, or more generally, any signature refines $\mathrm{Sign}_{\vec{H}_{\mathrm{EO}}}(\pi_{\mathrm{uni}})$ (which means that the VL case does not hold), and finally prove that $\dim(\mathcal{H}^{\vec{t}}_{\leq 0}) = m! - 1$, which means that $\ell_n = 1$.

**Maximin: $r$ refines $\overline{\mathrm{MM}}$.** We first prove the proposition for $2 \nmid n$, then show how to modify the proof for $2 \mid n$. As mentioned in the overview, the proof proceeds in the following steps.

**Constructing $P_{\mathrm{MM}}$ and $R_{\mathrm{MM}}$ that violates** PAR. Let $G_{\mathrm{MM}}$ denote the following weighted directed graph with weights $w_{\mathrm{MM}}$, where the weights on all edges are odd and different, except on $4 \to 1$ and $3 \to 2$.

- $w_{\mathrm{MM}}(4,1) = w_{\mathrm{MM}}(3,2) = 5$, $w_{\mathrm{MM}}(1,2) = 1$, $w_{\mathrm{MM}}(1,3) = 9$, $w_{\mathrm{MM}}(2,4) = 13$, and $w_{\mathrm{MM}}(3,4) = 17$;

- for every $5 \leq i \leq m$, $w_{\mathrm{MM}}(1,i) \geq 21$, $w_{\mathrm{MM}}(2,i) \geq 21$, $w_{\mathrm{MM}}(3,i) \geq 21$, and $w_{\mathrm{MM}}(4,i) \geq 21$;

- the weights on other edges are assigned arbitrarily. Moreover, the difference between any pair of edges is at least 4, except that the weights on $4 \to 1$ and $3 \to 2$ are the same.

See the middle graph in Figure 6 for an example of $m = 5$.

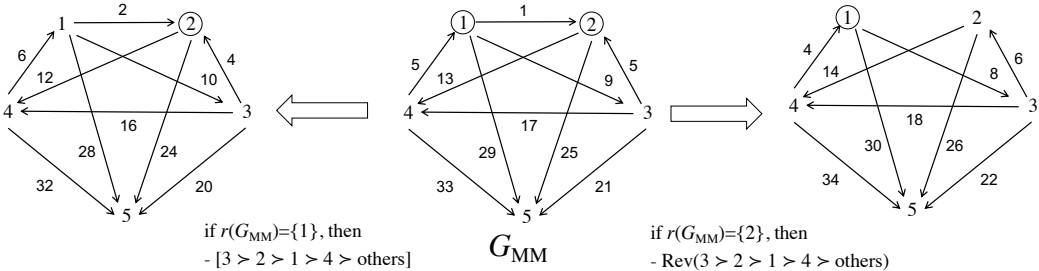

if $r(G_{\mathrm{MM}}) = \{1\}$, then
- $[3 \succ 2 \succ 1 \succ 4 \succ$ others]

$G_{\mathrm{MM}}$

if $r(G_{\mathrm{MM}}) = \{2\}$, then
- Rev$(3 \succ 2 \succ 1 \succ 4 \succ$ others)

Figure 6: WMGs for minimax. $\overline{\mathrm{MM}}$ (co)-winners are circled.

It follows from McGarvey's theorem [36] that for any $n > m^4$ and $2 \nmid n$, there exists an $n$-profile $P_{\mathrm{MM}}$ whose WMG is $G_{\mathrm{MM}}$. Therefore, for any $n > m^4 + 2$ and $2 \nmid n$, there exists an $n$-profile $P_{\mathrm{MM}}$ whose WMG is $G_{\mathrm{MM}}$, and $P_{\mathrm{MM}}$ includes the following two rankings:

$$[3 \succ 2 \succ 1 \succ 4 \succ \text{others}], \mathrm{Rev}\,(3 \succ 2 \succ 1 \succ 4 \succ \text{others}),$$

where for any ranking $R$, Rev $(R)$ denotes its reverse ranking. We now show that $\mathrm{PAR}(r, P_{\mathrm{MM}}) = 0$, which implies that the 1 case does not happen. Notice that the min-score of alternatives 1 and 2 are the highest, which means that $r(P_{\mathrm{MM}}) \subseteq \{1, 2\}$.

- If $r(P_{\mathrm{MM}}) = \{1\}$, then we let $R_{\mathrm{MM}} = [3 \succ 2 \succ 1 \succ 4 \succ$ others]. It follows that in $P_{\mathrm{MM}} - R_{\mathrm{MM}}$, the min-score of 2 is strictly higher than the min-score of any other alternative, which means that $r(P_{\mathrm{MM}} \setminus \{R_{\mathrm{MM}}\}) = \{2\}$. Notice that $2 \succ_{R_{\mathrm{MM}}} 1$, which means that $\mathrm{PAR}(r, P_{\mathrm{MM}}) = 0$. See the left graph in Figure 6 for an illustration.

- If $r(P_{\mathrm{MM}}) = \{2\}$, then we let $R_{\mathrm{MM}} = \mathrm{Rev}\,(3 \succ 2 \succ 1 \succ 4 \succ$ others). It follows that in $P_{\mathrm{MM}} - R_{\mathrm{MM}}$, the min-score of 1 is strictly higher than any the min-score of other alternatives, which mean that $r(P_{\mathrm{MM}} \setminus \{R_{\mathrm{MM}}\}) = \{1\}$. Notice that $1 \succ_{R_{\mathrm{MM}}} 2$, which again means that $\mathrm{PAR}(r, P_{\mathrm{MM}}) = 0$. See the right graph in Figure 6 for an illustration.

Let $\vec{t}_1 = \mathrm{Sign}_{\vec{H}_{\mathrm{EO}}}(P_{\mathrm{MM}})$, $R = R_{\mathrm{MM}}$ and $\vec{t}_2 = \mathrm{Sign}_{\vec{H}_{\mathrm{EO}}}(P_{\mathrm{MM}} \setminus \{R_{\mathrm{MM}}\})$ . We have $\vec{t}_1 \oplus \vec{t}_2 \in \mathrm{Vio}^r_{\mathrm{PAR}}(n) \neq \emptyset$, which means that the 1 case of Lemma 3 does not hold. The VL case of Lemma 3 does not hold because $\vec{t}_1 \oplus \vec{t}_2 \trianglelefteq \mathrm{Sign}_{\vec{H}_{\mathrm{EO}}}(\pi_{\mathrm{uni}})$ and $\pi_{\mathrm{uni}} \in \mathrm{CH}(\Pi)$.

**Prove $\dim(\mathcal{H}^{\vec{t}_{\mathrm{MM}}}_{\le 0}) = m! - 1$.** Let $e_1 = (4,1)$ and $e_2 = (3,2)$. Notice $[\vec{t}_1]_{(e_1, e_2)} = [\vec{t}_1]_{(e_2, e_1)} = 0$, where $[\vec{t}_1]_{(e_1, e_2)}$ is the $(e_1, e_2)$ component of $\vec{t}_1$, and all other components of $\vec{t}_1$ are non-zero. Also notice that $\vec{t}_2$ is a refinement of $\vec{t}_1$. This means that $\vec{t}_1 \oplus \vec{t}_2 = \vec{t}_1$. Notice that $\mathrm{Hist}(P_{\mathrm{MM}})$ is an inner point of $\mathcal{H}^{\vec{t}_1}_{\le 0}$, such that all inequalities are strict except the two inequalities about $e_1$ and $e_2$. This means that the essential equalities of $\mathbf{A}^{\vec{t}_1 \oplus \vec{t}_2}$ are equivalent to

$$(\mathrm{Pair}_{4,1} - \mathrm{Pair}_{3,2}) \cdot \vec{x} = \vec{0}$$

Therefore, $\dim(\mathcal{H}^{\vec{t}_1 \oplus \vec{t}_2}_{\le 0}) = m! - 1$.

The maximin part of the proposition when $2 \nmid n$ then follows after Lemma 3. When $2 \mid n$, we only need to modify $G_{\mathrm{MM}}$ in Figure 6 by increasing all positive weights by 1.

**Ranked Pairs: $r$ refines $\overline{\mathrm{RP}}$.** The proof is similar to the proof of the maximin part, except that a different graph $G_{\mathrm{RP}}$ (with weight $w_{\mathrm{RP}}$) is used, as shown in the middle graph in Figure 7. Formally, when $2 \nmid n$, let $G_{\mathrm{RP}}$ denote the following weighted directed graph, where the weights on all edges are odd and different, except on $4 \to 1$ and $3 \to 4$.

- $w_{\mathrm{RP}}(4,1) = w_{\mathrm{RP}}(3,4) = 9$, $w_{\mathrm{RP}}(1,2) = 5$, $w_{\mathrm{RP}}(1,3) = 13$, $w_{\mathrm{RP}}(2,4) = 17$, and $w_{\mathrm{RP}}(2,3) = 21$;

- for any $5 \le i \le m$, $w_{\mathrm{RP}}(1,i) \ge 25$, $w_{\mathrm{RP}}(2,i) \ge 25$, $w_{\mathrm{RP}}(3,i) \ge 25$, and $w_{\mathrm{RP}}(4,i) \ge 25$;

- the weights on other edges are assigned arbitrarily. Moreover, the difference between any pair of edges is at least 4, except that the weights on $4 \to 1$ and $3 \to 4$ are the same.

See the middle graph in Figure 7 for an example of $m = 5$.

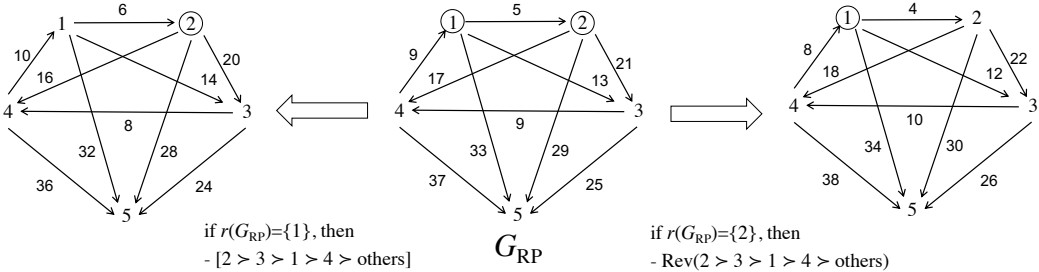

Figure 7: WMGs for ranked pairs. $\overline{\mathrm{RP}}$ (co)-winners are circled.

Again, according to McGarvey's theorem [36] that for any $n > m^4$ and $2 \nmid n$, there exists an $n$-profile $P_{\mathrm{RP}}$ whose WMG is $G_{\mathrm{RP}}$. Therefore, for any $n > m^4 + 2$ and $2 \nmid n$, there exists an $n$-profile $P_{\mathrm{RP}}$ whose WMG is $G_{\mathrm{RP}}$, and $P_{\mathrm{RP}}$ includes the following two rankings:

$$[2 \succ 3 \succ 1 \succ 4 \succ \text{others}], \mathrm{Rev}\,(3 \succ 2 \succ 1 \succ 4 \succ \text{others})$$

We now show that $\mathrm{PAR}(r, P_{\mathrm{RP}}) = 0$, which implies that the 1 case does not happen. Notice that depending on how the tie between $3 \to 4$ and $4 \to 1$ are broken, the $\overline{\mathrm{RP}}$ winner can be 1 or 2, which means that $\overline{\mathrm{RP}}(P_{\mathrm{RP}}) = \{1, 2\}$.

- If $r(P_{\mathrm{RP}}) = \{1\}$, then we let $R_{\mathrm{RP}} = [2 \succ 3 \succ 1 \succ 4 \succ \text{others}]$. It follows that in WMG$(P_{\mathrm{RP}} - R_{\mathrm{RP}})$, $4 \to 1$ has higher weight than $3 \to 4$, which means that $4 \to 1$ is fixed before $3 \to 4$, and therefore $r(P_{\mathrm{RP}} \setminus \{R_{\mathrm{RP}}\}) = \{2\}$. Notice that $2 \succ_{R_{\mathrm{RP}}} 1$, which means that $\mathrm{PAR}(r, P_{\mathrm{RP}}) = 0$. See the left graph in Figure 7 for an illustration.

- If $r(P_{\mathrm{RP}}) = \{2\}$, then we let $R_{\mathrm{RP}} = \mathrm{Rev}\,(2 \succ 3 \succ 1 \succ 4 \succ \text{others})$. It follows that in WMG$(P_{\mathrm{RP}} \setminus \{R_{\mathrm{RP}}\})$, $3 \to 4$ has higher weight than $4 \to 1$, which means $r(P_{\mathrm{RP}} - R_{\mathrm{RP}}) = \{1\}$. Notice that $1 \succ_{R_{\mathrm{RP}}} 2$, which means that $\mathrm{PAR}(r, P_{\mathrm{RP}}) = 0$. See the right graph in Figure 7 for an illustration.

The proof for $\ell_n = 1$ is similar to the proof for the maximin part. The only difference is that now let $e_1 = (4, 1)$, $e_2 = (3, 4)$, $\vec{t}_1 = \text{Sign}_{\vec{H}_{\text{EO}}}(P_{\text{RP}})$, and $\vec{t}_2 = \text{Sign}_{\vec{H}_{\text{EO}}}(P_{\text{RP}} \setminus \{R_{\text{RP}}\})$. When $2 \mid n$, we only need to modify $G$ in Figure 6 (b) such that all positive weights are increased by 1.

**Schulze:** $r$ **refines** $\overline{\text{Sch}}$. The proof is similar to the proof of the maximin part, except that a different graph $G_{\text{Sch}}$ is used, as shown in the middle graph in Figure 8. Formally, when $2 \nmid n$, let $G_{\text{Sch}}$ denote the following weighted directed graph, where the weights on all edges are odd and different, except on $4 \to 1$ and $2 \to 3$.

- $w_{\text{Sch}}(4, 1) = w_{\text{Sch}}(2, 3) = 9$, $w_{\text{Sch}}(1, 2) = 13$, $w_{\text{Sch}}(1, 3) = 5$, $w_{\text{Sch}}(2, 4) = 1$, and $w_{\text{Sch}}(3, 4) = 17$;

- for any $5 \le i \le m$, $w_{\text{Sch}}(1, i) \ge 21$, $w_{\text{Sch}}(2, i) \ge 21$, $w_{\text{Sch}}(3, i) \ge 21$, and $w_{\text{Sch}}(4, i) \ge 21$;

- the weights on other edges are assigned arbitrarily. Moreover, the difference between any pair of edges is at least 4, except that the weights on $4 \to 1$ and $3 \to 4$ are the same.

See the middle graph in Figure 8 for an example of $m = 5$.

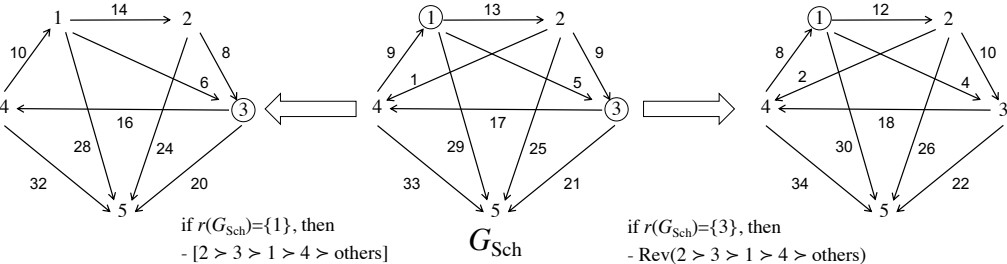

if $r(G_{\text{Sch}})=\{1\}$, then
- $[2 \succ 3 \succ 1 \succ 4 \succ \text{others}]$

$G_{\text{Sch}}$

if $r(G_{\text{Sch}})=\{3\}$, then
- $\text{Rev}(2 \succ 3 \succ 1 \succ 4 \succ \text{others})$

Figure 8: WMGs for Schulze. $\overline{\text{Sch}}$ (co)-winners are circled.

Again, according to McGarvey's theorem [36] that for any $n > m^4$ and $2 \nmid n$, there exists an $n$-profile $P_{\text{Sch}}$ whose WMG is $G_{\text{Sch}}$. Therefore, for any $n > m^4 + 2$ and $2 \nmid n$, there exists an $n$-profile $P_{\text{Sch}}$ whose WMG is $G_{\text{Sch}}$ and $P_{\text{Sch}}$ includes the following two rankings:

$$[2 \succ 3 \succ 1 \succ 4 \succ \text{others}], \text{Rev}(3 \succ 2 \succ 1 \succ 4 \succ \text{others})$$

We now show that $\text{PAR}(r, P_{\text{Sch}}) = 0$, which implies that the 1 case does not happen. Notice that $s[1, 3] = s[3, 1] = 9$, and for any alternative $a \in \mathcal{A} \setminus \{1, 3\}$ we have $s[1, a] > s[a, 1]$. Therefore, $\overline{\text{Sch}}(P_{\text{Sch}}) = \{1, 3\}$.

- If $r(P_{\text{Sch}}) = \{1\}$, then we let $R_{\text{Sch}} = [2 \succ 3 \succ 1 \succ 4 \succ \text{others}]$. It follows that in $P_{\text{Sch}} - R_{\text{Sch}}$ we have $s[1, 3] = 8 < 10 = s[3, 1]$, which means that $r(P_{\text{Sch}} \setminus \{R_{\text{Sch}}\}) = \{3\}$. Notice that $3 \succ_{R_{\text{Sch}}} 1$, which means that $\text{PAR}(r, P_{\text{Sch}}) = 0$. See the left graph in Figure 8 for an illustration.

- If $r(P_{\text{Sch}}) = \{3\}$, then we let $R_{\text{Sch}} = \text{Rev}(2 \succ 3 \succ 1 \succ 4 \succ \text{others})$. It follows that in $P_{\text{Sch}} \setminus \{R_{\text{Sch}}\}$, we have $s[1, 3] = 10 > 9 = s[3, 1]$, which means that $r(P_{\text{Sch}} - R_{\text{Sch}}) = \{1\}$. Notice that $1 \succ_{R_{\text{Sch}}} 3$, which means that $\text{PAR}(r, P_{\text{Sch}}) = 0$. See the right graph in Figure 8 for an illustration.

The proof for $\ell_n = 1$ is similar to the proof for the maximin part. The only difference is that now let $e_1 = (4, 1)$, $e_2 = (2, 3)$, $\vec{t}_1 = \text{Sign}_{\vec{H}_{\text{EO}}}(P_{\text{Sch}})$, and $\vec{t}_2 = \text{Sign}_{\vec{H}_{\text{EO}}}(P_{\text{Sch}} \setminus \{R_{\text{Sch}}\})$. When $2 \mid n$, we only need to modify $G_{\text{Sch}}$ in Figure 8 such that all positive weights are increased by 1.

This completes the proof of Theorem 3. $\qquad\square$

### F.3 Proof of Theorem 4

A voting rule $r$ is said to be *UMG-based*, if the winner only depends on UMG of the profile. Formally, $r$ is UMG-based if for all pairs of profiles $P_1$ and $P_2$ such that $\text{UMG}(P_1) = \text{UMG}(P_2)$, we have $r(P_1) = r(P_2)$.

**Theorem 4** (**Semi-random** PAR: **Copeland$_\alpha$**). *For any $m \geq 4$, any UMG-based int-GSR refinement of $\overline{Cd_\alpha}$, denoted by $Cd_\alpha$, and any strictly positive and closed $\Pi$ over $\mathcal{L}(\mathcal{A})$ with $\pi_{uni} \in CH(\Pi)$, there exists $N \in \mathbb{N}$ such that for every $n \geq N$,*

$$\widetilde{\text{PAR}}_\Pi^{\min}(Cd_\alpha, n) = 1 - \Theta(\frac{1}{\sqrt{n}})$$

*Proof.* Because $\text{Cd}_\alpha$ is UMG-based, we can represent $\text{Cd}_\alpha$ as a GSR with the $\vec{H}_{\text{Cd}_\alpha}$ defined in Definition 13, which consists of $\binom{m}{2}$ hyperplanes that represents the UMG of the profile. The high-level idea behind the proof is similar to the proof of Theorem 3: we first explicitly construct a violation of PAR under $\text{Cd}_\alpha$, then show that the dimension of the characteristic cone of the corresponding polyhedron is $m! - 1$.

Let $G^*$ denote the complete unweighted directed graph over $\mathcal{A}$ that consists of the following edges.

- $1 \rightarrow 2$, $2 \rightarrow 3$, $3 \rightarrow 1$.

- For any $i \in \{4, \ldots, m\}$, there are three edges $1 \rightarrow i$, $2 \rightarrow i$, $3 \rightarrow i$.

- The edges among alternatives in $i \in \{4, \ldots, m\}$ are assigned arbitrarily.

For example, Figure 9 (a) illustrates $G^*$ for $m = 4$. Let $P$ denote any profile whose UMG is $G^*$. It

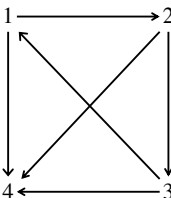

Figure 9: $G^*$ for Copeland with $m = 4$.

is not hard to verify that $\overline{\text{Cd}_\alpha}(P) = \{1, 2, 3\}$. W.l.o.g. let $\text{Cd}_\alpha(P) = \{1\}$.

**$2 \nmid n$ case.** The proof is done for the following two sub-cases: $\alpha > 0$ and $\alpha = 0$.

**$2 \nmid n$ and $\alpha > 0$.** Let $G_{\text{Cd}_\alpha}$ (with weights $w_{\text{Cd}_\alpha}$) denote the following weighted directed graph over $\mathcal{A}$ whose UMG is $G^*$, the weight on $2 \rightarrow 3$ is 1, and the weights on other edges are 3 or $-3$.

- $w_{\text{Cd}_\alpha}(2, 3) = 1$ and $w_{\text{Cd}_\alpha}(3, 1) = w_{\text{Cd}_\alpha}(1, 2) = 3$.

- For any $4 \leq i \leq m$, $w_{\text{Cd}_\alpha}(1, i) = w_{\text{Cd}_\alpha}(2, i) = w_{\text{Cd}_\alpha}(3, i) = 3$.

- The weights on other edges are 3 or $-3$.

See Figure 10 (a) for an example of $G_{\text{Cd}_\alpha}$. According to McGarvey's theorem [36] that for any $n > m^4$ and $2 \nmid n$, there exists an $n$-profile $P_{\text{Cd}_\alpha}$ whose WMG is $G_{\text{Cd}_\alpha}$. Therefore, for any $n > m^4 + 2$ and $2 \nmid n$, there exists an $n$-profile $P_{\text{Cd}_\alpha}$ whose WMG is $G_{\text{Cd}_\alpha}$, and $P_{\text{Cd}_\alpha}$ includes the following two rankings.

$$[4 \succ 2 \succ 3 \succ 1 \succ \text{others}], \text{Rev}(4 \succ 2 \succ 3 \succ 1 \succ \text{others})$$

We now show that $\text{PAR}(r, P_{\text{Cd}_\alpha}) = 0$, which implies that the 1 case Lemma 3 does not hold. Let $R_{\text{Cd}_\alpha} = [4 \succ 2 \succ 3 \succ 1 \succ \text{others}]$. Notice that in the profile $P_{\text{Cd}_\alpha} - R_{\text{Cd}_\alpha}$, the Copeland$_\alpha$ score of alternative 3 is $m - 2 + \alpha$, which is strictly higher than the Copeland$_\alpha$ score of alternative 1, which

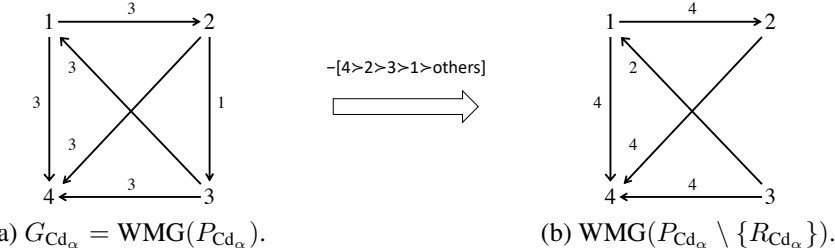

(a) $G_{\text{Cd}_\alpha} = \text{WMG}(P_{\text{Cd}_\alpha})$.

(b) $\text{WMG}(P_{\text{Cd}_\alpha} \setminus \{R_{\text{Cd}_\alpha}\})$.

Figure 10: $G_{\text{Cd}_\alpha}$ and $\text{WMG}(P_{\text{Cd}_\alpha} \setminus \{P_{\text{Cd}_\alpha}\})$ for $2 \nmid n$ and $\alpha > 0$.

is $m - 2$. Therefore, $\text{Cd}_\alpha(P_{\text{Cd}_\alpha} \setminus \{R_{\text{Cd}_\alpha}\}) = \{3\}$. See Figure 10 (b) for $\text{WMG}(P_{\text{Cd}_\alpha} \setminus \{R_{\text{Cd}_\alpha}\})$. Notice that $3 \succ_{R_{\text{Cd}_\alpha}} 1$, which means that the $\text{PAR}(r, P_{\text{Cd}_\alpha}) = 0$.

Therefore, the 1 case of Lemma 3 does not hold. Let $\vec{t}_1 = \text{Sign}_{\vec{H}_{\text{Cd}_\alpha}}(P_{\text{Cd}_\alpha})$ and $\vec{t}_2 = \text{Sign}_{\vec{H}_{\text{Cd}_\alpha}}(P_{\text{Cd}_\alpha} \setminus \{R_{\text{Cd}_\alpha}\})$. The VL case of Lemma 3 does not hold because $\vec{t}_1 \oplus \vec{t}_2 \trianglelefteq \text{Sign}_{\vec{H}_{\text{Cd}_\alpha}}(\pi_{\text{uni}})$ and $\pi_{\text{uni}} \in \text{CH}(\Pi)$.

Next, we prove that $\dim(\mathcal{H}_{\leq 0}^{\vec{t}_1 \oplus \vec{t}_2}) = m! - 1$. Notice that $[\vec{t}_1]_{(2,3)} = +$ and $[\vec{t}_2]_{(2,3)} = 0$, and all other components of $\vec{t}_1$ and $\vec{t}_2$ are the same and are non-zero. Therefore, $\vec{t}_1$ is a refinement of $\vec{t}_2$, which means that $\vec{t}_1 \oplus \vec{t}_2 = \vec{t}_2$. Notice that $\text{Hist}(P_{\text{Cd}_\alpha})$ is an inner point of $\mathcal{H}_{\leq 0}^{\vec{t}_2}$, in the sense that all inequalities are strict except the inequalities about $(2, 3)$. This means that the essential equalities of $\mathbf{A}^{\vec{t}_1 \oplus \vec{t}_2}$ are equivalent to $\text{Pair}_{2,3} \cdot \vec{x} = \vec{0}$. Therefore, $\dim(\mathcal{H}_{\leq 0}^{\vec{t}_2}) = \dim(\mathcal{H}_{\leq 0}^{\vec{t}_1 \oplus \vec{t}_2}) = m! - 1$. This proves the proposition when $2 \nmid n$, $\alpha > 0$, and $\text{Cd}_\alpha(P) = \{1\}$.

If $\text{Cd}_\alpha(P) = \{2\}$ (respectively, $\text{Cd}_\alpha(P) = \{3\}$), then we simply switch the weights on $2 \to 3$ and $3 \to 1$ (respectively, $2 \to 3$ and $1 \to 2$) in Figure 9 (b), and the rest of the proof is similar to the $\text{Cd}_\alpha(P) = \{1\}$ case. This proves Theorem 4 for $2 \nmid n$ and $\alpha > 0$.

**$2 \nmid n$ and $\alpha = 0$.** Let $G_{\text{Cd}_\alpha}$ (with weights $w_{\text{Cd}_\alpha}$) denote the following weighted directed graph over $\mathcal{A}$ whose UMG is $G^*$ as illustrated in Figure 9 (a).

- $w_{\text{Cd}_\alpha}(2, 3) = w_{\text{Cd}_\alpha}(3, 1) = w_{\text{Cd}_\alpha}(1, 2) = 3$.
- For any $4 \leq i \leq m$, $w_{\text{Cd}_\alpha}(1, i) = w_{\text{Cd}_\alpha}(2, i) = w_{\text{Cd}_\alpha}(3, i) = 3$, except $w_{\text{Cd}_\alpha}(4, 1) = 1$.
- The weights on edge between $\{4, \ldots, m\}$ are 3 or $-3$.

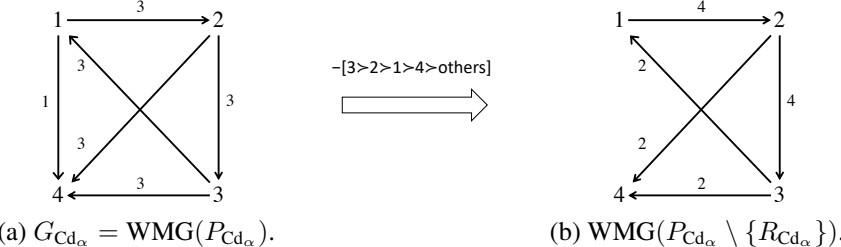

(a) $G_{\text{Cd}_\alpha} = \text{WMG}(P_{\text{Cd}_\alpha})$.

(b) $\text{WMG}(P_{\text{Cd}_\alpha} \setminus \{R_{\text{Cd}_\alpha}\})$.

Figure 11: $G_{\text{Cd}_\alpha}$ and $\text{WMG}(P_{\text{Cd}_\alpha} \setminus \{P_{\text{Cd}_\alpha}\})$ for $2 \nmid n$ and $\alpha = 0$.

See Figure 11 (a) for an example of $G_{\text{Cd}_\alpha}$. According to McGarvey's theorem [36] that for any $n > m^4$ and $2 \nmid n$, there exists an $n$-profile $P_{\text{Cd}_\alpha}$ whose WMG is $G_{\text{Cd}_\alpha}$. Therefore, for any $n > m^4 + 2$ and $2 \nmid n$, there exists an $n$-profile $P_{\text{Cd}_\alpha}$ whose WMG is $G_{\text{Cd}_\alpha}$ and $P_{\text{Cd}_\alpha}$ includes the following two rankings.

$$[3 \succ 2 \succ 1 \succ 4 \succ \text{others}], \text{Rev}\,(3 \succ 2 \succ 1 \succ 4 \succ \text{others})$$

We now show that $\text{PAR}(\text{Cd}_\alpha, P_{\text{Cd}_\alpha}) = 0$, which implies that the 1 case Lemma 3 does not hold. Let $R_{\text{Cd}_\alpha} = [3 \succ 2 \succ 1 \succ 4 \succ \text{others}]$. Notice that in the profile $P_{\text{Cd}_\alpha} \setminus \{R_{\text{Cd}_\alpha}\}$, the Copeland$_\alpha$

score of alternative 1 is $m - 3 + \alpha = m - 3$, which is strictly higher than the Copeland$_\alpha$ score of alternative 2 and 3, which means that $\mathrm{Cd}_\alpha(P_{\mathrm{Cd}_\alpha} - R_{\mathrm{Cd}_\alpha}) \subseteq \{2, 3\}$. See Figure 11 (b) for an example of $\mathrm{WMG}(P_{\mathrm{Cd}_\alpha} \setminus \{R_{\mathrm{Cd}_\alpha}\})$. Notice that $2 \succ_{R_{\mathrm{Cd}_\alpha}} 1$ and $3 \succ_{R_{\mathrm{Cd}_\alpha}} 1$, which means that $\mathrm{PAR}(\mathrm{Cd}_\alpha, P_{\mathrm{Cd}_\alpha}) = 0$.

The proofs for $\ell_n = 1$, the $\mathrm{Cd}_\alpha(P) = \{2\}$ case, and the $\mathrm{Cd}_\alpha(P) = \{3\}$ case are similar to their counterparts for the "$2 \nmid n$ and $\alpha = 0$" case above.

**$2 \mid n$.** The proof for the $2 \mid n$ case is similar to the proof of the $2 \nmid n$ case with the following modifications. The $n$-profile $P_{\mathrm{Cd}_\alpha}$ where $\mathrm{PAR}$ is violated is obtained from the profile in the $2 \nmid n$ plus $\mathrm{Rev}(R_{\mathrm{Cd}_\alpha})$. Below we present the full proof for the case of $2 \mid n$ and $\alpha > 0$ for example. The other cases can be proved similarly.

**$2 \mid n$ and $\alpha > 0$.** W.l.o.g. suppose $\mathrm{Cd}_\alpha(G^*) = \{1\}$. Let $G_{\mathrm{Cd}_\alpha}$ (with weights $w_{\mathrm{Cd}_\alpha}$) denote the weighted directed graph in Figure 10 (a). According to McGarvey's theorem [36] that for any $n > m^4$ and $2 \mid n$, there exists an $(n-1)$-profile $P'_{\mathrm{Cd}_\alpha}$ whose WMG is $G_{\mathrm{Cd}_\alpha}$. Let

$$P_{\mathrm{Cd}_\alpha} = P'_{\mathrm{Cd}_\alpha} + \mathrm{Rev}\,(4 \succ 2 \succ 3 \succ 1 \succ \text{others})$$

It is not hard to verify that in $P_{\mathrm{Cd}_\alpha}$, the Copeland$_\alpha$ score of alternative 3 is $m - 2 + \alpha$, which is strictly higher than the Copeland$_\alpha$ score of alternative 1, which is $m - 2$. Therefore, $\mathrm{Cd}_\alpha(P_{\mathrm{Cd}_\alpha}) = \{3\}$. Let $R_{\mathrm{Cd}_\alpha} = \mathrm{Rev}\,(4 \succ 2 \succ 3 \succ 1 \succ \text{others})$. Notice that $\mathrm{Cd}_\alpha(P_{\mathrm{Cd}_\alpha} \setminus \{R_{\mathrm{Cd}_\alpha}\}) = \mathrm{Cd}_\alpha(G^*) = \{1\}$ and $1 \succ_{R_{\mathrm{Cd}_\alpha}} 3$, which means that $\mathrm{PAR}(\mathrm{Cd}_\alpha, P_{\mathrm{Cd}_\alpha}) = 0$. Therefore, the 1 case in Lemma 3 does not hold. Let $\vec{t_1} = \mathrm{Sign}_{\vec{H}_{\mathrm{Cd}_\alpha}}(P_{\mathrm{Cd}_\alpha})$ and $\vec{t_2} = \mathrm{Sign}_{\vec{H}_{\mathrm{Cd}_\alpha}}(P_{\mathrm{Cd}_\alpha} \setminus \{R_{\mathrm{Cd}_\alpha}\})$. Like in other cases, the VL case of Lemma 3 does not holds because $\vec{t_1} \oplus \vec{t_2} \trianglelefteq \mathrm{Sign}_{\vec{H}_{\mathrm{Cd}_\alpha}}(\pi_{\mathrm{uni}})$.

Next, we prove that $\dim(\mathcal{H}_{\leq 0}^{\vec{t_1} \oplus \vec{t_2}}) = m! - 1$. Notice that $[\vec{t_1}]_{(2,3)} = 0$ and $[\vec{t_2}]_{(2,3)} = +$, and all other components of $\vec{t_1}$ and $\vec{t_2}$ are the same and are non-zero. Therefore, $\vec{t_1}$ is a refinement of $\vec{t_2}$, which means that $\vec{t_1} \oplus \vec{t_2} = \vec{t_1}$. Notice that $\mathrm{Hist}(P_{\mathrm{Cd}_\alpha})$ is an inner point of $\mathcal{H}_{\leq 0}^{\vec{t_1}}$, in the sense that all inequalities are strict except the inequalities about $(2, 3)$. This means that the essential equalities of $\mathbf{A}^{\vec{t_1} \oplus \vec{t_2}}$ are equivalent to

$$\mathrm{Pair}_{2,3} \cdot \vec{x} = \vec{0} \text{ and } - \mathrm{Pair}_{2,3} \cdot \vec{x} = \vec{0}$$

Therefore, $\dim(\mathcal{H}_{\leq 0}^{\vec{t_1} \oplus \vec{t_2}}) = m! - 1$, which means that $\ell_n = -(m! - (m! - 1)) = 1$. The $2 \mid n$ and $\alpha > 0$ case follows after Lemma 3.

The proof for other subcases of $2 \mid n$ are similar to the proof of $2 \mid n$ and $\alpha > 0$ case above. This completes the proof of Theorem 4. $\qquad\square$

## F.4 Proof of Theorem 5

**Theorem 5 (Semi-random PAR: int-MRSE).** *Given $m \geq 4$, any int-MRSE $\bar{r}$, any int-GSR $r$ that is a refinement of $\bar{r} = (\bar{r}_2, \ldots, \bar{r}_m)$, and any strictly positive and closed $\Pi$ over $\mathcal{L}(\mathcal{A})$ with $\pi_{uni} \in CH(\Pi)$, there exists $N \in \mathbb{N}$ such that for every $n \geq N$,*

$$\widetilde{\mathrm{PAR}}_\Pi^{\min}(r, n) = 1 - \Theta\left(\frac{1}{\sqrt{n}}\right)$$

*Proof.* The intuition behind the proof is similar to the proof of Theorem 3. Indeed, Lemma 3 can be applied to $r$, but it is unclear how to characterize $\ell_n$. Therefore, in this proof we do not directly characterize $dim(\mathcal{H}_{\leq 0}^{\vec{t}})$ as in the proof of Theorem 3, but will instead define another polyhedron $\mathcal{H}^r$ to characterize a set of sufficient conditions for $\mathrm{PAR}$ to be violated—and the dimension of the new polyhedron is easy to analyze. Let us start with defining sufficient conditions on a profile $P$ for $\mathrm{PAR}$ to be violated under any refinement of $\bar{r}$.

**Condition 1 (Sufficient conditions: violation of PAR under an MRSE rule).** *Given an MRSE $\bar{r}$, a profile $P$ satisfies the following conditions during the execution of $\bar{r}$.*

*(1) For every $1 \leq i \leq m - 4$, in the $i$-th round, alternative $i + 4$ drops out.*

*(2) In round $m-3$, 1 has the highest score, 2 has the second highest score, and 3 and 4 are tied for the last place.*

*(3) If 3 is eliminated in round $m-3$, then 2 and 4 are eliminated in round $m-2$ and $m-1$, respectively, which means that the winner is 1.*

*(4) If 4 is eliminated in round $m-3$, then 1 and 3 are eliminated in round $m-2$ and $m-1$, respectively, which means that the winner is 2.*

*(5) P contains at least one vote $[4 \succ 2 \succ 1 \succ 3 \succ \text{others}]$ and at least one vote $[3 \succ 1 \succ 2 \succ 4 \succ \text{others}]$, where "others" represents $5 \succ \cdots \succ m$.*

*(6) All losers described above, except in (2), are "robust", in the sense that after removing any vote from P, they are still the unique losers.*

Let us verify that for any profile $P$ that satisfies Condition 1, $\text{PAR}(r, P) = 0$. It is not hard to see that $\overline{r}(P) = \{1, 2\}$. If $r(P) = \{1\}$, then let $R_r = [4 \succ 2 \succ 1 \succ 3 \succ \text{others}]$. This means that when any voter whose preferences are $R_r$ abstain from voting, alternative 4 drops out in round $m-3$ of $(P \setminus \{R_r\})$, and consequently 2 becomes the winner. Notice that $2 \succ_{R_r} 1$, which means that $\text{PAR}(r, P) = 0$. Similarly, if $r(P) = \{2\}$, then let $R_r = [3 \succ 1 \succ 2 \succ 4 \succ \text{others}]$, which means that 3 drops out in round $m-3$ of $(P \setminus \{R_r\})$, and 1 becomes the winner. Notice that $1 \succ_{R_r} 2$. Again, we have $\text{PAR}(r, P) = 0$. The procedures of executing $\overline{r}$ under $P$ and $(P \setminus \{R_r\})$ are represented in Figure 12.

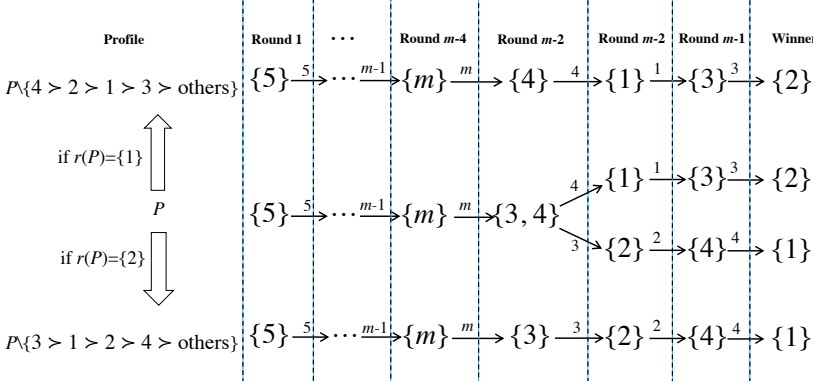

Figure 12: Executing $\overline{r}$ for a profile that satisfies Condition 1.

The rest of the proof proceeds as follows. In Step 1 below, We will prove by construction that for every sufficiently large $n$, there exists an $n$-profile $P_r$ that satisfies Condition 1. Then in Step 2, we formally define $\mathcal{H}^{\overline{r}}$ to represent profiles that satisfy Condition 1. Finally, in Step 3, we show that $\dim(\mathcal{H}^{\overline{r}}_{<0}) = m! - 1$ because there is essentially only one equality (in Condition 1 (2)). Theorem 5 then follows after 1 minus the polynomial case of the inf part of [55, Theorem 2].

**Step 1: define $P_r$.** Before defining $P_r$, we first define a profile $P^*$ that consists of a constant and odd number of votes in Steps 1.1–1.3. We then prove that PAR is violated at $P^*$ in Step 1.4 and 1.5, where in Step 1.4 we show that $\overline{r}(P) = \{1, 2\}$ and in Step 1.5 we point out a violation of PAR depending on $r(P^*)$. Then in Step 1.6, we show how to expand $P^*$ to an $n$-profile $P_r$ for any sufficiently large $n$.

Let $P^* = P_1 + P_2 + P_3$, where $P_1$ consists of even number of votes and is designed to guarantee Condition 1 (1), i.e., $5, \ldots, m$ are eliminated in the first $m-4$ rounds, respectively. This means that in the beginning of round $m-3$, the remaining alternatives are $\{1, 2, 3, 4\}$. $P_2$ consists of an odd number of votes and is designed to guarantee Condition 1 (2), i.e., in round $m-3$, $\overline{r}_4$ outputs the weak order $[1 \succ 2 \succ 3 = 4]$. $P_3$ consists of an even number of votes and is designed to guarantee Condition 1 (3) and (4), i.e., if 3 (respectively, 4) is eliminated then 1 (respectively, 2) wins.

**Step 1.1: define $P_1$.** Let $P_1^1$ denote the following profile of $(24m(m-4)! + \frac{(m+5)(m-4)}{2}(m-1)!)$ votes.

$$P_1^1 = m \times \{[R_1 \succ R_2 : \forall R_1 \in \mathcal{L}(\{1,2,3,4\}), R_2 \in \mathcal{L}(\{5,\ldots,m\})\} \cup \bigcup_{i=5}^{m} i \times \{[i \succ R_2] : \forall R_2 \in \mathcal{L}(\mathcal{A}\backslash\{i\})\}$$

For every $2 \le i \le m$, let the scoring vector of $\bar{r}_i$ be $(s_1^i, \ldots, s_i^i)$. For example, the scoring vector of $\bar{r}_4$ is $(s_1^4, s_2^4, s_3^4, s_4^4)$. We let $P_1 = (s_1^4 - s_4^4 + 1)|P_2| \times P_1^1$, where $|P_2|$ is the number of votes in $P_2$, which is a constant and will become clear after Step 1.2.

**Step 1.2: define $P_2$.** The main challenge in this step is to use an odd number of votes to define $P_2$ such that in round $m-3$, the score of 1 is strictly higher than the score of 2, which is strictly higher than the score of 3 and 4. We first define the following 8-profile, denoted by $P_2^1$.

$$P_2^1 = \{[1 \succ \text{others} \succ 3 \succ 4 \succ 2], [1 \succ \text{others} \succ 4 \succ 3 \succ 2],$$
$$3 \times [1 \succ \text{others} \succ 2 \succ 4 \succ 3], 3 \times [2 \succ \text{others} \succ 1 \succ 3 \succ 4]\}$$

The numbers of times alternatives $\{1, 2, 3, 4\}$ are ranked in each position in $P_2^1|_{\{1,2,3,4\}}$ are indicated in Table 5.

| Alternative | 1st | 2nd | 3rd | 4th |
|:-----------:|:---:|:---:|:---:|:---:|
| 1 | 5 | 3 | 0 | 0 |
| 2 | 3 | 3 | 0 | 2 |
| 3 | 0 | 1 | 4 | 3 |
| 4 | 0 | 1 | 4 | 3 |

Table 5: Number of times each alternative is ranked in each position in $P_2^1|_{\{1,2,3,4\}}$.

Next, we define a profile $P_2^2$ that consists of an odd number of votes where the scores of 3 and 4 are equal. Let $d_1 = s_1^4 - s_2^4$ and $d_2 = s_2^4 - s_3^4$. The construction is done in the following three cases.

- If $d_1 = 0$, then we let $P_2^2$ consist of a single vote $[3 \succ 4 \succ 1 \succ 2 \succ \text{others}]$.

- If $d_1 \neq 0$ and $d_2 = 0$, then we let $P_2^2$ consist of a single vote $[1 \succ 3 \succ 4 \succ 2 \succ \text{others}]$.

- If $d_1 \neq 0$ and $d_2 \neq 0$, then we let $d_1' = d_1/\gcd(d_1, d_2)$ and $d_2' = d_2/\gcd(d_1, d_2)$, where $\gcd(d_1, d_2)$ is the greatest common divisor of $d_1$ and $d_2$. It follows that at least one of $d_1'$ and $d_2'$ is an odd number.

  - If $d_1'$ is odd, then we let
    $$P_2^2 = (d_1' + d_2') \times [1 \succ 3 \succ 4 \succ 2 \succ \text{others}] + d_2' \times [4 \succ 1 \succ 3 \succ 2 \succ \text{others}]$$

  - Otherwise, we must have $d_1'$ is even and $d_2'$ is odd. Then, we let
    $$P_2^2 = (d_1' + d_2') \times [3 \succ 4 \succ 1 \succ 2 \succ \text{others}] + d_1' \times [4 \succ 1 \succ 3 \succ 2 \succ \text{others}]$$

It is not hard to verify that in either case $P_2^2$ consists of an odd number of votes, and the score of 3 and 4 are equal under $P_2^2$. To guarantee that 3 and 4 have the lowest $\bar{r}_4$ scores in $P_2|_{\{1,2,3,4\}}$, we include sufficiently many copies of $P_2^1$ in $P_2$. Formally, let

$$P_2 = (|P_2^2| + 1) \times P_2^1 + P_2^2$$

**Step 1.3: define $P_3$.** We let $P_3 = ((s_1 - s_3)|P_2| + 1) \times P_3^*$, where $P_3^* = P_3^{*1} + P_3^{*2}$ is the 36-profile defined as follows. $P_3^{*1}$ consists of 12 votes, where each alternative in $\{1, 2, 3, 4\}$ is ranked in the top in three votes, followed by the remaining three alternatives in a cyclic order.

$$P_3^{*1} = \{[1 \succ 2 \succ 3 \succ 4 \succ \text{others}], [1 \succ 3 \succ 4 \succ 2 \succ \text{others}], [1 \succ 4 \succ 2 \succ 3 \succ \text{others}],$$
$$[2 \succ 1 \succ 4 \succ 3 \succ \text{others}], [2 \succ 4 \succ 3 \succ 1 \succ \text{others}], [2 \succ 3 \succ 1 \succ 4 \succ \text{others}],$$
$$[3 \succ 1 \succ 4 \succ 2 \succ \text{others}], [3 \succ 4 \succ 2 \succ 1 \succ \text{others}], [3 \succ 2 \succ 1 \succ 4 \succ \text{others}],$$
$$[4 \succ 1 \succ 2 \succ 3 \succ \text{others}], [4 \succ 2 \succ 3 \succ 1 \succ \text{others}], [4 \succ 3 \succ 1 \succ 2 \succ \text{others}]\}$$

$P_3^{*2}$ consists of 24 votes that are defined in the following three steps. First, we start with $\mathcal{L}(\{1,2,3,4\})$, which consists of 24 votes. Second, we replace $[3 \succ 2 \succ 4 \succ 1]$ and $[4 \succ 1 \succ 3 \succ 2]$ by $[3 \succ 1 \succ 4 \succ 2]$ and $[4 \succ 2 \succ 3 \succ 1]$, respectively. That is, the locations of 1 and 2 are exchanged in the two votes. This is designed to guarantee that the $\bar{r}_4$ scores of all alternative are the same in $P_3^{*2}|_{\{1,2,3,4\}}$, and after 3 is removed, 1's $\bar{r}_3$ score is higher than 2's $\bar{r}_3$ score; and after 4 is removed, 2's $\bar{r}_3$ score is higher than 1's $\bar{r}_3$score. Third, we append the lexicographic order of $\{5,\ldots,m\}$ to the end of each of the 24 rankings. Formally, we define

$$P_3^{*2} = \{R_4 \succ 5 \succ \cdots \succ m : R_4 \in \mathcal{L}(\{1,2,3,4\})\} - [3 \succ 2 \succ 4 \succ 1 \succ \text{others}]$$
$$- [4 \succ 1 \succ 3 \succ 2 \succ \text{others}] + [3 \succ 1 \succ 4 \succ 2 \succ \text{others}] + [4 \succ 2 \succ 3 \succ 1 \succ \text{others}]$$

**Step 1.4: Prove $\bar{r}(P^*) = \{1,2\}$.** Recall that $P^* = P_1 + P_2 + P_3$. Notice that the $P_1$ part guarantees that $\{5,\ldots,m\}$ are dropped out in the first $m-4$ rounds, and the scores of all alternatives in $\{1,2,3,4\}$ are the same under $P_1$ no matter what alternatives are dropped out. Therefore, it suffices to calculate the results of the last three rounds based on $P_2 + P_3$, which is done as follows.

In round $m - 3$, it is not hard to check that every alternative in $\{1,2,3,4\}$ gets the same total score under $P_3$, where each of them is ranked at each position for 9 times. Therefore, due to $P_2$, alternative 3 and 4 are tied for the last place in round $m - 3$.

**If 3 is eliminated in round $m - 3$,** then $P_3^*|_{\{1,2,4\}} = P_3^{*1}|_{\{1,2,4\}} + P_3^{*2}|_{\{1,2,4\}}$ becomes the following.

$$P_3^{*1}|_{\{1,2,4\}} = \{2 \times [1 \succ 4 \succ 2], [1 \succ 2 \succ 4], 2 \times [2 \succ 1 \succ 4], [2 \succ 4 \succ 1],$$
$$[1 \succ 4 \succ 2], [4 \succ 2 \succ 1], [2 \succ 1 \succ 4], 2 \times [4 \succ 1 \succ 2], [4 \succ 2 \succ 1]\}$$
$$P_3^{*2}|_{\{1,2,4\}} = 4 \times \mathcal{L}(\{1,2,4\}) - [2 \succ 4 \succ 1] - [4 \succ 1 \succ 2] + [1 \succ 4 \succ 2] + [4 \succ 2 \succ 1]$$

It is not hard to verify that the numbers of times alternatives $\{1,2,4\}$ are ranked in each position in $P_3^*|_{\{1,2,4\}}$ are as indicated in Table 6 (a).

| Alternative | 1st | 2nd | 3rd |
|---|---|---|---|
| 1 | 13 | 12 | 11 |
| 2 | 11 | 12 | 13 |
| 4 | 12 | 12 | 12 |

(a) 3 is removed.

| Alternative | 1st | 2nd | 3rd |
|---|---|---|---|
| 1 | 11 | 12 | 13 |
| 2 | 13 | 12 | 11 |
| 3 | 12 | 12 | 12 |

(b) 4 is removed.

Table 6: Number of times each alternative is ranked in each position in round $m - 2$.

This means that the score of alternative 2 is strictly lower than the score of 1 or 3, because $s_1^3 - s_3^3 \geq 1$, where the score vector for $\bar{r}_3$ is $(s_1^3, s_2^3, s_3^3)$. Recall that $P_3$ consists of sufficiently large number of copies of $P_3^*$. Therefore, even considering the score difference between alternatives in $P_2$, the score of 2 is still the strictly lowest among $\{1,2,4\}$ in $P^*$ in round $m - 2$. This means that alternative 2 drops in round $m - 2$, and it is easy to check that $1 \succ 4$ in 20 votes in $P_3^*$, which is strictly more than half ($= 16$). This means that 1 is the $r$ winner if 3 is eliminated in round $m - 3$.

**If 4 is eliminated in round $m - 3$,** then $P_3^*|_{\{1,2,3\}} = P_3^{*1}|_{\{1,2,3\}} + P_3^{*2}|_{\{1,2,3\}}$ becomes the following.

$$P_3^{*1}|_{\{1,2,3\}} = \{2 \times [1 \succ 2 \succ 3], [1 \succ 3 \succ 2], 2 \times [2 \succ 3 \succ 1], [2 \succ 1 \succ 3],$$
$$2 \times [3 \succ 2 \succ 1], [3 \succ 1 \succ 2], [1 \succ 2 \succ 3], [2 \succ 3 \succ 1], [3 \succ 1 \succ 2]\}$$
$$P_3^{*2}|_{\{1,2,3\}} = 4 \times \mathcal{L}(\{1,2,3\}) - [3 \succ 2 \succ 1] - [1 \succ 3 \succ 2] + [3 \succ 1 \succ 2] + [2 \succ 3 \succ 1]$$

The numbers of times alternatives $\{1,2,3\}$ are ranked in each position in $P_3^*|_{\{1,2,3\}}$ are as indicated in Table 6 (b). Again, it is not hard to verify that alternative 1 drops in round $m - 2$, and 2 beats 3 in the last round to become the $r$ winner in this case.

**Step 1.5: Prove that PAR is violated at $P^*$.** At a high-level the proof is similar to Step 1.4, and the absent vote is effectively used as a tie breaker between alternatives 3 and 4. Recall that $r$ is a refinement of $\bar{r}$ and it was shown in Step 1.4 that $\bar{r}(P^*) = \{1,2\}$. Therefore, either $r(P^*) = \{1\}$ or $r(P^*) = \{2\}$. The proof is done in the follow two cases.

- If $r(P^*) = \{1\}$, then we let

$$R_r = [4 \succ 2 \succ 1 \succ 3 \succ \text{others}],$$

which is a vote in $P_3^2$. Then in $(P^* \setminus \{R_r\})$, alternative 4 is eliminated in round $m - 3$, and following a similar reasoning as in Step 1.4, we have $r(P^* \setminus \{R_r\}) = \{2\}$. Notice that $2 \succ_{R_r} 1$, which means that PAR is violated at $P^*$.

- If $r(P^*) = \{2\}$, then we let

$$R_r = [3 \succ 1 \succ 2 \succ 4 \succ \text{others}],$$

which is a vote in $P_3^2$. Then in $(P^* \setminus \{R_r\})$, alternative 3 is eliminated in round $m - 3$, and following a similar reasoning as in Step 1.4, we have $r(P^* \setminus \{R_r\}) = \{1\}$. Notice that $1 \succ_{R_r} 2$, which means that PAR is violated at $P^*$.

**Step 1.6: Construct an $n$-profile $P_r$.** The intuition behind the construction is the following. $P_r$ consists of three parts: $P_r^1$, $P_r^2$, and $P_r^3$. $P_r^1$ consists of multiple copies of $P^*$ defined in Steps 1.1-1.3 above, which is used to guarantee that PAR is violated at $P_r$ and the score difference between any pair of alternatives is sufficiently large so that votes in $P_r^3$ does not affect the execution of $r$. $P_r^2$ consists of multiple copies of $\mathcal{L}(\mathcal{A})$. $P_r^3$ consists of no more than $m! - 1$ votes, and $|P_r^3|$ is an even number.

**Define $P_r^1$.** To guarantee that $|P_r^3|$ is even, the definition of $P_r^1$ depends on the parity of $n$. Recall that $P^*$ consists of an odd number of votes. When $2 \mid n$, we let

$$P_r^1 = m! \left( s_1^3 - s_3^3 \right) \times P^*$$

When $2 \nmid n$, we let

$$P_r^1 = \left( m! \left( s_1^3 - s_3^3 \right) + 1 \right) \times P^*$$

**Define $P_r^2$.** Let $n_1 = |P_r^1|$. $P_r^2$ consists of as many copies of $\mathcal{L}(\mathcal{A})$ as possible, i.e.

$$P_r^2 = \left\lfloor \frac{n - n_1}{m!} \right\rfloor \times \mathcal{L}(\mathcal{A})$$

**Define $P_r^3$.** $P_r^3$ consists of multiple copies of pairs of rankings defined as follows.

$$P_r^3 = \left( \frac{n - n_1 - |P_r^2|}{2} \right) \times \{[1 \succ 2 \succ 3 \succ 4 \succ \text{others}], [2 \succ 1 \succ 4 \succ 3 \succ \text{others}]\}$$

It is not hard to verify that $P_r = P_r^1 + P_r^2 + P_r^3$ share the same properties as $P^*$: $\bar{r}(P_r) = \{1, 2\}$; if $[4 \succ 2 \succ 1 \succ 3 \succ \text{others}]$ is removed, then 2 is the unique winner; and if $[3 \succ 1 \succ 2 \succ 4 \succ \text{others}]$ is removed, then 1 is the unique winner. This means that PAR is violated at $P_r$.

**Step 2: define a polyhedron $\mathcal{H}^{\bar{r}}$ to represent profiles that satisfy Condition 1.** To define $\mathcal{H}^{\bar{r}}$, we recall from Definition 14 that for any $a, b$, any $B \subseteq \mathcal{A} \setminus \{a, b\}$, and any profile $P$, $\text{Score}_{B,a,b}^{\Delta} \cdot \text{Hist}(P)$ is the difference between the $\bar{r}_{m-|B|}$ score of $a$ and the $\bar{r}_{m-|B|}$ score of $b$ in $P|_{\mathcal{A} \setminus B}$. We are now ready to define $\mathcal{H}^{\bar{r}}$ whose $\mathbf{A}$ matrix has five parts that correspond to Condition 1 (1)–(5). Condition 1 (6) will be incorporated in the $\vec{\mathbf{b}}$ vector of $\mathcal{H}^{\bar{r}}$.

**Definition 31 ($\mathcal{H}^{\bar{r}}$).** *Given $\bar{r} = (\bar{r}_2, \ldots, \bar{r}_m)$, we let* $\mathbf{A}^{\bar{r}} = \begin{bmatrix} \mathbf{A}^{(1)} \\ \mathbf{A}^{(2)} \\ \mathbf{A}^{(3)} \\ \mathbf{A}^{(4)} \\ \mathbf{A}^{(5)} \end{bmatrix}$, *where*

- $\mathbf{A}^{(1)}$: *for every $1 \leq i \leq m - 4$ and every $j \in \mathcal{A} \setminus \{i + 4\}$, $\mathbf{A}^{(1)}$ has a row $\text{Score}_{\{5,\ldots,i+3\},i+4,j}^{\Delta}$.*

- $\mathbf{A}^{(2)}$, $\mathbf{A}^{(3)}$, *and* $\mathbf{A}^{(4)}$ *are defined as follows.*

$$\mathbf{A}^{(2)} = \begin{bmatrix} Score^{\triangle}_{\{5,\ldots,m\},2,1} \\ Score^{\triangle}_{\{5,\ldots,m\},3,2} \\ Score^{\triangle}_{\{5,\ldots,m\},4,3} \\ Score^{\triangle}_{\{5,\ldots,m\},3,4} \end{bmatrix}, \mathbf{A}^{(3)} = \begin{bmatrix} Score^{\triangle}_{\{3,5,\ldots,m\},4,1} \\ Score^{\triangle}_{\{3,5,\ldots,m\},2,4} \\ Score^{\triangle}_{\{2,3,5,\ldots,m\},4,1} \end{bmatrix}, \mathbf{A}^{(4)} = \begin{bmatrix} Score^{\triangle}_{\{4,5,\ldots,m\},3,2} \\ Score^{\triangle}_{\{4,5,\ldots,m\},1,3} \\ Score^{\triangle}_{\{1,4,5,\ldots,m\},3,2} \end{bmatrix}$$

- $\mathbf{A}^{(5)}$ *consists of two rows defined as follows.*

$$\mathbf{A}^{(5)} = \begin{bmatrix} -Hist(4 \succ 2 \succ 1 \succ 3 \succ others) \\ -Hist(3 \succ 1 \succ 2 \succ 4 \succ others) \end{bmatrix}$$

*Let* $\qquad \vec{\mathbf{b}^{r}} = [\underbrace{\vec{\mathbf{b}}^{(1)}}_{for\ \mathbf{A}^{(1)}}, \underbrace{(s_4^4 - s_1^4 - 1, s_4^4 - s_1^4 - 1, 0, 0)}_{for\ \mathbf{A}^{(2)}}, \underbrace{(s_3^3 - s_1^3 - 1, s_3^3 - s_1^3 - 1, s_2^2 - s_1^2 - 1)}_{for\ \mathbf{A}^{(3)}},$

$$\underbrace{(s_3^3 - s_1^3 - 1, s_3^3 - s_1^3 - 1, s_2^2 - s_1^2 - 1)}_{for\ \mathbf{A}^{(4)}}, \underbrace{(-1, -1)}_{for\ \mathbf{A}^{(5)}}],$$

*where for every* $1 \le i \le m-4$ *and every* $j \in \mathcal{A} \setminus \{i+4\}$, $\vec{\mathbf{b}}^{(1)}$ *contains a row* $s_{m+1-i}^{m+1-i} - s_1^{m+1-i} - 1$. *Let*

$$\mathcal{H}^{\overline{r}} = \left\{ \vec{x} \in \mathbb{R}^{m!} : \mathbf{A}^{\overline{r}} \cdot (\vec{x})^{\top} \le \left( \vec{\mathbf{b}^{r}} \right)^{\top} \right\}.$$

**Step 3: Apply Lemma 3 and [55, Theorem 2].** We first prove the following properties of $\mathcal{H}^{\overline{r}}$.

**Claim 14 (Properties of $\mathcal{H}^{\overline{r}}$).** *Given any integer MRSE rule* $\overline{r}$,

- *(i) for any integral profile* $P$, *if* $Hist(P) \in \mathcal{H}^{\overline{r}}$ *then* $\mathrm{PAR}(r, P) = 0$;

- *(ii)* $\pi_{uni} \in \mathcal{H}^{\overline{r}}_{\le 0}$;

- *(iii)* $\dim(\mathcal{H}^{\overline{r}}_{\le 0}) = m! - 1$.

*Proof.* Part (i) follows after a similar reasoning as in Step 1 of the proof of Theorem 5. To prove Part (ii), notice that for any $B \subseteq \mathcal{A}$ and $a, b \in (\mathcal{A} \setminus B)$, we have $Score^{\triangle}_{B,a,b} \cdot \vec{1} = 0$. Also notice that for any $R \in \mathcal{L}(\mathcal{A})$ we have $-Hist(R) \cdot \vec{1} = -1 < 0$. Therefore, $\mathbf{A}^{\overline{r}} \cdot \left( \vec{1} \right)^{\top} \le \left( \vec{0} \right)^{\top}$, which means that $\pi_{\mathrm{uni}} \in \mathcal{H}^{\overline{r}}_{\le 0}$. To prove Part (iii), notice that $\mathbf{A}^{\overline{r}} \cdot (\vec{x})^{\top} \le \left( \vec{0} \right)^{\top}$ contains one equality in $\mathbf{A}^{(2)}$, i.e.

$$Score^{\triangle}_{\{5,\ldots,m\},3,4} \cdot (\vec{x})^{\top} = 0 \qquad (15)$$

This means that $\dim(\mathcal{H}^{\overline{r}}_{\le 0}) \le m! - 1$. Recall that $P_r$ is the $n$-profile defined in Step 1 that satisfies Condition 1. Notice that $Hist(P_r)$ is an inner point of $\mathcal{H}^{\overline{r}}_{\le 0}$ in the sense that all inequalities in $\mathbf{A}^{\overline{r}} \cdot (\vec{x})^{\top} \le \left( \vec{0} \right)^{\top}$ except Equation (15) are strict, which means that $\dim(\mathcal{H}^{\overline{r}}_{\le 0}) \ge m! - 1$. This proves Claim 14. $\qquad \square$

Because of the existence of $P_r$ defined in Step 1, and Claim 14 (i) and (ii), the 1 case and the VL case of Lemma 3 do not hold for any sufficiently large $n$. Therefore, it follows from the L case of Lemma 3 that $\widetilde{\mathrm{PAR}}_{\Pi}^{\min}(r, n)$ is at least $1 - O(n^{-0.5})$, because $\ell_n \ge 1$. It remains to show that $\widetilde{\mathrm{PAR}}_{\Pi}^{\min}(r, n)$ is upper-bounded by $1 - \Omega(n^{-0.5})$. We have the following calculations.

$$\begin{aligned} 1 - \widetilde{\mathrm{PAR}}_{\Pi}^{\min}(r, n) &= \sup_{\vec{\pi} \in \Pi^n} \mathrm{Pr}_{P \sim \vec{\pi}}(\mathrm{PAR}(r, P) = 0) \\ &\ge \sup_{\vec{\pi} \in \Pi^n} \mathrm{Pr}_{P \sim \vec{\pi}}(Hist(P) \in \mathcal{H}^{\overline{r}}) \qquad\qquad \text{Claim 14 (i)} \\ &= \Theta(n^{-0.5}) \qquad\qquad \text{Claim 14 (ii), (iii), and [55, Theorem 2]} \end{aligned}$$

The last equation follows after applying the sup part of [55, Theorem 2] to $\mathcal{H}^{\overline{r}}$. More concretely, recall that in Step 1 above we have constructed an $n$-profile $P_r$ for any sufficiently large $n$ and it is not hard to verify that $\text{Hist}(P_r) \in \mathcal{H}^{\overline{r}}$, which means that $\mathcal{H}^{\overline{r}}$ is active at any sufficiently large $n$. Claim 14 (ii) implies that the polynomial case of [55, Theorem 2] holds, and Claim 14 (iii) implies that $\alpha_n = m! - 1$ for $\mathcal{H}^{\overline{r}}$.

This proves Theorem 5. $\qquad\qquad\qquad\qquad\qquad\qquad\qquad\qquad\qquad\qquad\qquad\qquad\qquad$ $\square$

### F.5    Proof of Theorem 6

**Theorem 6 (Semi-random PAR: Condorcetified Integer Positional Scoring Rules).** *Given $m \geq 4$, an integer positional irresolute scoring rule $\overline{r}_{\vec{s}}$, any Condocetified positional scoring rule $Cond_{\vec{s}}$ that is a refinement of $\overline{Cond_{\vec{s}}}$, and any strictly positive and closed $\Pi$ over $\mathcal{L}(\mathcal{A})$ with $\pi_{uni} \in CH(\Pi)$, there exists $N \in \mathbb{N}$ such that for every $n \geq N$,*

$$\widetilde{\text{PAR}}_{\Pi}^{\min}(Cond_{\vec{s}}, n) = 1 - \Theta(\frac{1}{\sqrt{n}})$$

*Proof.* The proof follows the same logic in the proof of Theorem 5. We first prove the theorem for even $n$ then show how to extend the proof to odd $n$'s.

**Intuition for $2 \mid n$.**    Let $\vec{s} = (s_1, \ldots, s_m)$. We first identify a set of sufficient conditions for PAR to be violated.

**Condition 2 (Sufficient conditions for the violation of PAR).** *Given a Condorcetified irresolute integer positional scoring rule $\overline{Cond_{\vec{s}}}$, $P$ satisfies the following conditions.*

- *(1) $\overline{Cond_{\vec{s}}}(P) = \{2\}$, and the score of $2$ is higher than the score of any other alternative by at least $s_1 - s_m + 1$.*

- *(2) Alternative $1$ is a weak Condorcet winner, $w_P(1,3) = 0$, and for every $i \in \mathcal{A} \setminus \{1,3\}$, $w_P(1,i) \geq 2$.*

- *(3) $P$ contains at least one vote of $[3 \succ 1 \succ 2 \succ \text{others}]$.*

Recall that $Cond_{\vec{s}}$ is a refinement of $\overline{Cond_{\vec{s}}}$ and due to Condition 2 (2), $P$ does not contain a Condorcet winner. Therefore, according to Condition 2 (1), we have $Cond_{\vec{s}} = \{2\}$. Any voter whose preferences are $[3 \succ 1 \succ 2 \succ \text{others}]$ has incentive to abstain from voting, because the voter prefers $1$ to $2$, and $\{1\}$ is the Condorcet winner in $P - [3 \succ 1 \succ 2 \succ \text{others}]$, which means that

$$\text{Cond}_{\vec{s}}(P - [3 \succ 1 \succ 2 \succ \text{others}]) = \{1\}$$

This means that $\text{PAR}(\text{Cond}_{\vec{s}}, P) = 0$ for any profile $P$ that satisfies Condition 2. The rest of the proof proceeds as follows. In Step 1, for any $n$ that is sufficiently large, we construct an $n$-profile $P_{\vec{s}}$ that satisfies Condition 2. Then in Step 2, we formally define $\mathcal{H}^{\overline{\text{Cond}_{\vec{s}}}}$ to represent profile that satisfy Condition 2. Finally, in Step 3 we formally prove properties about $\mathcal{H}^{\overline{\text{Cond}_{\vec{s}}}}$ and apply Lemma 3 and [55, Theorem 2] to prove Theorem 5.

**Step 1 for $2 \mid n$: define $P_{\vec{s}}$.**    The construction is similar to the construction in the proof of Claim 10, which is done for the following two cases: $\overline{r}_{\vec{s}}$ is the plurality rule and $\overline{r}_{\vec{s}}$ is not the plurality rule.

- **When $\overline{r}_{\vec{s}}$ is the plurality rule,** i.e. $s_2 = s_m$, we let

$$P_{\vec{s}} = \left(\frac{n}{2} - 6\right) \times [2 \succ 1 \succ 3 \succ \text{others}] + 4 \times [2 \succ 3 \succ 1 \succ \text{others}]$$
$$+ \left(\frac{n}{2} - 6\right) \times [3 \succ 1 \succ 2 \succ \text{others}] + 6 \times [1 \succ 2 \succ 3 \succ \text{others}]$$

It is not hard to verify that $P_{\vec{s}}$ satisfies Condition 2 for any even number $n \geq 28$.

- **When $\overline{r}_{\vec{s}}$ is not the plurality rule,** i.e., $s_2 > s_m$, like Step 1 in the proof of Theorem 5, we first construct a profile $P^*$ that consists of a constant number of votes and satisfies Condition 2, then extend it to arbitrary odd number $n$. Let $2 \le k \le m-1$ denote the smallest number such that $s_k > s_{k+1}$. Let $A_1 = [4 \succ \cdots \succ k+1]$ and $A_2 = [k+2 \succ \cdots \succ m]$, and let $P^* = P_1^* + P_2^*$, where $P_1^*$ is the following 10-profile that is used to guarantee Condition 2 (2) and (3).

$$P_1^* = \{4 \times [1 \succ 2 \succ A_1 \succ 3 \succ A_2] + 3 \times [2 \succ 3 \succ A_1 \succ 1 \succ A_2]$$
$$+2 \times [3 \succ 1 \succ A_1 \succ 2 \succ A_2] + [2 \succ 1 \succ A_1 \succ 3 \succ A_2]\}$$

And let $P_2^*$ denote the following $36(m-3)!$-profile, which is used to guarantee that 2 is the unique winner under $P^*$, i.e., Condition 2 (1).

$$P_2^* = 6 \times \{[R_1 \succ R_2] : \forall R_1 \in \mathcal{L}(\{1,2,3\}), R_2 \in \mathcal{L}(\{4,\ldots,m\}),\}$$

It is not hard to verify that the following observations hold for $P_1^*$.

- 1 is the Condorcet winner, $w_{P_1^*}(1,3) = 0$, and for any $i \in \mathcal{A} \setminus \{1,3\}$, we have $w_{P_1^*}(1,i) \ge 2$.
- The total score of 1 under $P_1^*$ is $4s_1 + 3s_2 + 3s_{k+1}$, the total score of 2 under $P_1^*$ is $4s_1 + 4s_2 + 2s_{k+1}$, and the total score of 3 under $P_1^*$ is $2s_1 + 3s_2 + 5s_{k+1}$. Recall that we have assumed that $s_2 > s_{k+1}$. Therefore,

$$4s_1 + 4s_2 + 2s_{k+1} > 4s_1 + 3s_2 + 3s_{k+1} > 2s_1 + 3s_2 + 5s_{k+1},$$

which means that the score of 2 is strictly higher than the scores of 1 and 3 in $P_1^*$.

Given these observations, it is not hard to verify that $P^* = P_1^* + P_2^*$ satisfies Condition 2. Let $P_{\vec{s}}$ denote as many copies of $P^*$ as possible, plus pairs of rankings $\{[2 \succ 1 \succ 3 \succ \text{others}], [2 \succ 3 \succ 1 \succ \text{others}]\}$. More precisely, let

$$P_{\vec{s}} = \left\lfloor \frac{n}{|P^*|} \right\rfloor \times P^* + \left( \frac{n - |P^*| \cdot \lfloor \frac{n}{|P^*|} \rfloor}{2} \right) \times \{[2 \succ 1 \succ 3 \succ \text{others}], [2 \succ 3 \succ 1 \succ \text{others}]\}$$

It is not hard to verify that $P_{\vec{s}}$ satisfies Condition 2, which concludes Step 1 for the $2 \mid n$ case.

**Step 2 for $2 \mid n$: define a polyhedron $\mathcal{H}^{\overline{\text{Cond}_{\vec{s}}}}$ to represent profiles that satisfy Condition 2.**

**Definition 32 ($\mathcal{H}^{\overline{\text{Cond}_{\vec{s}}}}$).** *Given an irresolute integer positional scoring rule $\overline{r}_{\vec{s}} = (s_1, \ldots, s_m)$, we let* $\mathbf{A}^{\vec{s}} = \begin{bmatrix} \mathbf{A}^{(1)} \\ \mathbf{A}^{(2)} \\ \mathbf{A}^{(3)} \end{bmatrix}$, *where*

- $\mathbf{A}^{(1)}$: *for every $i \in \mathcal{A} \setminus \{2\}$, $\mathbf{A}^{(1)}$ contains a row $Score_{i,2}$.*

- $\mathbf{A}^{(2)}$ *contains two rows $Pair_{1,3}$ and $Pair_{3,1}$, and for every $i \in \mathcal{A} \setminus \{1,3\}$, $\mathbf{A}^{(1)}$ contains a row $Pair_{i,1}$.*

- $\mathbf{A}^{(3)}$ *consists of a single row $-Hist(3 \succ 1 \succ 2 \succ \text{others})$.*

$$\text{Let} \quad \mathbf{b}^{\vec{s}} = \left[ \underbrace{(s_m - s_1 - 1) \cdot \vec{1}}_{\text{for } \mathbf{A}^{(1)}}, \underbrace{(0, 0, -2, \ldots, -2)}_{\text{for } \mathbf{A}^{(2)}}, \underbrace{-1}_{\text{for } \mathbf{A}^{(3)}} \right]$$

$$\text{and} \quad \mathcal{H}^{\vec{s}} = \left\{ \vec{x} \in \mathbb{R}^{m!} : \mathbf{A}^{\vec{s}} \cdot (\vec{x})^{\top} \le \left( \mathbf{b}^{\vec{s}} \right)^{\top} \right\}.$$

**Step 3 for $2 \mid n$: Apply Lemma 3 and [55, Theorem 2].**  We first prove the following properties of $\mathcal{H}^{\overline{\mathrm{Cond}_{\vec{s}}}}$.

**Claim 15 (Properties of $\mathcal{H}^{\overline{\mathrm{Cond}_{\vec{s}}}}$).** *Given any integer positional scoring rule $\vec{s}$,*

  *(i) for any integral profile $P$, if $\mathrm{Hist}(P) \in \mathcal{H}^{\overline{\mathrm{Cond}_{\vec{s}}}}$ then $\mathrm{PAR}(\mathrm{Cond}_{\vec{s}}, P) = 0$;*

  *(ii) $\pi_{uni} \in \mathcal{H}^{\overline{\mathrm{Cond}_{\vec{s}}}}_{\leq 0}$;*

  *(iii) $\dim(\mathcal{H}^{\overline{\mathrm{Cond}_{\vec{s}}}}_{\leq 0}) = m! - 1$.*

*Proof.* The proof for Part (i) and (ii) are similar to the proof of Claim 14. To prove Part (iii), notice that $\mathbf{A}^{\vec{s}} \cdot (\vec{x})^{\top} \leq \left(\vec{0}\right)^{\top}$ contains one equality in $\mathbf{A}^{(2)}$, i.e.

$$\mathrm{Pair}_{1,3} \cdot (\vec{x})^{\top} = (0)^{\top} \tag{16}$$

This means that $\dim(\mathcal{H}^{\overline{\mathrm{Cond}_{\vec{s}}}}_{\leq 0}) \leq m! - 1$. Notice that $\mathrm{Hist}(P_{\vec{s}})$ is an inner point of $\mathcal{H}^{\overline{\mathrm{Cond}_{\vec{s}}}}_{\leq 0}$ in the sense that all other inequalities except Equation (16) are strict, which means that $\dim(\mathcal{H}^{\overline{\mathrm{Cond}_{\vec{s}}}}_{\leq 0}) \geq m! - 1$. This proves Claim 15. $\qquad\square$

Therefore, we have the following bound.

$$
\begin{aligned}
&1 - \widetilde{\mathrm{PAR}}_{\Pi}^{\min}(\mathrm{Cond}_{\vec{s}}, n)\\
&= \sup_{\vec{\pi} \in \Pi^n} \Pr_{P \sim \vec{\pi}}(\mathrm{PAR}(\mathrm{Cond}_{\vec{s}}, P) = 0)\\
&\geq \sup_{\vec{\pi} \in \Pi^n} \Pr_{P \sim \vec{\pi}}(\mathrm{Hist}(P) \in \mathcal{H}^{\overline{\mathrm{Cond}_{\vec{s}}}}) && \text{Claim 15 (i)}\\
&= \Theta(n^{-0.5}) && \text{Claim 15 (ii), (iii), and [55, Theorem 2]}
\end{aligned}
$$

Consequently, $\widetilde{\mathrm{PAR}}_{\Pi}^{\min}(\mathrm{Cond}_{\vec{s}}, n) = 1 - \Omega(n^{-0.5})$. Notice that the 1 case and VL case Lemma 3 do not hold because of the existence of $P_{\vec{s}}$ and Claim 15 (ii). Therefore, Theorem 6 for the $2 \mid n$ case follows after the $1 - O(n^{-0.5})$ upper bound proved in Lemma 3.

**Proof for the $2 \nmid n$ case.**  When $2 \nmid n$, we modify the proof as follows.

- First, Condition 2 (2) is replaced by the following condition:

  (2′): Alternative 1 is the Condorcet winner under $P$, $w_P(1,3) = 1$, and for every $i \in \mathcal{A} \setminus \{1,3\}$, $w_P(1,i) \geq 3$.

- Second, in Step 1, $P_{\vec{s}}$ has an additional vote $[2 \succ 1 \succ 3 \succ \text{others}]$.

- Third, in Step 2 Definition 32, the $\vec{b}^{\vec{s}}$ components corresponding to $\mathbf{A}^2$ is $(1, -1, -3, \ldots, -3)$.

A similar claim as Claim 15 can be proved for the $2 \nmid n$ case. This proves Theorem 6. $\qquad\square$

## G  Experimental Results

We report satisfaction of CC and PAR using simulated data and Preflib linear-order data [35] under four classes of commonly-used voting rules studied in this paper, namely positional scoring rules (plurality, Borda, and veto), voting rules that satisfy CONDORCET CRITERION (maximin, ranked pairs, Schulze, and Copeland$_{0.5}$), MRSE (STV), and Condorcetified positional scoring rule (Black's rule).  All experiments were implemented in Python 3 and were run on a MacOS laptop with 3.1 GHz Intel Core i7 CPU and 16 GB memory.

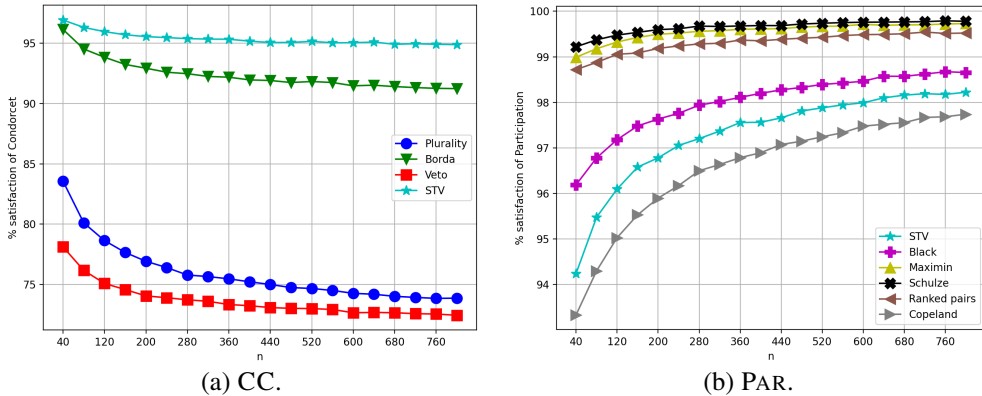

(a) CC.

(b) PAR.

Figure 13: Satisfaction of CC and PAR under IC for $m = 4$, $n = 40$ to 800, 200000 trials.

**Synthetic data.** We generate profiles of $m = 4$ alternatives under IC.[3] The number of alternatives $n$ ranges from 40 to 800. In each setting we generate 200000 profiles. The satisfaction of CC under plurality, Borda, veto, and STV are presented in Figure 13 (a), and the satisfaction of PAR under STV, maximin, ranked pairs, Schulze, Black, and Copeland$_{0.5}$ are presented in Figure 13 (b). Notice that voting rules not in Figure 13 (a) always satisfy CC and voting rules not in Figure 13 (b) always satisfy PAR.

The results provide a sanity check for the theoretical results proved in this paper. In particular, Figure 13 (a) confirms that the satisfaction of CC is $\Theta(1)$ and $1 - \Theta(1)$ under positional scoring rules (Theorem 1) and STV (Corollary 1) w.r.t. IC. Figure 13 (b) confirms that the satisfaction of PAR is $1 - \Theta(n^{-0.5})$ under maximin, ranked pairs, Schulze (Theorem 3), Copeland$_\alpha$ (Theorem 4), STV (Theorem 5), and Black (Theorem 6). Figure 14 in Appendix G summarizes results with large $n$ (1000 to 10000) that further confirm the asymptotic observations described above.

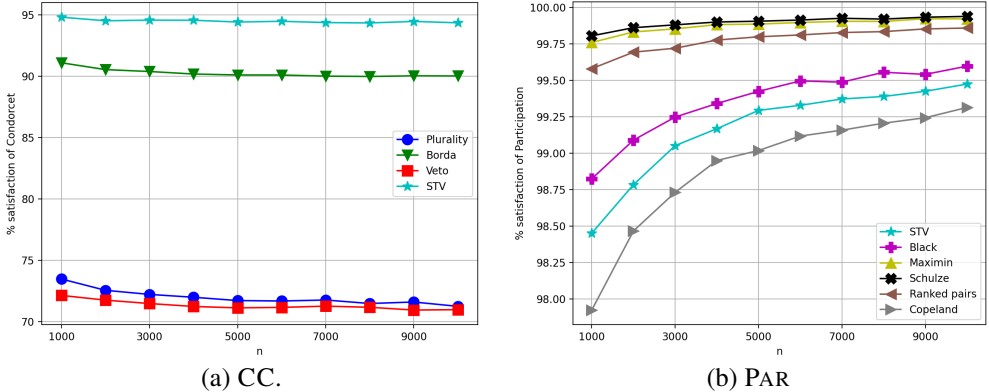

(a) CC.

(b) PAR

Figure 14: Satisfaction of CC and PAR under IC for $m = 4$, $n = 1000$ to 10000, 200000 trials.

**Preflib data.** We also calculate the satisfaction of CC and PAR under all voting rules studied in this paper with lexicographic tie-breaking for all 315 Strict Order-Complete Lists (SOC) under election data category from Preflib [35]. The results are summarized in Table 7, which is the bottom part of Table 2.

Table 7 delivers the following message, that PAR is less of a concern than CC in Preflib data—all voting rules have close to 100% satisfaction of PAR, while the satisfaction of CC is much lower for plurality, Borda, and Veto. The most interesting observations are: first, maximin, Schulze, and ranked pairs achieve 100% satisfaction of CC and PAR in Preflib data, which is consistent with the belief that Schulze and ranked pairs are superior in satisfying voting axioms, and maximin is doing well in PAR (and indeed, maximin satisfies PAR when $m = 3$). Second, STV does well in CC

---

[3]See [10] for theoretical results and extensive simulation studies of PAR under the IAC model.

Table 7: Satisfaction of CC and PAR in 315 Preflib SOC profiles. Some statistics of the data are shown in Figure 15.

| | Plurality | Borda | Veto | STV | Black | Maximin | Schulze | Ranked pairs | Copeland$_{0.5}$ |
|---|---|---|---|---|---|---|---|---|---|
| CC | 96.8% | 92.4% | 74.2% | 99.7% | 100% | 100% | 100% | 100% | 100% |
| PAR | 100% | 100% | 100% | 99.7% | 99.4% | 100% | 100% | 100% | 99.7% |

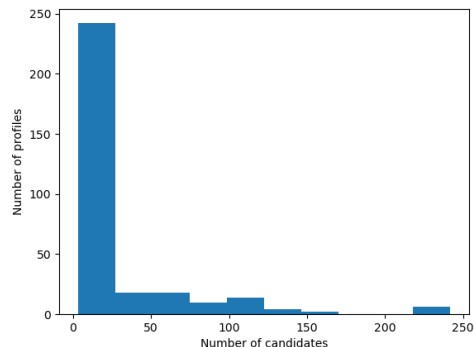 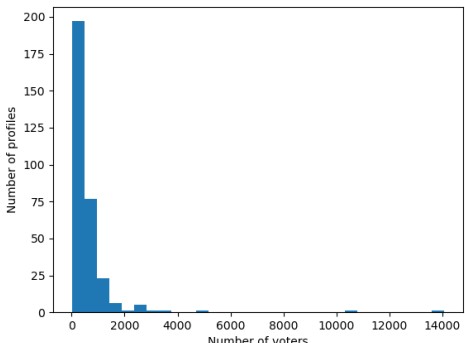

Figure 15: Histograms of number of candidates and number of voters in the 315 Preflib SOC data studied in this paper.

and PAR, though it does not satisfy either in the worst case. Third, veto has poor satisfaction of CC (74.2%), which is mainly due to the profiles where the number of alternatives is more than the number of voters, so that a Condorcet winner exists and is also a veto co-winner, but loses due to the tie-breaking mechanism.