# OpenReview forum: "The Semi-Random Satisfaction of Voting Axioms"
_NeurIPS.cc/2021/Conference — NeurIPS 2021 Spotlight_

### Official Review · Reviewer_mywX · 2021-07-16

**Rating:** 7
**Confidence:** 3

**Summary:**

The authors study the likelihood that two voting axioms -- the Condorcet criterion and participation -- will be satisfied when an adversary chooses an arbitrary set of preferences of the agents, and then these preferences are perturbed with random noise. They establish two main theorems that show that the likelihood that each axiom is satisfied under smooth analysis is very case-dependent, but that the Condorcet criterion is generally less likely to be satisfied than participation.



**Ethical Concerns:**

None.

**Limitations And Societal Impact:**

Yes.

**Main Review:**

On a high level, I agree that voting axioms are generally quite rigid and oftentimes too coarse. It is also definitely discouraging sometimes to only prove impossibility results. In this sense, smoothed analysis makes sense for studying the satisfiability of voting axioms under more realistic models of vote generation, and I appreciate the approach taken by the authors in this paper.

In general, I found the technical exposition to be quite dense and incredibly notation-heavy. I appreciated the efforts of the authors to provide high-level intuition for many of the results, and the tables really help the reader.

Overall, I was very impressed (and daunted) by the technical content of the paper (really the appendix, I guess, given the page limit), and I think the authors did a good job cleaning up the presentation and adding intuition for a shorter format. I do think the best fit for this work is in a social choice journal, given the scope of the work, but I do not think it is outside the scope of NeurIPS.



**Time Spent Reviewing:**

2

---

> ### Author Response · Authors · 2021-08-10
> **Thanks for your review!**
>
> We appreciate the reviewer’s time and thoughtful feedback! We will try to further improve the presentation by adding more discussions and moving some technical contents to the appendix. We also plan to submit the full version of the paper to a journal afterwards.

---

### Official Review · Reviewer_mLZB · 2021-07-16

**Rating:** 8
**Confidence:** 4

**Summary:**

This paper studies the smoothed satisfaction of voting axioms, which is a novel notion to quantify the dissatisfaction of voting axioms based on smoothed analysis. This paper aims to address the following open question: "How likely are voting axioms satisfied under realistic models?"

The main contribution of this work is that they propose the framework of smoothed satisfaction and give pretty general quantitative results of dissatisfaction of various voting axioms for different voting rules. In particular, their results imply two important but contradicting axioms (under worst-case analysis): Condorcet criterion (CC) and Participation both can be actually satisfied in many realistic settings.


**Limitations And Societal Impact:**

The authors adequately addressed the limitation

**Main Review:**

The smoothed social choice framework was first proposed by Xia, which is a mid-point between the impartial culture setting and the traditional worst-case analysis. In this paper, they use this framework to define smoothed satisfaction and analyze how likely voting axioms satisfied under realistic settings. I believe it is a very important question to ask, and it is a sweet spot for smoothed analysis. This paper demonstrates that some axioms which cannot coexist in worst-case analysis can be satisfied simultaneously in some realistic settings.

Major technical contribution of this paper is that authors show several pretty general quantitative results of how likely various voting axioms could coexist in realistic setting. Having that said, I do think the writing of the technical part can be improved. I feel it is just too dense, notations are a bit overwhelming.

Overall, it is a very strong paper, it addresses a long-standing (and important) open problem and demonstrate the power of smoothed analysis. The highlight of this paper is that it proposes a pretty general (and sound) framework to analyze satisfaction of voting axioms in somewhat realistic settings. This paper also shows quantitative results for two important axioms (Condorcet and Participation) under many important voting rules. The major concern is that this paper might be too technical to follow, I would strongly recommend the authors to make the body less technical, front-loading some intuition and maybe dumping some results into appendix may help.

typos:
1) Line 28, "exist" -> "exists"

**Time Spent Reviewing:**

4

---

> ### Author Response · Authors · 2021-08-10
> **Thanks for your review!**
>
> We appreciate the reviewer’s time and thoughtful feedback! We will try to further improve the presentation by adding more discussions and moving some technical contents to the appendix (as other reviewers also suggested).

---

> > ### Comment · Reviewer_mLZB · 2021-08-30
> > **Thanks for the response**
> >
> > Thanks for your response! I have no further questions and I will keep my score.

---

### Official Review · Reviewer_Ck6b · 2021-07-19

**Rating:** 7
**Confidence:** 3

**Summary:**

The paper builds up on a semi-random model for preference profiles by Xia. For a constant number of alternatives and a number of agents going to infinity, the authors study how likely different voting rules violate Condorcet consistency / participation on the semi-randomly drawn preference profile.

**Limitations And Societal Impact:**

I think that more could be said in terms of the limitations of the paper: To which degree can and should one see the model as a proxy for preference profiles in practice? What are examples for properties that cannot be phrased as the union of finitely many polyhedra (i.e., for which Lemma 1 does not apply)?

**Main Review:**

The paper contains a trove of fascinating results. The main results characterize integer scoring rules and their asymptotic behavior with respect to Condorcet consistency and participation, and the appendix give a range of additional results characterizing participation for other prominent voting rules. The paper makes a convincing case in the introduction that these (fairly abstract) characterizations have plenty of implications that social-choice researchers should care about, such as new results for participation under impartial culture and general take-aways for the relative likelihood of violations of the two properties (where it is known that Condorcet consistency and participation cannot be simultaneously guaranteed for 4 and more items).
At least as important is Lemma 1, an abstracted version of the results by Xia 2021, which builds the foundation of the previous results, but is also clearly applicable to axioms other than participation and Condorcet consistency. I cannot judge how much of that generality was already present in Xia 2021, but, if this is indeed new, this seems like a big and valuable contribution that can likely draw a whole string of papers behind it.

A first comment that I have is that I think the term "smoothed analysis" is a misleading description of what the paper does. I realize, of course, that this terminology is taken over from Xia 2020 (where another reviewer pointed out the same thing [1]), but I think it would be a service to the community to adapt the terminology. In smoothed analysis as introduced by Spielman and Teng, (a) a worst-case instance is randomly perturbed and (b) results are analyzed as a function of both n and sigma, the variance of the noise. By contrast, Xia's model presumes a menu of distributions and then has an adversary combine these distributions; and the paper does not analyze the dependency on the "noise" (here: the epsilon in the definition of a strictly-positive single-agent preference model).
I think a better terminology would be to call the model a semi-random model since it interleaves worst-case and probabilistic steps. In the literature on semi-random models, adversarial and probabilistic steps have been interleaved in many different ways (see, for example, Section 2.3 of this paper by Kolla, Makarychev, and Makarychev [2]). None of this should be understood as a critique of the model, and I think that the model indeed is useful for similar reasons as a smoothed-analysis model, which makes sense that both models are semi-random. Still, the term smoothed analysis is misleading and should be avoided.

My second concern is with the presentation of the paper. My impression is that the nine-page format does this paper a real disservice. The authors clearly know their craft, but the space constraints riddle the paper with abbreviations, quick sequences of theorem environments, and a lot of the paper "tells rather than shows", which is natural given the 60 pages of theoretical appendices and the fact that Theorems 3–6 don't even fit into the body. While reading, I was missing more signposting of what was up ahead, and more patient discussions relating the results (the introduction does this very well, I just would have liked more of it). Within these constraints, the exposition is clear, but it is a harder read than it would have to be with extra pages, and I am concerned that there is to little space to reflect on the results and make them shine. I don't think this is reason enough to reject the paper, but I think it severely limits the usefulness of the paper as printed at NeurIPS. I strongly encourage the authors to publish a longer, more patient version on arxiv/in a journal to maximize the impact of the work.


Smaller comments:
- I think \Theta(1) /\ 1 - Theta(1) should be spelled out somewhere: for large enough n, the probabilities will be bounded by some constants alpha, beta with 0 < alpha < beta < 1.
- In Figure 1, the formulation "other ranking w.p. 1/8" was very misleading to me. After reading a section coming much later, I think this is supposed to mean that "every other ranking" has probability 1/8 (the adding-up-to-1 constraint was less obvious to me given that your weighted preference profiles don't have normalized weights). Moreover, I'm pretty sure that the weighted majority graph should have weights 1/8 rather than 1/4.
- in line 254, it wasn't clear to me what in the condition corresponds to the "small perturbation", this deserves more details.
- based on the introduction, I would have expected Lemma 1 to appear before the main results rather than after. I think this should be signposted in the sections where the main results appear.
- the first paragraph of Section 3 did little to explain what was the goal of the section, and instead paraphrased the immediately following definition 2. This paraphrase threw me off rather than helping me.
- a whole range of abbreviations were undefined at first use (there might be some false positives): ANR in line 142, MRSE is used in line 181 before being introduced in line 190, CH in line 240, PMV in line 324 is only defined in line 361.
- in the equation in line 240, for example, many sets are intersected with A \setminus r(\pi). It seems easier to just subtract the set r(\pi), which is equivalent.
- typo: "Condocetified" in line 336, "sutdied" in line 379

––
[1] https://papers.nips.cc/paper/2020/file/7e05d6f828574fbc975a896b25bb011e-Review.html
[2] https://arxiv.org/pdf/1104.3806.pdf

**Time Spent Reviewing:**

5

---

> ### Author Response · Authors · 2021-08-10
> **Thanks for your review!**
>
> We appreciate the reviewer’s time and thoughtful feedback!
>
> Re. First concern. The reviewer is absolutely correct about the difference between the setting of smoothed analysis by Spielman and Teng and the setting of this paper. Semi-random model is an awesome name (thanks for the suggestion!). We will make the change throughout the paper, and specifically the title will be changed to “The Semi-Random Satisfaction of Voting Axioms”. We will also add discussions on its connection to smoothed analysis and why we used a different terminology than Xia 2020. As a side note, at a high level our setting can be translated to the setting by Spielman and Teng, by modeling the input by the normalized anonymized profile (i.e., the histogram of an n-profile divided by n), and then viewing the noise as (approximated) Gaussian with diminishing sigma at the rate of \sqrt n, in light of various multi-dimensional Berry-Esseen-type theorems. That said, we found the semi-random name suggested by the reviewer a better alternative. Thanks for the suggestion again!
>
> Re. Second concern. We will submit the full version to a journal and put it online as the suggested. We will also proofread and polish the paper (including addressing all “Smaller concerns” raised by the reviewer) to further optimize the presentation under the page limit.
>
> Re. Limitations. We will add more discussions as mentioned in our response to Reviewer XDdy.
>
> Side comment on Lemma 1 vs. Xia 2021. Technically, Lemma 1 is a somewhat straightforward application and combination of  the main theorem in Xia 2021 to the union of finitely many polyhedra and their complements. We believe that it’s main contribution is conceptual (as briefly discussed in L145 and L369), because as the reviewer observed, it provides a high-level picture on what can be expected about the satisfaction of other axioms (while the direct applications of the main result in Xia 2021 would not distinguish the three Theta(1) cases in Lemma 1 of this paper, i.e., M, L, and VL). The main technical contribution of this paper is the applications of Lemma 1 to CC and Par (instead of proving Lemma 1 itself).

---

> > ### Comment · Reviewer_Ck6b · 2021-08-27
> > **Post rebuttal**
> >
> > I thank the authors for their response, and I’m excited to see the journal version of the paper!

---

### Official Review · Reviewer_XDdy · 2021-07-20

**Rating:** 7
**Confidence:** 3

**Summary:**

This paper addressed a comprehensive picture of smoothed satisfaction of voting axioms
under commonly-studied voting rules, by focusing on CC and PAR (Condorcet criterion and participation).

**Limitations And Societal Impact:**

Not addressed.

**Main Review:**

The theoretical contribution of the paper is sound and overall comprehensive to parse. Under certain assumptions, the main results derive the smoothed satisfaction of Condorcet Criterion (for m>3 and sufficiently large n) and PAR, under a wide range of voting rules.

There should be a discussion on how brittle the analyses are under the imposed assumptions (Thm 1 to 6), to what generality the results can be argued upon lifting them? How the current analyses could be extended to smoothed satisfaction of other voting axioms?

**Time Spent Reviewing:**

~3-4

---

> ### Author Response · Authors · 2021-08-10
> **Thanks for your review!**
>
> We appreciate the reviewer’s time and thoughtful feedback!
>
> Re. Generality of results. We believe that the assumptions (of strict positiveness and closedness) are mild, though it is definitely an important and interesting open question to extend the analysis to variable \epsilon, and perhaps establish bounds that explicitly depend on \epsilon.
>
> Re. Other axioms. The analysis of this paper (applications of Lemma 1) can be performed for any axiom under any rule that can be represented by the union of finitely many polyhedral, which we believe to cover a wide range of situations. Section 4 intended to demonstrate this possibility, and some observations and discussions were made in Appendix B (where the definitions of some axioms naturally lead to their polyhedral representations, while some may not have straightforward polyhedral representations). We believe that even for axioms in Appendix B that admit natural polyhedral representations under some commonly-studied voting rules, developing conditions and characterizing the degree of polynomial like in the theorems in this paper can still be highly challenging and non-trivial even in light of Lemma 1. These are natural and interesting topics for future work.
>
> We will add more concrete and explicit discussions about the limitations and possible generalizations to other axioms in the revision.

---

> > ### Comment · Reviewer_XDdy · 2021-08-27
> > **Post rebuttal**
> >
> > The authors' responses are satisfactory, I keep my score unchanged.

---

### Decision · Program_Chairs · 2021-09-27

**Decision:**

Accept (Spotlight)

**Comment:**

This paper gives several interesting results on the smoothed satisfaction of voting axioms. I, and all of the reviewers, have a positive view of the paper. I recommend acceptance.